# FROM COMPARISON TO COMPOSITION: TOWARDS UNDERSTANDING MACHINE COGNITION OF UNSEEN CATEGORIES

## ABSTRACT

Humans are known to acquire and generalize visual concepts through a natural compare–then–compose process. We ask whether this mechanism can provide principled conditions under which machines generalize existing knowledge to unseen categories. In this work, we formalize cognition of the unseen as two complementary mechanisms for deep learning models: *comparison*, which uncovers latent concepts by capturing cross-category variations among seen classes, and *composition*, which extrapolates these concepts continuously to unseen classes. Even without parametric assumptions, we establish identifiability guarantees for learning latent concepts and unseen categories via sufficient contrast and independent support separation, denoted as **C**omparison–**C**omposition **C**ognition ($\mathbf{C}^3$). Guided by these results, we instantiate a structurally constrained generative model mirroring our theoretical assumptions. Our results on simulated data corroborate our theoretical claims and the effectiveness of our proposed methodology. In the setting of visual cognition with unseen labels, aka *On-the-fly Category Discovery*, our instantiated approach improves state-of-the-art baselines by +3.8% average accuracy across fine-grained benchmarks. We hope that the $\mathbf{C}^3$ framework mirrors human cognition to practical guidance for *representational compositionality*, illuminating how and why machines can generalize to unseen categories.

## 1 INTRODUCTION

Despite impressive conventional performance (Krizhevsky et al., 2012), recognition systems remain plagued by semantic generalization failures in open-world conditions with test-time novel classes (Zhu et al., 2024; Udandarao et al., 2024; Shu et al., 2018). Moreover, recent investigations demonstrate that scaled foundation models do not resolve generalization beyond training semantics (Udandarao et al., 2024; Zhang et al., 2024d; Conti et al., 2025; Han et al., 2023; Zhang et al., 2024c).

In contrast, seminal cognitive and neuroscience studies show that humans can recognize unseen categories by transferring primitive visual concepts acquired from known ones (Rosch, 1973; Biederman, 1987; Gentner, 1983; Gentner et al., 2007). A large body of work characterizes this ability as a *compare–then–compose* process: structure-mapping theory models comparison as *structural alignment* of salient relational features across instances (Gentner, 1983), while Marr's vision theory on visual primitives / 2.5D sketch

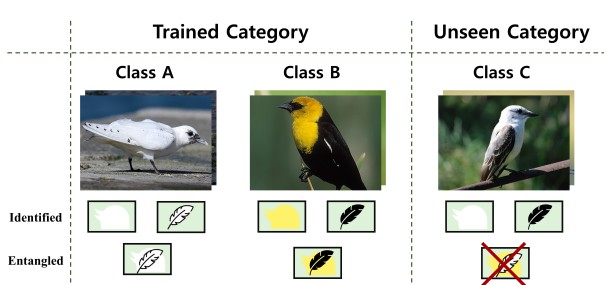

Figure 1: An illustrative example of the semantic cognition process with identified concepts v/s entangled features.

and Biederman's recognition-by-components emphasize recognition via compositions of reusable parts (Marr, 2010; Biederman, 1987). This perspective suggests that humans first identify concept primitives that reliably distinguish categories, and then cognize novel categories as new combinations of these primitives. However, modern deep networks often learn entangled features plagued by

spurious correlations (Izmailov et al., 2022), which undermines such recomposition capability and interpretability: as illustrated in Fig. 1, features such as *white head* and *black wings* may be spuriously entangled, leading to misrepresentation of objects from novel classes. Hence, designing disentangled, *concept-based* representations, aligned with human cognitive mechanisms, is the key to open-world generalization, yet remains challenging.

Despite broad progress, *genuine* semantic generalization, *without* any access to novel categories during training, remains open (Du et al., 2023; Zheng et al., 2024). Prior work largely targets adjacent settings that assume exposure to the target semantics (Vaze et al., 2022; Wen et al., 2023; Li et al., 2023; Cao et al., 2022); empirically, these approaches often yield entangled, non–concept-aligned representations and lack theoretical guarantees (Li et al., 2022; Udandarao et al., 2024). A complementary line derives generalization bounds for semantic transferability (Sun et al., 2023; 2024), but the bounds hinge on access or alignment assumptions that do not hold in the genuine setting and therefore do not extend. This gap highlights the need for a principled framework that offers operational guidelines and actionable generalization guarantees.

Our key insight is to formalize the human generalization cognitive mechanism into a concept-aligned framework with a compare-then-compose process grounded in cognitive theories and studies. This framework draws on two fundamental cognitive principles: (i) comparison-driven concept extraction, where humans learn identifiable concepts through systematic comparison and alignment of relations across examples (Gentner, 1983; Markman & Gentner, 2000; Kong et al., 2022; Zheng et al.), with contrasting cases amplifying discriminative dimensions that distinguish categories (Goldstone, 1994; Kruschke, 1996); and (ii) compositional category construction, where complex concepts are built from primitive components by composing learned primitives (Fodor & Pylyshyn, 1988; Biederman, 1987), enabling rapid generalization from few examples (Lake et al., 2015; 2017). Crucially, comparison facilitates abstraction that can be reused compositionally in novel domains (Gentner & Namy, 1999; Gentner et al., 2007). This cognitive foundation motivates two fundamental research questions toward achieving genuine semantic generalization:

*(RQ1) Under what conditions can transferable concept representations be learned?*
*(RQ2) When can learned concepts be composed to identify unseen categories at test time?*

Guided by this cognitive perspective, we answer these two questions within our formalized framework, **C**omparison–**C**omposition **C**ognition ($\mathbf{C}^3$). *Specifically*, we establish identifiability guarantees for identifying composable latent concepts under the condition of sufficient cross-category contrast without parametric assumptions (RQ1). This ensures that the learned latent components serve as physically meaningful or *faithful* descriptors of category-specific semantics. With sufficient semantic diversity, all semantic concepts can be identified. *Subsequently*, we establish that reliable recognition of unseen categories is guaranteed when they emerge as compositions of identifiable concepts, under assumptions of disjoint concept supports and marginal coverage of reusable seen concepts (RQ2).

We instantiate our general theoretical framework in a genuine generalization setting, *i.e.*, *On-the-fly Category Discovery (OCD)*. *Specifically*, our method separates contextual factors (*e.g.*, background and surroundings) from semantics (*e.g.*, regional texture, color, or parts) while extracting high-level semantic concepts (Cao et al., 2025) composed of sparse primitives (Saban et al., 2021). It can serve as a plug-in module on a pretrained encoder, *e.g.*, DINO (Caron et al., 2021).

Our contributions are summarized as: (1) We propose a cognitive-inspired framework, $\mathbf{C}^3$, which formalizes human generalization mechanisms through a compare-then-compose process for learning concept-aligned representations and semantic generalization. (2) We establish theoretical identifiability guarantees, proving that composable latent concepts can be reliably identified under sufficient semantic contrast and that unseen categories can be recognized through concept compositions under mild assumptions. (3) We experimentally validate our approach on On-the-fly Category Discovery benchmarks, achieving +3.8% average accuracy improvement over state-of-the-art methods, demonstrating successful cognitive-to-computational translation.

## 2 PRELIMINARIES

**Data-generating Process.** We formulate the image generation as a process governed by latent concepts associated with labels, as illustrated on the *left* of Figure 2 and formalized in the equation

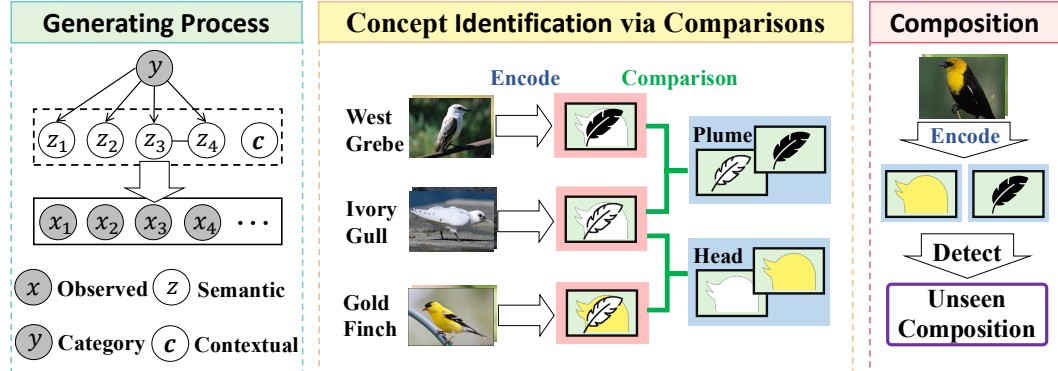

Figure 2: **The Pipeline of $C^3$ Framework.** *Left:* The generating process from latent concepts and categories to observations. *Middle:* Concept identification through comparison across labels, which excludes the invariant part and distinguishes the changing part. pink indicate the hidden concepts are unidentified or entangled, and blue denotes they have been identified or recovered. *Right:* Inference through previously identified concepts, and detect whether this composition is seen or unseen.

below:

$$\mathbf{x} = g(\mathbf{z}, \mathbf{c}), \quad \mathbf{z} = f(\mathbf{z}, y, \boldsymbol{\epsilon}), \quad \mathbf{c} \sim p_{\mathbf{c}}, \quad \boldsymbol{\epsilon} \sim p_{\boldsymbol{\epsilon}}. \tag{1}$$

We assume that the data $\mathbf{x} \in \mathbb{R}^{d_x}$ (*e.g.*, images or their frozen embeddings) are directly generated from these visual primitives, comprising *contextual concepts* $\mathbf{c} \in \mathbb{R}^{d_c}$ (*e.g.*, background, surroundings), which have category-invariant distributions, and *semantic concepts* $\mathbf{z} \in \mathbb{R}^{d_z}$ (*e.g.*, fur textures, plant colors, regional shapes), which are category-dependent and governed by a basis distribution $p_{\boldsymbol{\epsilon}}$ across labels $y$ through a function $f$. The motivation behind this formulation falls in a similarity between a reliable human cognition (Biederman, 1987) and a sparse causal mechanism (Schölkopf et al., 2021). Recognition hypothesis represents the concepts as *individual* components, which, coincidentally, align with the requirement that latent sources be *independent* for unique recovery from observed data (Hyvärinen & Pajunen, 1999). Notably, $\mathbf{z}$ is included inside $f(\cdot)$ to capture interactions among hidden concepts, accounting for how different arrangements of latent components give rise to latent representation (Ballard & Brown, 1982), where the dependencies are captured as a latent Markov network $\mathcal{M}$. In presence of it, we introduce a graphical condition for whether a latent concept can be disentangled in a Markov network $\mathcal{M}$:

**Definition 1** (Maximum Clique). *Consider a Markov network $\mathcal{M}$ over the latent variable set $\mathbf{z}$, the maximum clique of variable $z_i$ is defined as*

$$\Psi_{\mathcal{M}}(z_i) \coloneqq \{z_j \in \mathbf{z} \backslash \{z_i\} \mid z_j \in \mathbf{N}(z_i) \wedge z_j \in \mathbf{N}(z_k), \forall z_k \in \mathbf{N}(z_i)/\{z_i, z_j\}\}.$$

where $\mathbf{N}(\cdot)$ denotes the neighbor set. Thus, $|\Psi_{\mathcal{M}}(z_i)| = 0$ means $z_i$ is self-determined (or only unary factors), whereas $|\Psi_{\mathcal{M}}(z_i)| > 0$ means $z_i$ is coupled to others via multivariate cliques. For example, in Fig. 2, we have $\Psi_{\mathcal{M}}(z_1) = \Psi_{\mathcal{M}}(z_2) = \emptyset$, $\Psi_{\mathcal{M}}(z_3) = \{z_4\}$, and $\Psi_{\mathcal{M}}(z_4) = \{z_3\}$.

## 3 A THEORETICAL LADDER FOR $C^3$ FRAMEWORK

In this section, we mirror the human cognition process above to rigorous theoretical guarantees in machines. To determine under what conditions training on seen classes aids in classifying unseen classes, we frame the first step, *comparison*, as latent variable identification. In Theorem 1, we show that contextual concepts can be isolated by noting that their distributions are independent of the category $y$; in other words, an object's surrounding scene does not alter its category. Next, using the label $y$ as side information, we prove that data across categories can be compared contrastively to yield identifiable concepts. Finally, in Theorem 3, we establish the conditions under which unseen *compositions* of concepts generalize cognition to novel categories. For clarity, we now formally define our step-by-step targets.

**Criteria of Identification.** Let $\mathbf{X} = \{\mathbf{x}_i\}_{\mathcal{X}}$ be a set of observed images generated by the true model $(f, g, p(\epsilon))$ in Eq. 1. A learned model $(\hat{f}, \hat{g}, \hat{p}(\epsilon))$ is *observationally equivalent* to the true model if

$p_{\hat{f},\hat{g},\hat{p}(\epsilon)}(\hat{\mathbf{x}}) = p_{f,g,p(\epsilon)}(\mathbf{x})$. Under this equivalence, we define the following identifiability goals ($h$ is a bijective function with an arbitrary nonlinear form):

  i. Contextual Subspace: the estimated $\mathbf{c}$ does not contain any information in $\mathbf{z}$, *i.e.*, $\hat{\mathbf{c}} = h(\mathbf{c})$.

  ii. Semantic Component: the estimated $z_i$ contains information from connected $z_k, z_l$, $\hat{z}_i = h(z_{\pi(k)}, z_{\pi(l)})$, or from a component if isolated, $\hat{z}_i = h_i(z_{\pi(i)})$, where $\pi$ is a permutation.

  iii. Unseen Category: Let $\Phi_S = \{\phi(y) : y^s \in Y_S\}$ denote the set of concepts encountered in training (see) categories, where $\phi(y) = (z_1, \ldots, z_{d_z})$ is the tuple of latent concepts. For each concept $z_i$, let $\mathcal{S}_i(z_i) = \operatorname{supp} P(z_i \mid y)$ denote the support set of that concept value, and let $\mathcal{R}(z) = \bigtimes_{i=1}^{d_z} \mathcal{S}_i(z_i)$ be the product region corresponding to a full tuple. Then for any $y^q \in Y_Q$,

$$\phi(y^q) = (z_1^q, \ldots, z_{d_z}^q) \notin \Phi_S \iff z^q \in \mathcal{R}(z^q), \quad \mathcal{R}(z^q) \cap \mathcal{R}(z^s) = \emptyset, \; \forall z^s \in \Phi_S.$$

After that, we provide the theorem concerning the information recovery of the contextual concepts $\mathbf{c}$.

**Theorem 1.** *(**Recovery of Contextual Concepts**) We follow the data generation process in Eq. 1. Suppose a learned model has the observational equivalence. We make the following assumptions:*

*A1 (Well-Posed Density): The joint probability density function $p_{\mathbf{z},\mathbf{c}|y}$ is smooth and positive.*

*A2 (Contextual Comparison): For any $\mathbf{z} \in \mathcal{Z} \subseteq \mathbb{R}^{d_z}$, there exist $d_z + 1$ values of $y$, i.e., $y_j$ with $j = 0, 1, \cdots, d_z$, such that these $d_z$ vectors $\mathbf{v}(\mathbf{z}, y_j) - \mathbf{v}(\mathbf{z}, y_0)$ with $j = 1, \cdots, d_z$ are linearly independent, where the vector $\mathbf{v}(\mathbf{z}, y_j)$ is defined as: $\mathbf{v}(\mathbf{z}, y_j) = \left( \frac{\partial \log p(z_1 | y_j)}{\partial z_1}, \cdots \frac{\partial \log p(z_{d_z} | y_j)}{\partial z_{d_z}} \right)$.*

*Then the contextual concepts are able to be recovered as $\hat{\mathbf{c}} = h(\mathbf{c})$, where $h$ is an invertible function.*

**Remarks.** A1 is a mild assumption for the computability of the probability density functions, implicitly satisfied when they are meaningful and continuous (Feller, 1950). A2 requires that the semantic subspace $\mathbf{z}$ is diverse across categories, a precondition in multi-label datasets.

**Proof Sketch.** The full proof can be found in Appendix A2.3. Intuitively, with different labels $y$, the distribution of $\mathbf{c}$ remains static since it is not relevant to $y$, whereas $\mathbf{z}$ exhibits variability. To reconstruct the observations, the model captures such invariance-variance information, and then $\mathbf{z}$ and $\mathbf{c}$ are separated naturally. Notably, our conditions not rely on $\mathbf{z}$ being conditionally independent given $y$ (Kong et al., 2022), nor on parametric assumptions (Wiedemer et al., 2023).

> **Key Insights.** Once contextual concepts and semantic concepts are effectively disentangled, the estimated semantic concepts $\hat{\mathbf{z}}$ are supposed to preserve all the information $\mathbf{z}$ while excluding $\mathbf{c}$. This guarantees the robustness and fidelity of employing $\hat{\mathbf{z}}$ as an integrative variable for inter-category comparisons, e.g., cluster-based methodologies (Hartigan, 1975). Beyond the subspace, we provide a theorem below that each individual concept can be recovered under sufficient comparisons and a sparse mechanism.

**Theorem 2.** *(**Recovery of Semantic Concepts**) Consider the data generation process specified in Eq. 1, and assume the conditions in Theorem 1 are satisfied. Let the learned model be observationally equivalent to the true model. We impose the following additional assumptions:*

- *A3 (Semantic Comparison): For any $\mathbf{z} \in \mathbb{R}^{d_z}$, there exist $2d_z + |\mathcal{M}| + 1$ distinct label values $y$, i.e., $y = 1, \ldots, 2d_z + |\mathcal{M}|$, such that the vectors $\boldsymbol{w}(\mathbf{z}, y) - \boldsymbol{w}(\mathbf{z}, 0)$ are linearly independent, where the vector $\boldsymbol{w}(\mathbf{z}, y)$ is defined as*

$$\boldsymbol{w}(\mathbf{z}, y) = \left( \frac{\partial \log p(\mathbf{z}|y)}{\partial z_1}, \ldots, \frac{\partial \log p(\mathbf{z}|y)}{\partial z_{d_z}}, \frac{\partial^2 \log p(\mathbf{z}|y)}{\partial z_1^2}, \ldots, \frac{\partial^2 \log p(\mathbf{z}|y)}{\partial z_{d_z}^2} \right) \oplus \left( \frac{\partial^2 \log p(\mathbf{z}|y)}{\partial z_i \partial z_j} \right)_{(i,j) \in \mathcal{M}}.$$

- *A4 (Sparse Arrangement): The sparsity of the estimated latent Markov network does not exceed that of the true one, i.e., $|\hat{\mathcal{M}}| \leq |\mathcal{M}|$.*

*Under these conditions, the following identifiability results hold ($h$ is a arbitrary bijective function):*

i. *(Semantic Component Recovery): For any $z_i$ such that $|\Psi_{\mathcal{M}}(z_i)| = 0$, we have $\hat{z}_i = h(z_{\pi(i)})$.*

ii. *(Semantic Module Recovery): For any $z_i$ such that $|\Psi_{\mathcal{M}}(z_i)| \neq 0$, we have that $z_i$ admits modular identifiability, i.e., $\hat{z}_i = h(z_{\pi(i)} \cup \Psi_{\mathcal{M}}(z_{\pi(i)}))$.*

**Remarks.** A3 requires that each semantic concept $\mathbf{z}$ varies sufficiently with $y$, also satisfied by multi-labels but need more than A2. It is consistent with the principle underlying classification tasks in computer vision, where class-discriminative features are emphasized (Krizhevsky et al., 2012; Bellet et al., 2013). A4 reflects a standard prior in disentangled representation learning (Peebles et al., 2020), assuming that most concepts are conditionally independent, with only a few forming modular dependencies. Moreover, it can be operationalized as a regularization on the flow model.

**Key Insights.** The detailed proof is provided in Appendix A2.4. The key idea is to build on the subspace identifiability results established in Theorem 1. Let $h'_{i,l} := \frac{\partial z_i}{\partial \hat{z}_l}$ and $h''_{i,kl} := \frac{\partial^2 z_i}{\partial \hat{z}_k \partial \hat{z}_l}$. Under A3 and A4, it can be shown that the following constraints are satisfied:

$$h'_{i,l} h'_{i,k} = 0, \quad h'_{j,l} h'_{i,k} = 0, \quad h''_{i,kl} = 0. \tag{2}$$

These conditions imply that $z_i$ depends on at most one of $\hat{z}_k$ and $\hat{z}_l$. Furthermore, if $z_i$ and $z_j$ are adjacent in the Markov network $\mathcal{M}$, then due to the sparsity constraint, at most one of them can be a function of either $\hat{z}_k$ or $\hat{z}_l$. In other words, this structural restriction enables component recovery. Combining Theorems 1 and 2, we guarantee that each concept is reliably learned through *comparison*. Consequently, we equip them with the following theorem to construct reliable category descriptors.

**Theorem 3.** *(Compositional Concept Generalization) Let the learned model be observationally equivalent to the true model specified in Eq. 1. We impose the following additional assumptions:*

- *A5 (Concept Isolation): For any $z_i^{(k)}, z_i^{(l)} \in \mathcal{Z}_i$, $k \neq l \implies \mathcal{S}_i(z_i^{(k)}) \cap \mathcal{S}_i(z_i^{(l)}) = \varnothing$.*

- *A6 (Coverage in Training Set): Given a learned model $(\hat{f}, \hat{g}, \hat{p}(\epsilon))$ on seen categories $Y_S$,*

$$\forall \hat{z}_i \in \hat{\mathcal{Z}}_i : \quad \Pr_{y \in \mathcal{Y}_S, \hat{\mathbf{z}} \in \mathcal{Z}, \hat{\epsilon} \sim \hat{p}_\epsilon} \left[ \hat{f}_i(y, \hat{\mathbf{z}}, \hat{\epsilon}) = \hat{z}_i \right] > 0.$$

*Then for any unseen tuple $z^q = (z_1^q, \ldots, z_{d_z}^q)$, the region $\mathcal{R}(z^q) = \bigtimes_{i=1}^{d_z} \mathcal{S}_i(z_i^q)$ is disjoint from all other tuples in the seen categories.*

**Discussions.** These conditions are not restrictive but are naturally satisfied in practice. A5 specifies that different descriptions of a concept, such as "red" and "yellow" for color, correspond to disjoint support sets. A6 requires that the concepts employed in unseen categories are still contained within the original functional space of the data-generating process. In other words, under the learned model $(\hat{f}, \hat{g}, \hat{p}(\epsilon))$, an image from an unseen category can still be reconstructed using the concepts obtained from the training set, reminiscent of generative models.

**Key Insights.** The key property used in the proof is factorization, where, conditional on the label $y$, the latent distribution factorizes as $p(\mathbf{z} \mid y) = \prod_{i=1}^{d_z} p_i(z_i \mid y, \Psi_{\mathcal{M}}(z_i))$, with each $z_i$ depending only on $y$ and its maximum clique. If supports of different concepts are disjoint, factorization ensures the joint space is a *product space* of coordinate-wise supports, and marginal coverage guarantees learnability of each building block. Together, these conditions imply that unseen categories can be systematically identified as novel compositions of known concepts.

## 4 METHODOLOGY

In this section, we describe how our theoretical framework is instantiated to address the practical semantic generalization problem: genuinely generalizing to novel visual semantics using only training data from known semantics (Du et al., 2023). Given sufficient data from known categories, we train a

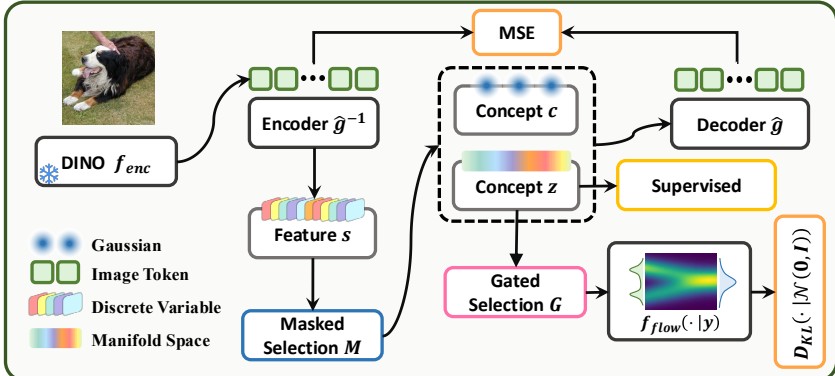

Figure 3: **Architectural overview** of our instantiated $\mathbf{C}^3$ theoretical framework for OCD problem. The latent autoencoder representation $s$ can be partitioned into semantic concept $\mathbf{z}$ and contextual concept $\mathbf{c}$ (dotted box), assuming sufficient semantic comparison. During training, we enforce Markov network structural priors on sparsely activated semantic neurons together with sparsity constraints and diversified prototypical learning. Taken together, these components guarantee the identifiability of the concepts.

discriminative–generative model (Fig. 3) with the bottleneck representation decomposed into two latent subspaces (Sec. 4.1): (i) semantic (semantic-related features) and (ii) contextual (semantics-irrelevant, *e.g.*, background and low-level features) (Ganin et al., 2016; Arjovsky et al., 2019). We enforce identifiability of these concepts under a Markov network using sparsity, structural, and diversity regularizers (Sec. 4.2). At test time, the classifier extrapolates to unseen categories by composing the learned semantic concepts while marginalizing nuisance context (Sec. 4.3). This instantiation follows the $\mathbf{C}^3$ framework (Fig. 2), aligning learning with comparison to uncover latent concepts across seen classes and composition to extend them to the unseen.

**Problem Formulation.** We operationalize our framework in the setting of *On-the-fly Category Discovery* (OCD) (Du et al., 2023; Zheng et al., 2024), a problem that demands genuine semantic generalization. The full data $\mathcal{D}$ consists of a support set $\mathcal{D}_S$ for training and a query set $\mathcal{D}_Q$ for testing. Given $N$ training samples $\mathcal{D}_S = \{(\mathbf{x}_i, y_i)\}_{i=1}^N \in \mathcal{X}_S \times \mathcal{Y}_S$ from a seen category space $\mathcal{Y}_S$, our objective is to categorize $M$ query samples $\mathcal{D}_Q = \{(\mathbf{x}_i, y_i)\}_{i=1}^M \in \mathcal{X}_Q \times \mathcal{Y}_Q$ drawn from a superset semantic space $\mathcal{Y}_Q$ where $\mathcal{Y}_S \subseteq \mathcal{Y}_Q$. Critically, following the OCD protocol: (i) only the support set $\mathcal{D}_S$ is used for model training (*i.e.*, inductive learning), and (ii) during inference, query instances arrive in a stream and must be categorized individually in real-time (*i.e.*, instant feedback), mirroring the operating demands of human cognitive ability in open-ended environments.

## 4.1 ENCODING LATENT REPRESENTATION

Widely corroborated by cognitive literature, humans learn to identify and distinguish different categories with sparse key visual features while taking semantic-irrelevant information as background context (Olshausen & Field, 1996; Biederman, 1987; Tversky, 1977; Kruschke, 2020; Nosofsky, 1986). Drawing from this insight, we aim to first identify the two disentangled latent concept subspaces (Sec. 3) grounded on the conditions in Theorem 1.

Concretely, given any input image $\mathbf{o}$, we partition the bottleneck representation upstreaming the decoder and classifier we extract the features from a frozen $f_{\text{enc}}(\cdot)$ as the input to the autoencoder we first employ a frozen encoder $f_{\text{enc}}(\cdot)$, instantiated by DINO, to obtain a set of frozen embeddings. The *[CLS] token* is selected as the observation $\mathbf{x}$, which is then transformed through an inverse mapping $\hat{g}^{-1}$ into an intermediate, unstructured representation $\mathbf{s}$:

$$\mathbf{x} = f_{\text{enc}}(\mathbf{o}), \quad \mathbf{s} = \hat{g}^{-1}(\mathbf{x}), \quad [\hat{\mathbf{z}}, \hat{\mathbf{c}}] = M(\mathbf{s}). \tag{3}$$

Since this latent representation $\mathbf{s}$ contains both semantic and contextual information, we introduce a learnable selection module $M(\cdot)$ to disentangle it into semantic concepts $\hat{\mathbf{z}}$ and contextual concepts $\hat{\mathbf{c}}$. The module $M(\cdot)$ is implemented as a 1-layer MLP equipped with a learnable binary mask.

Specifically, the selection for the $i$-th element $\mathbf{s}_i$ is modeled as a Bernoulli random variable: $m_i \sim \text{Ber}(\sigma(\gamma_i))$, where $\gamma_i$ is optimized using the Gumbel-Softmax reparameterization (Jang et al., 2016). If $m_i = 1$, the component $\mathbf{s}_i$ is assigned to $\hat{\mathbf{z}}$; otherwise to the contextual set $\hat{\mathbf{c}}$.

## 4.2 LATENT CONCEPT IDENTIFICATION

**Contextual Concepts.** To ensure that $\hat{\mathbf{c}}$ retains only contextual information, which is category-invariant, we impose an independence constraint by aligning its distribution to a standard Gaussian:

$$\mathcal{L}_{\text{ctx}} = D_{\text{KL}}\left(p(\hat{\mathbf{c}}) \mid \mathcal{N}(\mathbf{0}, \mathbf{I})\right). \tag{4}$$

This regularization encourages $\hat{\mathbf{c}}$ to discard category-related variations while preserving only contextual information, since its distribution keeps static given different $y$.

**Semantic Sparsification.** Inspired by findings in cognitive science (Cao et al., 2025) that neurons responsible for recognition exhibit both sparse and distributed activations, we promote sparsity in the semantic representation $\hat{\mathbf{z}}$. This is implemented by (i) introducing $\ell_1$ regularization and (ii) significantly enlarging the latent dimensionality ($d_z = 3072$). The sparsity constraint is defined as:

$$\mathcal{L}_{\text{s}} = \sum_{i=1}^{n_c} \|[\hat{\mathbf{c}}]_i\|_1, \tag{5}$$

which improves disentanglement and facilitates the invertibility of the decoding function $g$, allowing the model to select flexible regions in latent space.

**Structuralize Semantic Concepts.** After learning a semantic subspace with with sparsification, in pursuit of uncovering the relational structure among semantic components, we impose a sparsity-aware structure learning mechanism. We employ a sparsely-gated Mixture-of-Experts (MoE) (Shazeer et al., 2017) to learn the adjacency matrix $M_{\mathbf{z}}$, corresponding the latent Markov network $\mathcal{M}$. A gating function computes a selection mask based on the semantic representation:

$$\mathbf{G} = \text{Softmax}(W_g\hat{\mathbf{z}} + b_g), \quad M_{\mathbf{z}} = \text{TopKMask}(\mathbf{G}, k), \tag{6}$$

where $W_g$ and $b_g$ are learnable, and $\text{TopKMask}(\cdot, k)$ retains only the $k$ highest dimensions. The masked representation is then passed through a normalizing flow to model the conditional generative process:

$$\hat{\epsilon} = f_{\text{flow}}(M_{\mathbf{z}} \odot \hat{\mathbf{z}}, y). \tag{7}$$

And then, we compute the flow loss using normalizing flow (Rezende & Mohamed, 2015) as follows:

$$\mathcal{L}_{\text{flow}} = -\mathbb{E}\left[\log p_s(f_{\text{flow}}(\hat{\epsilon})) + \log\left|\det\left(\frac{\partial f_{\text{flow}}}{\partial \hat{\epsilon}}\right)\right|\right]. \tag{8}$$

**Concept Diversification.** The essence of comparison lies in capturing differences across labels, which requires diversity in the learned concepts. To satisfy conditions A2 and A3, the model must acquire representations that vary across categories while remaining coherent within each class. To this end, we train the semantic representation $\hat{\mathbf{z}}$ with supervised prototype-based learning (Zheng et al., 2024; Chen et al., 2019), using the label $y$ to enforce discriminative subspaces. This objective, denoted $\mathcal{L}_{\text{proto}}$, encourages cross-category diversity and intra-class consistency. Implementation details are given in Appendix A3.1. The intuition is that the family of supervised learning encourages recovering sufficiently diverse concepts required for identifiability (Reizinger et al., 2024), and we can show that prototype learning, which models $\mathbf{z}$ along with a sparse mechanism, can further impose the component-wise concept diversity to learn the semantically meaningful concepts. Here, let $U = [U_{jy}] \in \mathbb{R}^{m \times |\mathcal{Y}|}$ denotes the coefficient matrix that linearly maps the prototype similarities to the class logits, where each element $U_{jy}$ specifies the contribution of the $j$-th prototype $\boldsymbol{p}_j$ to the $y$-th class. In particular, the $y$-th column $U_{\cdot y}$ represents the prototype composition of class $y$, and the contrast between classes is characterized by $\beta_y := U_{\cdot y} - U_{\cdot 0}$.

**Proposition 1** (Prototype Learning Ensures Concept Diversity). *Under Eq. 1 and the conditions of Theorem 1, consider a prototype layer $g_p$ with $m$ learnable prototypes $\mathbf{P} = \{\boldsymbol{p}_j\}_{j=1}^m$ and similarity*

$$s_j(\mathbf{z}) = \log\frac{\|\mathbf{z} - \boldsymbol{p}_j\|_2^2 + 1}{\|\mathbf{z} - \boldsymbol{p}_j\|_2^2 + \epsilon}, \quad 0 < \epsilon < 1.$$

*If (1) $m \geq 2d_z + |\mathcal{M}|$ and $\{p_j\}$ are in general position, and (2) there exist $K := 2d_z + |\mathcal{M}|$ labels whose coefficient differences $\beta_y := U_{.y} - U_{.0}$ are linearly independent, then after convergence of $L_{proto}$ (i.e., $p_\theta(y \mid \mathbf{z}) = p(y \mid \mathbf{z})$), Assumption A3 (Semantic Comparison) holds. Moreover,*

$$\sigma_{\min}\big(W(\mathbf{z})\big) \;\geq\; \sigma_{\min}\big(G(\mathbf{z})\big)\,\sigma_{\min}(B) \;>\; 0,$$

*where $W(\mathbf{z}) = [\,\boldsymbol{w}(\mathbf{z}, y_i) - \boldsymbol{w}(\mathbf{z}, 0)\,]_{i=1}^{K}$, $G(\mathbf{z}) \in \mathbb{R}^{(2d_z + |\mathcal{M}|) \times m}$, and $B = [\,\beta_{y_1} \;\cdots\; \beta_{y_K}\,]$.*

Here, we prove that the prototype layer explicitly enforces that each category acquires distinct anchor points $\{p_j\}$, ensuring that the gradients and Hessians of $\log p(\mathbf{z} \mid y)$ vary across labels. Consequently, the learned representation satisfies the diversity and linear-independence conditions required by Assumption A2 / A3, thereby providing a concrete architectural mechanism that guarantees sufficient semantic comparison for identifiability.

## 4.3 Unseen Category Generalization

As stated in Theorem 2, if the maximum clique of an estimated concept $\hat{z}_i$ is nonzero, only a modular (entangled) concept can be identified. To make such modules effective category descriptors, we introduce a residual integration mechanism:

$$\hat{\mathbf{a}} = \hat{\mathbf{z}} + f_{\text{residual}}\big(M_{\mathbf{z}} \odot \hat{\mathbf{z}}\big), \tag{9}$$

where $f_{\text{residual}}$ is a lightweight MLP that aggregates information from connected components in $M_{\mathbf{z}}$. This ensures the learned representation serves as a reliable descriptor for downstream classification.

After obtaining meaningful latent representations, we use the refined components $\hat{\mathbf{a}}$ for downstream category discovery. For composition, since clear support separation across concepts is required by A5, we approximate it with discrete hashing techniques (Du et al., 2023; Zheng et al., 2024), which enforce separation even when only a single latent unit differs. Binary encodings further enhance component-wise separation (Hoe et al., 2021; Yuan et al., 2020; Wang et al., 2023). The supervised objective includes prototype and hashing losses: $\mathcal{L}_{\text{sup}} = \mathcal{L}_{\text{proto}} + \mathcal{L}_{\text{hash}}$. Further theoretical details are in Appendix A3.2.

## 4.4 Learning Objective

**Variational Objective.** We use a variational objective to maximize the estimated likelihood:

$$\mathcal{L}_{\text{ELBO}} = \mathbb{E}_{q(\hat{\mathbf{z}}, \hat{\mathbf{c}} \mid \mathbf{x})}\Big[\log p(\mathbf{x} \mid \hat{\mathbf{z}}, \hat{\mathbf{c}})\Big] - D_{\text{KL}}\Big(q(\hat{\mathbf{z}} \mid \mathbf{x}) | p(\hat{\mathbf{z}})\Big) - D_{\text{KL}}\Big(q(\hat{\mathbf{c}} \mid \mathbf{x}) | \mathcal{N}(\mathbf{0}, \mathbf{I})\Big), \tag{10}$$

which captures the fidelity of the reconstruction as well as prior alignment for latent variables.

**Overall Objective.** Our overall training loss is formulated as:

$$\mathcal{L}_{\text{ALL}} = \mathcal{L}_{\text{sup}} + \mathcal{L}_{\text{ELBO}} + \lambda_1 \mathcal{L}_{\text{flow}} + \lambda_2 \mathcal{L}_{\text{s}} + \lambda_3 \mathcal{L}_{\text{ctx}}, \tag{11}$$

where $\lambda_{\{1,2,3\}}$ are empirically determined parameters (see Appendix A4.3) that balances the intensity of the structure learning, sparsity regularization, and invariance on the contextual concepts.

## 5 Experimental Results

**Datasets.** Our experiments were performed on eight fine-grained datasets: CUB-200 (Wah et al., 2011), Stanford Cars (Krause et al., 2013), Oxford-IIIT Pet (Parkhi et al., 2012), Food-101 (Bossard et al., 2014), and four super-categories from the iNaturalist dataset (Van Horn et al., 2018)—Fungi, Arachnida, Animalia, and Mollusca. These super-categories were selected due to their increased complexity. Following the methodology outlined in OCD (Du et al., 2023; Zheng et al., 2024), the categories within each dataset were divided into seen and unseen subsets. For training, 50% of the samples from the seen categories were allocated to the labeled set $\mathcal{D}_S$, while the remaining samples were assigned to the unlabeled set $\mathcal{D}_Q$, which was used for the stage of on-the-fly testing. Additional details regarding the datasets can be found in the Appendix A4.1.

Table 1: Comparison with SOTA methods. The best / second-best results are **bolded** / underlined.

| Method | CUB (%) | | | Stanford Cars (%) | | | Oxford Pets (%) | | | Food101 (%) | | | Average (%) | | |
|---|---|---|---|---|---|---|---|---|---|---|---|---|---|---|---|
| | All | Old | New | All | Old | New | All | Old | New | All | Old | New | All | Old | New |
| SLC | 31.3 | 48.5 | 22.7 | 24.0 | 45.8 | 13.6 | 35.5 | 41.3 | 33.1 | 20.9 | 48.6 | 6.8 | 27.9 | 46.1 | 19.1 |
| RankStat | 27.6 | 46.2 | 18.3 | 18.6 | 36.9 | 9.7 | 33.2 | 42.3 | 28.4 | 22.3 | 50.7 | 7.8 | 25.4 | 44.0 | 16.1 |
| WTA | 26.5 | 20.0 | 38.8 | 10.6 | 24.4 | 13.6 | 35.2 | 46.3 | 29.3 | 18.2 | 40.5 | 6.1 | 25.0 | 42.7 | 15.8 |
| SMILE | 32.2 | 50.8 | 22.9 | 26.1 | 46.6 | 16.2 | 41.2 | 42.1 | 40.7 | 24.0 | 54.6 | 8.4 | 30.9 | 48.6 | 22.1 |
| PHE | 36.4 | 55.8 | 27.0 | 31.3 | 61.9 | 16.8 | 48.3 | 53.8 | 45.4 | 29.1 | 64.7 | 11.1 | 36.3 | 59.1 | 25.1 |
| C³ | **40.1** | **62.1** | **29.5** | **34.1** | **69.0** | **17.8** | **54.7** | **63.9** | **49.6** | **31.5** | **68.3** | **11.9** | **40.1** | **65.8** | **27.2** |

| Method | Fungi (%) | | | Arachnida (%) | | | Animalia (%) | | | Mollusca (%) | | | Average (%) | | |
|---|---|---|---|---|---|---|---|---|---|---|---|---|---|---|---|
| | All | Old | New | All | Old | New | All | Old | New | All | Old | New | All | Old | New |
| SLC | 27.7 | 60.0 | 13.4 | 25.4 | 44.6 | 11.4 | 32.4 | **61.9** | 19.3 | 31.1 | 59.8 | 15.0 | 29.2 | 56.6 | 14.8 |
| RankStat | 23.8 | 50.5 | 12.0 | 26.6 | 51.0 | 10.0 | 31.4 | 54.9 | 21.6 | 29.3 | 52.9 | 14.8 | 27.8 | 52.9 | 14.8 |
| WTA | 27.5 | 65.6 | 12.0 | 28.1 | 55.5 | 10.9 | 33.4 | 59.8 | 22.4 | 30.3 | 55.4 | 17.0 | 29.8 | 59.1 | 15.6 |
| SMILE | 29.3 | 64.6 | 13.6 | 29.9 | 57.9 | 12.2 | 35.9 | 49.4 | 30.3 | 33.3 | 44.5 | 27.2 | 32.1 | 54.1 | 20.8 |
| PHE | 31.4 | 67.9 | 15.2 | 37.0 | **75.7** | 12.6 | 40.3 | 55.7 | 31.8 | 39.9 | 65.0 | 26.5 | 37.2 | 66.1 | 21.5 |
| C³ | **32.9** | **69.8** | **16.2** | **38.1** | 72.0 | **13.1** | **42.0** | 60.1 | **32.2** | **41.9** | **70.2** | **27.3** | **38.7** | **68.1** | **22.2** |

**Evaluation Metrics.** We follow Du et al. (2023); Zheng et al. (2024) and adopt clustering accuracy as an evaluation protocol. The accuracy calculation via *Strict-Hungarian* algorithm can be formulated as $\text{ACC} = \frac{1}{|\mathcal{D}_Q|} \sum_{i=1}^{|\mathcal{D}_Q|} \mathbb{I}(y_i = C(\overline{y}_i))$, where $\overline{y}_i$ represents the predicted labels and $y_i$ denotes the ground truth. The function $C$ denotes the optimal permutation that aligns predicted cluster assignments with the actual class labels.

**Implementation Details.** To ensure a fair and consistent comparison, we follow the standard evaluation setup established in the OCD literature (Du et al., 2023; Zheng et al., 2024). All models are built upon the DINO-pretrained ViT-B/16 backbone (Caron et al., 2021), with all transformer layers frozen except for the final block. For discriminative component learning, we adopt a prototype-based strategy in which each category is associated with 10 prototypes. During training, positive prototype pairs (from the same category) are assigned a weight of 1, while negative pairs (from different categories) are assigned a weight of $-0.5$ to encourage inter-class separation. While prior OCD baselines adopt a fixed category encoding dimensionality of 12, our method differs by incorporating a selective module that automatically identifies it. More implementation details are in Appendix A4.3.

**Baseline Methods.** Since OCD is a relatively new task that demands real-time inference, conventional baselines from NCD and GCD are not suitable for this setting. Therefore, we conduct comparisons including SLC (Hartigan, 1975), RankStat (Han et al., 2021), WTA (Jia et al., 2021), SMILE (Du et al., 2023), and PHE (Zhang et al., 2024b). The first set of strong baselines follows the configuration of SMILE (Du et al., 2023), with implementation details in Appendix A4.3.

**Comparison with SOTAs.** We conduct comparative experiments against the aforementioned competitors across all six datasets. The results are presented in Table 1. Compared to the state-of-the-art model PHE (Zheng et al., 2024), the proposed method achieves a notable average improvement of 3.8%, a large progress in this task, demonstrating the effectiveness of identification-guided principles. Additionally, our method outperforms PHE by 6.7% on the average performance of old classes across four datasets. Furthermore, on the more challenging iNaturalist dataset, our method achieves the best results on most evaluation metrics, with a 1.5% improvement in the overall average score.

**Concept Visualization.** We acknowledge that estimated concept perfect alignment with human semantic understanding may not always be achieved, since the true concepts are *inherently unobserved*. Hence, we attempt to verify this indirectly. We introduce a new concept-dimension visualization pipeline, where each feature-space dimension is treated as an individual concept axis. The activation value along each dimension represents the response strength of that concept and is projected back onto the spatial grid to form a heatmap. This additional analysis reveals how latents encode semantically meaningful cues, thereby offering finer-grained interpretability of the learned concepts.

**Ablations on Identifiability Guarantees.** To demonstrate the necessity of our methodology in practice, we perturb the conditions imposed in our model and examine the consequences of losing identifiability. Specifically, we remove the compositional structure of the data-generating process from our model. As shown in Table 2, the accuracy on both CUB and SCARS deteriorates significantly

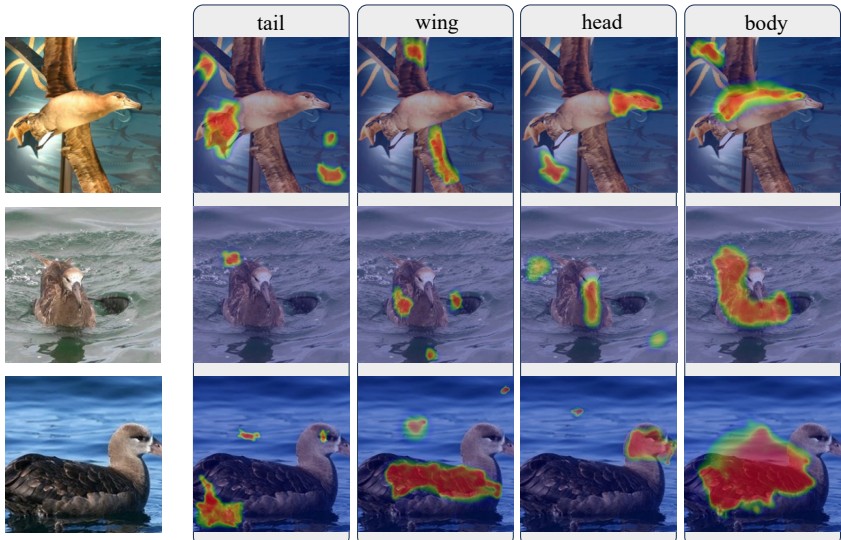

Figure 4: Concept Visualization on the Black-footed Albatross. Each column represents the image area activated by the learned concept, and each row corresponds to an image sample. We provide textual descriptions for each concept based on the most plausible semantic interpretation.

Table 2: **Ablations on Losses for Identifiability Guarantees.** The best results are marked in **bold**.

| $\mathcal{L}_{\text{flow}}$ | $\mathcal{L}_{\text{s}}$ | $\mathcal{L}_{\text{ELBO}}$ | CUB (%) | | | SCars (%) | | |
|:---:|:---:|:---:|:---:|:---:|:---:|:---:|:---:|:---:|
| | | | All | Old | New | All | Old | New |
| ✓ | ✓ | | 38.4 | 62.8 | 26.2 | 30.2 | 57.4 | 17.2 |
| ✓ | | ✓ | 39.3 | 62.4 | 27.7 | 31.1 | 59.0 | 17.6 |
| | ✓ | ✓ | 38.5 | 61.3 | 27.0 | 30.2 | 59.6 | 16.0 |
| ✓ | ✓ | ✓ | **40.1** | **62.1** | **29.5** | **34.1** | **69.0** | **17.8** |

after this perturbation, indicating that disentangled concepts are essential for extrapolating to novel categories.

# 6 CONCLUSION AND DISCUSSIONS

We proposed a theoretical framework that models unseen category cognition as *concept identification via comparison* and *re-composition*, with guarantees of identifiability. Building on this foundation, we introduced a novel algorithm, which separates contextual from semantic factors and composes sparse, interpretable primitives. Experiments on multiple OCD benchmarks show that our method achieves state-of-the-art performance, validating the effectiveness of theory-driven, concept-based representations for open-world unseen category cognition. Although our framework is theoretically grounded and empirically validated on top of DINO representation, we have not yet evaluated its scalability to large-scale settings or integration with foundational models. Extending our method to foundation-scale architectures remains an important avenue for future work.

## ETHICS STATEMENT

This work relies solely on publicly available benchmark datasets and does not involve human subjects or sensitive information. The methodology is intended for scientific research, and we are not aware of harmful applications.

## REPRODUCIBILITY STATEMENT

Details of simulated data generation are provided in Appendix A4.2, dataset descriptions in Appendix A4.1, and implementation settings for real-world experiments in Appendix A4.3. Extended results and ablations (*e.g.*, Tables A3, A4) are also reported, and we will release code and configurations for full replication.

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

*Appendix for*

**"From Comparison to Composition:**
**Towards Understanding Machine Cognition of Unseen Categories"**
Table of Contents:

## A1 RELATED WORK

### A1.1 CATEGORY DISCOVERY

*Novel Category Discovery (NCD)* problem, originally formulated in DTC (Han et al., 2019), aims to cluster unlabeled data sampled from novel categories by leveraging knowledge from labeled known categories. *Generalized Category Discovery (GCD)*, as proposed in (Vaze et al., 2022; Cao et al., 2022), extends NCD by assuming that unlabeled data may encompass both known and novel categories instead of solely novel ones. Despite promising progress in NCD/GCD (Han et al., 2019; Li et al., 2023; Sun et al., 2023; Wen et al., 2023; Zhang et al., 2023; Choi et al., 2024; Rastegar et al., 2024), their underlying assumptions are unrealistic in real-world scenarios. *On-the-fly Category Discovery (OCD)*, as proposed by (Du et al., 2023), recalibrates these assumptions in two key aspects: (1) the exclusion of unlabeled data from novel categories during training, as such data is often inaccessible in practice; and (2) the streaming arrival of unseen categories during inference, necessitating the model to provide instant feedback. Therefore, compared to NCD and GCD, OCD offers a more realistic setting as it better aligns with the target challenge of known-to-novel semantic generalization that we seek to tackle. However, this practical yet challenging problem remains largely unexplored.

The central challenge of OCD, compared to NCD/GCD, lies in learning a universal semantic representation purely from known categories that can effectively generalize to unseen visual semantics during inference. Previous NCD/GCD study explores myriad approaches to enhance transferable semantic representation from known to novel categories (Vaze et al., 2022; Wen et al., 2023; Zhang et al., 2023), primarily through cross-view consistency (Fini et al., 2021; Zhao & Han, 2021; Han et al., 2020), graph-based positive mining (Zhang et al., 2023; Huynh et al., 2022; Sun & Li, 2022; Hao et al., 2023), decoupled pertaining (Vaze et al., 2024), information regularization (Li et al., 2023; Chiaroni et al., 2023; Tan et al., 2024; Rastegar et al., 2024), knowledge distillation (Gu et al., 2023; Lin et al., 2024; Wang et al., 2024), *etc.* Nevertheless, since all the aforementioned methods heavily depend on auxiliary unlabeled data containing target categories, they are thus inapplicable to the OCD problem. Two prior works have tackled the coding problem in OCD: SMILE (Du et al., 2023) proposes hash code-based prototypes as classifiers to cluster novel categories; meanwhile, PHE (Zheng et al., 2024) further enhances the semantic discriminativeness of discrete codes. In contrast to these methods, we introduce a principled theoretical framework that not only provides concept-level insights into the underlying semantic generalization process but also serves as a guideline for enhancing technical design.

### A1.2 MECHANISMS ON CATEGORY DISCOVERY

Several studies have explored the theoretical aspects of semantic transfer from different perspectives. For instance, (Li et al., 2022) examines the countereffects of supervised knowledge in NCD by modeling the transfer from known to novel categories using category-wise maximum mean discrepancy (MMD) metrics (Smola et al., 2006). Besides, (Sun et al., 2023) investigates knowledge transfer by deriving a generalization error bound for NCD, based on representations learned through a semi-supervised contrastive loss. Further, (Sun et al., 2024) investigates the knowledge transferability of the representation learned with a spectral graph-based contrastive loss for GCD. Meanwhile, other studies investigate the desirable properties of generalizable representation for NCD/GCD from the empirical perspective. For example, the recent study (Vaze et al., 2024) investigates the enhanced representation transferability benefiting from physically-grounded concepts (*e.g.*, color, shape, and material).

The aforementioned theoretical results are limited by their reliance on unlabeled novel categories, and thus inapplicable to the open-world challenge of OCD. By contrast, our method introduces a theoretical framework for semantic generalization for general OCD problem, modeling latent concepts based on human cognitive principles, and enabling the recognition of novel visual patterns by transferring of foundational primitives and disentangled concepts. While some prior works have attempted to learn latent visual concepts to improve representation transferability, these concepts often lack identifiability or are not physically grounded (Han et al., 2019; Zhang et al., 2024a; Rastegar et al., 2024).

### A1.3 VISUAL CONCEPT LEARNING

Visual concept learning has been a significant long-standing interdisciplinary problem in machine learning and cognitive science (Saban et al., 2021; Voges et al., 2024), which has broad applications in retrieval (Kiros et al., 2014), image captioning (Karpathy & Fei-Fei, 2015), scene understanding (Wu

et al., 2017), explainable image classification (Xiong et al., 2021; Jia et al., 2013), image generation and editing (Kumari et al., 2023a; Gandikota et al., 2024; Kumari et al., 2023b), *etc.* Related recent work on Concept-Bottleneck Models (CBMs) explores the use of human-annotated textual concepts to enhance explainable representation learning in the intermediate (bottleneck) latent space, complementing standard supervised loss functions (Koh et al., 2020; Shin et al., 2023; Yuksekgonul et al., 2022). However, these methods come with several limitations. *First*, they rely heavily on expert-annotated concepts, which not only incur exorbitant annotation costs but also introduce subjective biases, restricting their applicability to diverse or rapidly evolving domains. *Second*, the capacity of human-annotated textual attributes is inherently limited – they struggle to capture abstract visual patterns beyond human intuitions (Margeloiu et al., 2021). *Third*, these approaches inherently lack adaptability, as they cannot generalize to novel categories without additional expert guidance. In this work, we propose a hierarchical concept learning framework that autonomously learns theoretically identifiable semantic and contextual concepts. This identifiability ensures the *reliability* and *faithfulness* of transferred semantic knowledge when generalizing to novel categories without the reliance on expert annotations.

Meanwhile, we notice that Tang et al. (2025a) similarly decomposes objects into primitive representations and employs multiplex consensus between dominant and contextual components to improve performance. However, their approach assumes access to novel categories during training in the GCD context, and thus does not achieve the genuine generalization we target. Additionally, Tang et al. (2025b) utilizes relational concept-based graphs to enhance out-of-domain generalization and relational reasoning robustness in foundation models. None of these works establishes the theoretical formalization, generalization conditions, or identifiability guarantees necessary to characterize semantic transferability.

### A1.4 LATENT VARIABLE IDENTIFICATION

Latent Variable Identification (LVI) aims to recover the underlying latent variables that govern observed data, particularly when direct measurements are incomplete or obscured. This problem is fundamental across various disciplines, *e.g.*, weather analysis (Fu et al., 2025) and video processing (Yao et al., 2022), given only incomplete observations. A key challenge in LVI is that latent variables often remain unidentifiable due to nontrivial transformations (Hyvärinen & Pajunen, 1999), even when independent factors of variation are known (Locatello et al., 2019). To overcome this, existing methods introduce structural constraints, exploit statistical dependencies, or incorporate auxiliary information (Zhang et al., 2024b) to infer latent causal structures.

Several studies have further explored LVI in hierarchical settings, aiming to uncover multi-level latent structures. For instance, (Huang et al., 2022) employed rank constraints to identify hierarchical latent representations, while (Xie et al., 2020) utilized the Generalized Independent Noise (GIN) condition to estimate latent causal graphs in linear, non-Gaussian settings. Building on these foundations, (Kong et al., 2023) investigated the conditions necessary for identifying hierarchical latent variable models in continuous settings, while (Kong et al., 2024) extended this analysis to the discrete case within the framework of latent diffusion models. However, these approaches primarily assume linear relationships, which limits their applicability in complex visual domains where hierarchical structures exhibit nonlinear dependencies.

On the empirical side, prior work has sought to enhance inference models in deep generative frameworks. For example, (Sønderby et al., 2016) improved the inference model of vanilla Variational Autoencoders (VAEs) by integrating bottom-up data-dependent likelihood terms with prior generative distribution parameters. (Lachapelle et al., 2022) further highlighted the necessity of sparsity mechanisms for interpreting vision concepts. In parallel, (Kivva et al., 2021) analyzed the identifiability of latent representations and causal models, focusing on dependencies between high-level features rather than raw observations. In the context of computer vision, hierarchical identification techniques have been employed to extract interpretable features, enabling models to organize high-level semantic concepts from visual data. (Kwon et al., 2022) and (Park et al., 2023) observed that the UNet bottleneck representation exhibits highly structured semantic properties, such that traversing its latent space can manipulate generated images in a meaningful manner.

### A1.5 COMPOSITIONAL GENERALIZATION

Prior work has approached compositional generalization from empirical, theoretical, and structural perspectives. Du et al. (2020) combined energy-based models for Cartesian-product extrapolation but assumed datasets where only one latent factor varies. Besserve et al. (2021) developed a theoretical view where out-of-distribution samples result from transformations within decoder layers, and

Krueger et al. (2021) introduced a domain generalization method robust beyond the convex hull of training tasks. More recent work exploits stronger inductive biases: Lachapelle et al. (2023) uses additive decoders to learn latent factors, Brady et al. (2024) formalizes concept composition via *interaction asymmetry*, and Zheng et al. identifies the latent factors through the diversity on the sparse connectivity structure. Distinct from these approaches, we leverage the sufficient diversity in data distributions to disentangle latent concepts without any parametric assumptions and transfer them to unseen categories, only relying on disjoint support sets.

## A2 PROOFS

To better understand our proof, we first present some useful definitions regarding the graphical model.

### A2.1 MARKOV NETWORK

A Markov network (or Markov random field) is a graphical model that represents the joint distribution of a set of random variables using an undirected graph.

**Definition 2** (Markov Network). *Markov network is an undirected graph $G = (V, E)$ with a set of random variables $X_{v \in V}$, where any two non-adjacent variables are conditionally independent given all other variables. That is,*

$$X_a \perp X_b | X_{V \setminus \{a,b\}}, \quad \forall (a,b) \notin E. \tag{12}$$

Markov Networks and Directed Acyclic Graphs (DAGs) are both graphical models employed to represent joint distributions and to illustrate conditional independence properties.

### A2.2 ISOMORPHISM OF MARKOV NETWORKS

**Definition 3** (Isomorphism of Markov networks). *We let the $V(\cdot)$ be the vertical set of any graphs, an isomorphism of Markov networks $M$ and $\hat{M}$ is a bijection between the vertex sets of $M$ and $\hat{M}$*

$$f : V(M) \to V(\hat{M})$$

*such that any two vertices $u$ and $v$ of $M$ are adjacent in $G$ if and only if $f(u)$ and $f(v)$ are adjacent in $\hat{M}$.*

### A2.3 PROOF OF THEOREM 1

*Proof.* We begin with the matched marginal distribution $p_{\mathbf{x}|y}$ to bridge the relation between $\mathbf{z}$ and $\hat{\mathbf{z}}$. For brevity, we use $y$ to represent the labels of known categories, $y^s \in \mathcal{Y}_s$. Suppose that $\hat{g} : \mathcal{Z} \times \mathcal{C} \to \mathcal{X}$ is an invertible estimated generating function, we have Eq. 13.

$$\forall y \in \mathcal{Y}_s, \quad p_{\hat{\mathbf{x}}|y} = p_{\mathbf{x}|y} \iff p_{\hat{g}(\hat{\mathbf{z}},\hat{\mathbf{c}})|y} = p_{g(\mathbf{z},\mathbf{c})|y}. \tag{13}$$

Sequentially, by using the change of variables formula, we can further obtain Eq. 14

$$p_{\hat{g}(\hat{\mathbf{z}},\hat{\mathbf{c}}|y)} = p_{g(\mathbf{z},\mathbf{c}|y)} \iff p_{g^{-1} \circ g(\hat{\mathbf{z}},\hat{\mathbf{c}})|y} |\mathbf{J}_{g^{-1}}| = p_{\mathbf{z},\mathbf{c}|y} |\mathbf{J}_{g^{-1}}| \iff p_{h(\hat{\mathbf{z}},\hat{\mathbf{c}})|y} = p_{\mathbf{z},\mathbf{c}|y}, \tag{14}$$

where $h := g^{-1} \circ g$ is the transformation between the ground-true and the estimated latent variables, respectively. $\mathbf{J}_{g^{-1}}$ denotes the absolute value of Jacobian matrix determinant of $g^{-1}$. Since we assume that $g$ and $\hat{g}$ are invertible, $|\mathbf{J}_{g^{-1}}| \neq 0$ and $h$ is also invertible.

According to A2 (conditional independent assumption), we can have Eq. 15.

$$p_{\mathbf{z}|y}(\mathbf{z}|y) = p_{\mathbf{c}|y}(\mathbf{c}|y) \cdot p_{\mathbf{z}|y}(\mathbf{z}|y); \quad p_{\hat{\mathbf{z}}|y}(\hat{\mathbf{z}}|y) = p_{\hat{\mathbf{z}}|y}(\hat{\mathbf{z}}|y) \cdot p_{\hat{\mathbf{c}}|y}(\hat{\mathbf{c}}|y). \tag{15}$$

For convenience, we take the logarithm on both sides of Eq. 15 and further let $q_s = \log p_{\mathbf{z}|y}(\mathbf{z}|y), q_c = \log p_{\mathbf{c}|y}(\mathbf{c}|y), p_s = \log p_{\hat{\mathbf{z}}|y}(\hat{\mathbf{z}}|y), p_c = \log p_{\hat{\mathbf{c}}|y}(\hat{\mathbf{c}}|y)$. Hence we have:

$$\log p_{\mathbf{z}|y}(\mathbf{z}|y) = q_s + q_c; \quad \log p_{\hat{\mathbf{z}}|y}(\hat{\mathbf{z}}|y) = p_s + p_c. \tag{16}$$

By combining Eq. 16 and Eq. 14, we have:

$$p_{\mathbf{z}|y} = p_{h(\hat{\mathbf{z}}|y)} \iff p_{\hat{\mathbf{z}}|y} = p_{\mathbf{z}|y} |\mathbf{J}_{h^{-1}}| \iff q_s + q_c + \log |\mathbf{J}_{h^{-1}}| = p_s + p_c, \tag{17}$$

where $\mathbf{J}_{h^{-1}}$ are the Jacobian matrix of $h^{-1}$.

Sequentially, we take the first-order derivative with $\hat{z}_j$ on Eq. (17), where $\hat{z}_j$ is from $\mathbf{c}$, and have

$$\sum_{z_i \in \mathbf{z}} \frac{\partial q_s}{\partial z_i} \cdot \frac{\partial z_i}{\partial \hat{z}_j} + \sum_{z_i \in \mathbf{c}} \frac{\partial q_c}{\partial z_i} \cdot \frac{\partial z_i}{\partial \hat{z}_j} + \frac{\partial \log |\mathbf{J}_{h^{-1}}|}{\partial \hat{z}_j} = \frac{\partial p_s}{\partial \hat{z}_j} + \frac{\partial p_c}{\partial \hat{z}_j}. \tag{18}$$

Suppose $y = y_0, y_1, \cdots, y_{n_z}$, we subtract the Eq. 18 corresponding to $y_k$ with that corresponds to $y_0$, and we have:

$$\sum_{z_i \in \mathbf{z}} \left( \frac{\partial q_s(y_k)}{\partial z_i} - \frac{\partial q_s(y_0)}{\partial z_i} \right) \cdot \frac{\partial z_i}{\partial \hat{z}_j} + \sum_{z_i \in \mathbf{c}} \left( \frac{\partial q_c(y_k)}{\partial z_i} - \frac{\partial q_c(y_0)}{\partial z_i} \right) \cdot \frac{\partial z_i}{\partial \hat{z}_j}$$
$$= \frac{\partial \hat{q}_s(y_k)}{\partial \hat{z}_j} - \frac{\partial \hat{q}_s(y_0)}{\partial \hat{z}_j} + \frac{\partial \hat{q}_c(y_k)}{\partial \hat{z}_j} - \frac{\partial \hat{q}_c(y_0)}{\partial \hat{z}_j}. \tag{19}$$

Since the distribution of estimated $\hat{z}_j$ does not change across different categories, $\frac{\partial \hat{q}_s(y_k)}{\partial \hat{z}_j} - \frac{\partial \hat{q}_s(y_0)}{\partial \hat{z}_j} = 0$. Since $\frac{\partial q_s(y_k)}{\partial z_i}$ does not change across different categories, $\frac{\partial q_c(y_k)}{\partial z_i} = \frac{\partial q_c(y_0)}{\partial z_i}$, $\frac{\partial q_s(y_k)}{\partial z_i} = \frac{\partial q_s(y_0)}{\partial z_i}$ for $z_i \in \mathcal{Z}_s$. So we have

$$\sum_{i \in \mathbf{c}} \left( \frac{\partial q_s(y_k)}{\partial z_i} - \frac{\partial q_s(y_0)}{\partial z_i} \right) \cdot \frac{\partial z_i}{\partial \hat{z}_j} = 0. \tag{20}$$

Based on the linear independence assumption (A3), the linear system is a $n_z \times n_z$ full-rank system. Therefore, the only solution is $\frac{\partial z_i}{\partial \hat{z}_j} = 0$.

Since $h(\cdot)$ is smooth over $\mathcal{Z}$, its Jacobian can be formalized as follows

$$\boldsymbol{J}_h = \left[ \begin{array}{c|c} \mathbf{A} := \frac{\partial \mathbf{z}}{\partial \hat{\mathbf{z}}} & \mathbf{B} := \frac{\partial \mathbf{z}}{\partial \hat{\mathbf{c}}} \\ \hline \mathbf{C} := \frac{\partial \mathbf{c}}{\partial \hat{\mathbf{z}}} & \mathbf{D} := \frac{\partial \mathbf{c}}{\partial \hat{\mathbf{c}}}. \end{array} \right] \tag{21}$$

Note that $\frac{\partial z_i}{\partial \hat{z}_j} = 0$ for $z_i \in \mathcal{Z}$ and $z_j \in \mathcal{Z}$ means that $\mathbf{B} = 0$. Since $h(\cdot)$ is invertible, $\boldsymbol{J}_h$ is a full-rank matrix. Therefore, for each $\mathbf{z}$, there exists a $h_i$ such that $\mathbf{z} = h_i(\hat{\mathbf{z}})$. $\qquad\square$

### A2.4 PROOF OF THEOREM 2

We begin by presenting a useful lemma from (Zhang et al., 2024b), which connects group-wise transformations to component-wise transformations in a Markov network. This lemma is instrumental for the subsequent proof, in particular, it enables us to first recover the latent variables within groups of adjacent nodes in the Markov network.

**Lemma 1** (Identifiability of Hidden Causal Variables). *If $z_i$ is a function of at most one of $\hat{z}_k$ and $\hat{z}_l$, and given that $z_i$ and $z_j$ are adjacent in Markov network $\mathcal{M}_{\mathbf{z}}$, at most one of them is a function of $\hat{z}_k$ or $\hat{z}_l$. Then, there exists a permutation $\pi$ of the estimated hidden variables, denoted as $\hat{z}_\pi$, such that each $\hat{z}_{\pi(i)}$ is a function of (a subset of) the variables in $\{\mathbf{z}_i\} \cup \Psi_{\mathbf{z}_i}$.*

*Proof. Step 0 (Setup and change of variables).* By Theorem 1, there exists an invertible, dimension-preserving $h$ such that

$$h(\hat{\mathbf{z}}) = \mathbf{z} \implies p_{h(\hat{\mathbf{z}})} = p_{\mathbf{z}}.$$

Let $J_h$ be the Jacobian of $h$ and $J_{h^{-1}}$ that of $h^{-1}$. By the change-of-variables formula,

$$p(\hat{\mathbf{z}} \mid \hat{y}^s) \, |\det J_{h^{-1}}(\hat{\mathbf{z}})| \; = \; p(\mathbf{z} \mid y) \quad \implies \quad \log p(\hat{\mathbf{z}} \mid \hat{y}^s) \; = \; \log p(\mathbf{z} \mid y) + \log |\det J_h(\mathbf{z})|. \tag{22}$$

Suppose $\hat{z}_k \perp\!\!\!\perp \hat{z}_l \mid \hat{z}_{[n] \setminus \{k,l\}}$ (i.e., $k, l$ are non-adjacent in the Markov network over $\hat{\mathbf{z}}$). Then for each $\hat{y}^s$, by (Lin, 1997),

$$\frac{\partial^2}{\partial \hat{z}_k \, \partial \hat{z}_l} \log p(\hat{\mathbf{z}} \mid \hat{y}^s) \; = \; 0. \tag{23}$$

Differentiate Eq. 22 w.r.t. $\hat{z}_k$:

$$\frac{\partial}{\partial \hat{z}_k} \log p(\hat{\mathbf{z}} \mid \hat{y}^s) = \sum_{i=1}^n \frac{\partial \log p(\mathbf{z} \mid y)}{\partial z_i} \frac{\partial z_i}{\partial \hat{z}_k} + \frac{\partial}{\partial \hat{z}_k} \log |\det J_h(\mathbf{z})|.$$

Introduce the shorthand

$$\eta(y) := \log p(\mathbf{z} \mid y), \quad \eta_i'(y) := \frac{\partial \log p(\mathbf{z} \mid y)}{\partial z_i}, \quad \eta_{ij}''(y) := \frac{\partial^2 \log p(\mathbf{z} \mid y)}{\partial z_i \partial z_j}, \quad h_{i,l}' := \frac{\partial z_i}{\partial \hat{z}_l}, \quad h_{i,kl}'' := \frac{\partial^2 z_i}{\partial \hat{z}_k \partial \hat{z}_l}.$$

Differentiating again w.r.t. $\hat{z}_l$ and using Eq. 23 yields

$$0 = \sum_{j=1}^{n}\sum_{i=1}^{n} \eta_{ij}''(y)\, h_{j,l}'\, h_{i,k}' \;+\; \sum_{i=1}^{n} \eta_i'(y)\, h_{i,kl}'' \;+\; \frac{\partial^2}{\partial \hat{z}_k \partial \hat{z}_l} \log|\det J_h(\mathbf{z})|$$

$$= \sum_{i=1}^{n} \eta_{ii}''(y)\, h_{i,l}'\, h_{i,k}' \;+\; \sum_{j=1}^{n}\sum_{i:\{z_j,z_i\}\in\mathcal{E}(\mathcal{M}_{\mathbf{z}})} \eta_{ij}''(y)\, h_{j,l}'\, h_{i,k}' \;+\; \sum_{i=1}^{n} \eta_i'(y)\, h_{i,kl}'' \;+\; \frac{\partial^2}{\partial \hat{z}_k \partial \hat{z}_l} \log|\det J_h(\mathbf{z})|.$$

$$(24)$$

Here $\mathcal{E}(\mathcal{M}_{\mathbf{z}})$ denotes the edges of the Markov network over $\mathbf{z}$.

By Assumption 2, pick $2d_z + |\mathcal{M}_{\mathbf{z}}| + 1$ values $y^{(u)}$, $u = 0,\ldots,2d_z + |\mathcal{M}_{\mathbf{z}}|$, so that Eq. 24 holds. Subtract the $u = 0$ instance from each $u \geq 1$, the Jacobian term cancels, yielding constraints below.

From the linear independence condition (Assumption 2), we deduce that for any edge $\{i,j\} \in \mathcal{E}(\mathcal{M}_{\mathbf{z}})$,

$$h_{i,k}' h_{i,l}' = 0, \qquad h_{i,k}' h_{j,l}' + h_{j,k}' h_{i,l}' = 0, \qquad h_{i,kl}'' = 0.$$

The constraints imply that each $z_i$ can depend on at most one of $\hat{z}_k$, $\hat{z}_l$. By contradiction: if $h_{i,k}' h_{j,l}' \neq 0$, then $h_{i,l}' = 0$ by the first constraint, which makes the second constraint force $h_{i,k}' h_{j,l}' = 0$, contradiction. Thus at most one of an adjacent pair $(z_i, z_j)$ depends on a given recovered coordinate. Hence, isolated $z_i$ yield component-wise identifiability ($\hat{z}_{\pi(i)} = h_i(z_i)$), while nodes in a clique can only be recovered modularly. Sparsity ensures most concepts are identifiable as individual components.

By Lemma 1, there exists a permutation $\pi$ such that each $\hat{z}_{\pi(i)}$ is a function of $\{z_i\} \cup \Psi_{z_i}$, where $\Psi_{z_i}$ are the neighbors of $z_i$ in $\mathcal{M}_{\mathbf{z}}$. In sparse cases this reduces to invertible component-wise identifiability.

□

**Illustrative Examples of Assumptions**   This assumption characterizes the discriminative component of the model. The condition about the linear independence implies that there exists a unique characteristic of the concept that cannot be linearly represented by other variables. To clarify this assumption, we provide two examples (Yao et al., 2022) to demonstrate scenarios where the assumption holds and where it does not. Let $\eta_k = \frac{\partial q_k(d_k, y)}{\partial d_k}$.

*Example 1*: Violation of the Assumption (Additive Gaussian Noise) Consider a case where the assumption is violated due to the presence of additive Gaussian noise. Let $y$ denote the label, and let $d_k = q_k(y) + \epsilon_k$, where $\epsilon_k \sim N(0,1)$. In this scenario, we have: $\eta_k = -\log\sqrt{2\pi} - \frac{(d_k - q_k(y))^2}{2}$, and $\frac{\partial^2 \log P(d_k|y)}{\partial^2 d_k} = 0$. This result violates the assumption because the second derivative of the log-likelihood with respect to $d_k$ is zero, indicating a lack of discriminative power in the latent variables.

*Example 2*: Validation of the Assumption (Generalized Normal Distribution) Conversely, consider a case where the assumption holds. Let $\epsilon_k$ follow a zero-mean generalized normal distribution: $P(\epsilon_k) \propto e^{-\lambda|\epsilon_k|^\beta}$, where $\lambda > 0$, $\beta > 2$, and $\beta \neq 3$. Let $d_k = q_k(y) + \epsilon_k$, where $q$ is a linear function. If, for each $d_k$, there exists at least one $l$ such that $c_{kl} = \frac{\partial d_k}{\partial y_l} \neq 0$, the assumption must hold.

In this case, we derive the following:

$$\frac{\partial^3 \eta_k}{\partial^2 d_k \partial y_l} = -\lambda \operatorname{sgn}(\epsilon_k)\,\beta(\beta-1)(\beta-2)|\epsilon_k|^{\beta-3} c_{kl},$$

and

$$\frac{\partial^2 \eta_k}{\partial d_k \partial y_l} = -\lambda\beta(\beta-1)|\epsilon_k|^{\beta-2} c_{kl}.$$

Here, $|\epsilon_k|^{\beta-2}$ and $|\epsilon_k|^{\beta-3}$ are linearly independent because their ratio, $|\epsilon_k|$, is not constant. Furthermore, the functions $|\epsilon_{lt}|^{\beta-2}$ and $|\epsilon_{lt}|^{\beta-3}$, for $l = 1, 2, \ldots, n$, are $2n$ linearly independent functions due to their distinct arguments.

Suppose there exist coefficients $\alpha_{l1}$ and $\alpha_{l2}$ for $l = 1, 2, \ldots, n$ such that the weighted sum with respect to $\mathbf{w}_{l,t}$ is zero:

$$\alpha_{k1} c_{kl}|\epsilon_k|^{\beta-2} + \alpha_{k2} c_{kl}|\epsilon_k|^{\beta-3} + \sum_{l \neq k}\left(\alpha_{l1} c_{ll}|\epsilon_{lt}|^{\beta-2} + \alpha_{l2} c_{ll}|\epsilon_{lt}|^{\beta-3}\right) = 0.$$

Since $|\epsilon_k|^{\beta-2}$ and $|\epsilon_k|^{\beta-3}$ are linearly independent and $c_{kl} \neq 0$, the only way for the above Eq. to hold is if $\alpha_{k1} = \alpha_{k2} = 0$ for all $k$. This implies that $\alpha_{l1}$ and $\alpha_{l2}$ must be zero for all $l = 1, 2, \ldots, n$. Consequently, the set $\{\mathbf{w}_{lt}\}$ is linearly independent, confirming that the assumption holds in this case.

### A2.5 PROOF OF THEOREM 3

*Proof.* By Theorem 1, there exist a permutation $\pi$ of $\{1, \ldots, d_z\}$ and invertible, dimension-preserving maps $h_i : \mathbb{R}^{d_i} \to \mathbb{R}^{d_i}$ such that

$$\hat{z}_{\pi(i)} \;=\; h_i(z_i), \qquad i = 1, \ldots, d_z. \tag{25}$$

For a fixed coordinate $i$ and a value $z_i$, let $\mathcal{S}_i(z_i) \subset \mathbb{R}^{d_i}$ denote the support set from Assumption A5. Define the pushed-forward supports in the recovered coordinates by

$$\widehat{\mathcal{S}}_{\pi(i)}(z_i) \;:=\; h_i\big(\mathcal{S}_i(z_i)\big) \;\subset\; \mathbb{R}^{d_i}. \tag{26}$$

For a full concept tuple $z = (z_1, \ldots, z_{d_z})$, define the product (rectangle) regions

$$\mathcal{R}(z) \;:=\; \prod_{i=1}^{d_z} \mathcal{S}_i(z_i), \qquad \widehat{\mathcal{R}}(z) \;:=\; \prod_{i=1}^{d_z} \widehat{\mathcal{S}}_{\pi(i)}(z_i). \tag{27}$$

By Assumption A5, for any two distinct values $u \neq u'$ of the $i$-th concept, $\mathcal{S}_i(u) \cap \mathcal{S}_i(u') = \emptyset$. Since $h_i$ in Eq. 25 is bijective, it preserves set disjointness:

$$\widehat{\mathcal{S}}_{\pi(i)}(u) \cap \widehat{\mathcal{S}}_{\pi(i)}(u') = h_i(\mathcal{S}_i(u)) \cap h_i(\mathcal{S}_i(u')) = h_i\big(\mathcal{S}_i(u) \cap \mathcal{S}_i(u')\big) = \emptyset.$$

Hence, for each coordinate $i$, the family $\{\widehat{\mathcal{S}}_{\pi(i)}(u)\}_u$ is pairwise disjoint.

For each $i$, define a decoder $\psi_{\pi(i)} : \mathbb{R}^{d_i} \to$ (value set of $z_i$) by membership in the disjoint sets:

$$\psi_{\pi(i)}(\hat{z}_{\pi(i)}) \;=\; u \quad \Longleftrightarrow \quad \hat{z}_{\pi(i)} \in \widehat{\mathcal{S}}_{\pi(i)}(u). \tag{28}$$

The right-hand side determines $u$ uniquely by Step 1, so $\psi_{\pi(i)}$ is well-defined (ties can only occur on set boundaries, which have probability zero under standard absolute continuity assumptions). Moreover, Eq. 25 and the definition in Eq. 26 give, for any realization from the true model,

$$\hat{z}_{\pi(i)} \;=\; h_i(z_i) \in h_i\big(\mathcal{S}_i(z_i)\big) \;=\; \widehat{\mathcal{S}}_{\pi(i)}(z_i),$$

and therefore $\psi_{\pi(i)}(\hat{z}_{\pi(i)}) = z_i$ almost surely.

Define $\psi : \mathbb{R}^{d_z} \to$ (value set of $z$) by

$$\psi(\hat{\mathbf{z}}) \;:=\; \big(\psi_{\pi(1)}(\hat{z}_{\pi(1)}), \ldots, \psi_{\pi(d_z)}(\hat{z}_{\pi(d_z)})\big).$$

Applying Step 2 coordinate-wise gives $\psi(\hat{\mathbf{z}}) = z$ almost surely for any sample generated by the true model and mapped by Eq. 25. Thus the tuple $z$ is a function of $\hat{\mathbf{z}}$ via support membership.

Let $z \neq z'$ be two tuples. Then there exists some index $i$ such that $z_i \neq z'_i$. By Assumption A5, $\mathcal{S}_i(z_i) \cap \mathcal{S}_i(z'_i) = \emptyset$. Consequently,

$$\mathcal{R}(z) \cap \mathcal{R}(z') \;=\; \Big( \prod_{j \neq i} \mathcal{S}_j(z_j) \cap \mathcal{S}_j(z'_j) \Big) \times \big( \mathcal{S}_i(z_i) \cap \mathcal{S}_i(z'_i) \big) \;=\; \emptyset,$$

where $\mathcal{R}$ is defined in Eq. 27. Hence the rectangles $\{\mathcal{R}(z)\}$ are pairwise disjoint. The same argument on the pushed-forward sets shows $\{\widehat{\mathcal{R}}(z)\}$ are pairwise disjoint.

Fix an unseen tuple $z^q = (z_1^q, \ldots, z_{d_z}^q)$ and suppose a test latent $z^\sharp$ lies in the rectangle $\mathcal{R}(z^q)$, i.e., $z_i^\sharp \in \mathcal{S}_i(z_i^q)$ for all $i$. Then $\hat{z}_{\pi(i)}^\sharp = h_i(z_i^\sharp) \in h_i(\mathcal{S}_i(z_i^q)) = \widehat{\mathcal{S}}_{\pi(i)}(z_i^q)$, so by Eq. 28 we obtain $\psi_{\pi(i)}(\hat{z}_{\pi(i)}^\sharp) = z_i^q$ for each $i$, and therefore $\psi(\hat{\mathbf{z}}^\sharp) = z^q$. By Step 4, $\widehat{\mathcal{R}}(z^q)$ is disjoint from $\widehat{\mathcal{R}}(z')$ for any $z' \neq z^q$, which implies that the decoding to $z^q$ is unique on $\widehat{\mathcal{R}}(z^q)$.

Assumption A6 states that for the learned model on seen categories, each coordinate value that may occur at test time is realized by at least one training label through the learned generator $\hat{f}$. Operationally, this ensures that every transformed support $\widehat{\mathcal{S}}_{\pi(i)}(\cdot)$ appearing in Step 2 is estimable from training data in the recovered space, so that the membership-based decoders $\{\psi_{\pi(i)}\}$ can be implemented (*e.g.*, by empirical support estimation or consistent plug-in rules). This bridges the population-level identifiability shown in Steps 1–5 with a practical decoding rule learned on seen categories. □

**Proposition 2** (Prototype learning $\Rightarrow$ A3 with a Spectral Bound). *Assume Eq. 1, the conditions of Theorem 1, and observational equivalence. Let the prototype layer $g_p$ contain $m$ learnable prototype vectors*

$$\mathbf{P} = \{\boldsymbol{p}_j\}_{j=1}^m, \qquad \boldsymbol{p}_j \in \mathbb{R}^{d_z},$$

*and define the prototype similarity as*

$$s_j(\mathbf{z}) = \log \frac{\|\mathbf{z} - \boldsymbol{p}_j\|_2^2 + 1}{\|\mathbf{z} - \boldsymbol{p}_j\|_2^2 + \epsilon}, \qquad 0 < \epsilon < 1.$$

*Let the class potentials be linear in similarities:*

$$\phi_y(\mathbf{z}) = b_y + \sum_{j=1}^m U_{jy}\, s_j(\mathbf{z}), \qquad p_\theta(y \mid \mathbf{z}) = \frac{e^{\phi_y(\mathbf{z})}}{\sum_{y'} e^{\phi_{y'}(\mathbf{z})}}.$$

*Suppose the supervised prototype loss $L_{proto}$ converges so that $p_\theta(y \mid \mathbf{z}) = p(y \mid \mathbf{z})$ (Fisher consistency). If (1) $m \geq 2d_z + |\mathcal{M}|$ and the prototypes $\{\boldsymbol{p}_j\}$ are in general position, and (2) there exist $K := 2d_z + |\mathcal{M}|$ labels whose coefficient differences $\beta_y := U_{\cdot y} - U_{\cdot 0} \in \mathbb{R}^m$ are linearly independent, then for any $\mathbf{z}$, Assumption A3 (Semantic Comparison) holds. Moreover, letting*

$$W(\mathbf{z}) := \big[\, \boldsymbol{w}(\mathbf{z}, y_1) - \boldsymbol{w}(\mathbf{z}, 0) \ \cdots \ \boldsymbol{w}(\mathbf{z}, y_K) - \boldsymbol{w}(\mathbf{z}, 0) \,\big],$$

*where*

$$\boldsymbol{w}(\mathbf{z}, y) = \left( \frac{\partial \log p(\mathbf{z}|y)}{\partial z_1}, \ldots, \frac{\partial \log p(\mathbf{z}|y)}{\partial z_{d_z}}, \frac{\partial^2 \log p(\mathbf{z}|y)}{\partial z_1^2}, \ldots, \frac{\partial^2 \log p(\mathbf{z}|y)}{\partial z_{d_z}^2} \right) \oplus \left( \frac{\partial^2 \log p(\mathbf{z}|y)}{\partial z_i \partial z_j} \right)_{(i,j) \in \mathcal{M}},$$

*we have*

$$\sigma_{\min}\big(W(\mathbf{z})\big) \ \geq \ \sigma_{\min}\big(G(\mathbf{z})\big)\, \sigma_{\min}(B) \ > \ 0,$$

*where $G(\mathbf{z}) \in \mathbb{R}^{(2d_z + |\mathcal{M}|) \times m}$ and $B := [\, \beta_{y_1} \ \cdots \ \beta_{y_K} \,] \in \mathbb{R}^{m \times K}$.*

*Proof.* By Bayes' rule, $\log p(\mathbf{z} \mid y) = \log p(y \mid \mathbf{z}) + \log p(\mathbf{z}) - \log p(y)$. Differentiating and subtracting the base class $y = 0$ removes the $y$-independent term:

$$\nabla\big(\log p(\mathbf{z} \mid y) - \log p(\mathbf{z} \mid 0)\big) = \nabla\big(\log p(y \mid \mathbf{z}) - \log p(0 \mid \mathbf{z})\big).$$

Since $\log p(y \mid \mathbf{z}) = \phi_y(\mathbf{z}) - \log \sum_{y'} e^{\phi_{y'}(\mathbf{z})}$, the partition term cancels in differences, so

$$\nabla\big(\log p(\mathbf{z} \mid y) - \log p(\mathbf{z} \mid 0)\big) = \sum_{j=1}^m \beta_{y,j}\, \nabla s_j(\mathbf{z}), \tag{29}$$

$$\nabla^2\big(\log p(\mathbf{z} \mid y) - \log p(\mathbf{z} \mid 0)\big) = \sum_{j=1}^m \beta_{y,j}\, \nabla^2 s_j(\mathbf{z}). \tag{30}$$

with $\beta_y = U_{\cdot y} - U_{\cdot 0}$. For $r_j^2 = \|\mathbf{z} - \boldsymbol{p}_j\|_2^2$,

$$\nabla s_j(\mathbf{z}) = 2s_j'(r_j^2)\, (\mathbf{z} - \boldsymbol{p}_j), \qquad \nabla^2 s_j(\mathbf{z}) = 2s_j'(r_j^2) I + 4s_j''(r_j^2)\, (\mathbf{z} - \boldsymbol{p}_j)(\mathbf{z} - \boldsymbol{p}_j)^\top,$$

and $s_j'(t) = (\epsilon - 1)/((t+1)(t+\epsilon)) \neq 0$. Stack the entries required by A3 in

$$g_j(\mathbf{z}) := \left( \nabla s_j(\mathbf{z}),\ \mathrm{diag}\big(\nabla^2 s_j(\mathbf{z})\big),\ (\nabla^2 s_j(\mathbf{z}))_{(i,k) \in \mathcal{M}} \right)^\top \in \mathbb{R}^{2d_z + |\mathcal{M}|},$$

and define $G(\mathbf{z}) := [\, g_1(\mathbf{z}) \ \cdots \ g_m(\mathbf{z}) \,]$. Then for each $y$,

$$\boldsymbol{w}(\mathbf{z}, y) - \boldsymbol{w}(\mathbf{z}, 0) = G(\mathbf{z})\, \beta_y. \tag{31}$$

Stacking Eq. 31 across the $K$ labels yields $W(\mathbf{z}) = G(\mathbf{z})B$.

Because $\{\boldsymbol{p}_j\}$ are in general position and $m \geq 2d_z + |\mathcal{M}|$, some $(2d_z + |\mathcal{M}|) \times (2d_z + |\mathcal{M}|)$ minor of $G(\mathbf{z})$ is nonzero (its analytic expression is not identically zero), hence $\mathrm{rank}[\, G(\mathbf{z})\,] = 2d_z + |\mathcal{M}|$. The discriminative prototype training induces class-wise exclusivity so $B$ has full column rank $K$. Therefore $\mathrm{rank}[\, W(\mathbf{z})\,] = 2d_z + |\mathcal{M}|$, implying the vectors $\{\boldsymbol{w}(\mathbf{z}, y_i) - \boldsymbol{w}(\mathbf{z}, 0)\}_{i=1}^K$ are linearly independent and A3 holds. Finally,

$$\sigma_{\min}\big(W(\mathbf{z})\big) \ \geq \ \sigma_{\min}\big(G(\mathbf{z})\big)\, \sigma_{\min}(B) \ > \ 0$$

by sub-multiplicativity of singular values, meaning that we can leverage prototype learning to sufficiently learn the distinctive concepts. $\square$

**Remarks.** $\sigma_{\min}(G(\mathbf{z}))$ captures geometric diversity of prototypes, while $\sigma_{\min}(B)$ reflects class diversity induced by the prototype loss. Their product lower-bounds the degree of semantic comparison, linking prototype learning convergence to A3.

## A3 METHODOLOGY DETAILS

### A3.1 ENCOURAGE DISCRIMINATIVE SUBSPACE

Our approach integrates a prototype layer, $g_p$, which transforms $\hat{\mathbf{z}}$ into a similarity score vector $\mathbf{s} \in \mathbb{R}^m$. This layer consists of $m$ learnable prototype vectors, denoted as $\{\mathbf{P}_1, \mathbf{P}_2, \ldots, \mathbf{P}_m\}$. For instance, (Chen et al., 2019) typically establishes the correspondence between the feature map and the prototype by extracting the maximum pooled value from the similarity map. Following a similar rationale, to capture the block-wise or subspace distance between the learned $\hat{\mathbf{z}}$ across different $y$, we draw inspiration from ProtoPFormer (Xue et al., 2022) and its specialized adaptation for OCD (Zheng et al., 2024) to compute the prototype similarity score. The similarity score $s_{ij}$ between the $i$-th sample and the $j$-th prototype is defined as:

$$s_{i \to j} = g_{p_j}(\hat{\mathbf{z}}) = \log\left(\frac{\|\hat{\mathbf{z}} - \mathbf{p}_j\|_2^2 + 1}{\|\hat{\mathbf{z}} - \mathbf{p}_j\|_2^2 + \epsilon}\right), \tag{32}$$

where $\hat{\mathbf{z}}$ represents the subspace associated with the low-level changing concept corresponding to $y_i$, and $\epsilon$ is a small constant introduced for numerical stability. This formulation embeds an inductive bias that encourages block separation—a crucial aspect that remains a challenge, leaving room for improvement in effectively leveraging supervised signals.

### A3.2 ENCOURAGES DISCRIMINATIVE COMPONENT

We explore how hash learning, or discrete representation, can strengthen the *Discriminative Component* in Theorem 1.

**Hash Center Learning** Given a set of varying low-level concepts $\mathbf{z}$, we define the mean representation for category $y_i$ as $\bar{z}_{s,i} = \frac{1}{k}\sum P_{z_s,i}$. This mean vector serves as a category prototype and is mapped to a hash center via a linear transformation to approximate the inverse mapping function $\hat{m}^{-1}$, yielding $\hat{\mathbf{z}} = \hat{m}^{-1}(\hat{\mathbf{z}}) \in \mathbb{R}^{n_d}$, where $n_d$ represents the dimensionality of the high-level concept space. Similarly, an individual image feature $\mathbf{z}_i$ is transformed into a corresponding hash feature $b_i = \text{hash}(\hat{z}_{s,i})$. To ensure consistency within a category, we enforce alignment between hash features $\mathbf{b}_i$ and their corresponding category hash centers while maintaining distinctiveness from other category centers. This objective is formulated as

$$\mathcal{L}_f = \frac{1}{|B|}\sum_{i \in B} \ell(y_i, \text{sim}(\mathbf{b}_i, \bar{\mathbf{z}}_s)),$$

where $\text{sim}(\mathbf{b}_i, \bar{\mathbf{z}}_s)$ is a similarity vector containing the cosine similarities between the hash feature $\mathbf{b}_i$ and all category hash centers.

**Hamming Distance in Discrete Space** In discrete space, the *Hamming distance* measures the number of differing components between two binary vectors:

$$d_H(\mathbf{b}_1, \mathbf{b}_2) = \sum_{i=1}^{n} \mathbf{1}(b_{1,i} \neq b_{2,i}) \tag{33}$$

where $\mathbf{b}_1, \mathbf{b}_2 \in \{0,1\}^n$. If the two vectors differ in only one component, the Hamming distance is exactly $d_H = 1$, ensuring a fixed, non-zero separation.

**Euclidean Distance in Continuous Space** In contrast, in continuous space, the *Euclidean distance* is defined as:

$$d_E(\mathbf{x}_1, \mathbf{x}_2) = \sqrt{\sum_{i=1}^{n}(x_{1,i} - x_{2,i})^2} \tag{34}$$

where $\mathbf{x}_1, \mathbf{x}_2 \in \mathbb{R}^n$. If only one component $i$ differs by $\epsilon$, the Euclidean distance reduces to: $d_E = |\epsilon|$.

**Implications for Representation Learning** To ensure clear separation between representation vectors when only a single component differs, discrete representations (e.g., binary vectors) provide a stronger separation than continuous ones. Hamming distance enforces a minimum unit difference, meaning that even a single bit flip results in a fixed, non-zero separation. In contrast, in continuous space, Euclidean distance can be arbitrarily small depending on the magnitude of the change in a single component. For example, if two binary vectors differ in only one position, their Hamming distance remains precisely 1, while the Euclidean distance between two continuous vectors may be significantly smaller if the change is minor. This property is particularly crucial in representation learning, including prototype-based learning, contrastive learning, and hashing-based retrieval methods, where robust separation is necessary.

Empirical studies in binary hashing for similarity search (Weiss et al., 2009) and deep metric learning (Schroff et al., 2015) support this observation, demonstrating that discrete constraints can enhance the robustness and interpretability of learned representations. Additionally, (Gong et al., 2013) showed that binary representations achieve superior separation in large-scale image retrieval tasks compared to continuous embeddings.

### A3.3 ASSUMPTION-COMPONENT-INTUITION

To clarify the motivation of each component in our implemented framework, we present a straightforward assumption-component-intuition specification in Table A1.

### A3.4 ALGORITHM

To clarify related details, we present the entire training algorithm of our $C^3$ for the OCD problem in Algo. 1.

## A4 IMPLEMENTATION DETAILS

### A4.1 DATASET DESCRIPTION

We evaluate our method following established benchmarks in the OCD literature (Du et al., 2023; Zheng et al., 2024). Specifically, we use four challenging fine-grained subsets from iNaturalist 2017 (Van Horn et al., 2018) (Fungi, Arachnida, Animalia, and Mollusca), as well as CUB (Wah et al., 2011), Stanford Cars (Krause et al., 2013), Oxford Pets (Parkhi et al., 2012), and Food101 (Bossard et al., 2014). Following the standard protocol (Du et al., 2023; Zheng et al., 2024), the categories are split into two subsets (seen and unseen) at the super-category. By default, 50% of the samples from the seen classes are included in the training set $\mathcal{D}_S$, while the remaining samples are used in the unlabeled set $\mathcal{D}_Q$ for evaluation. During inference, test images are processed independently as they arrive in a streaming fashion (Du et al., 2023).

### A4.2 EXPERIMENTS ON SYNTHETIC DATA

**Simulated Data Generation** We generate synthetic data following the generating process of $\mathbf{x}$ in Eq. 1. We work with latent variables $\mathbf{z}$ of 4 dimensions with $n_{zc} = 2$, $n_z = 2$ and $n_d = 2$. We sample $\mathbf{z}_c \sim \mathcal{N}(\mathbf{0}, \mathbf{I})$ and $\hat{\mathbf{z}} \sim \mathcal{N}(\mu_y, \sigma_y^2 \mathbf{I})$ where for each label $y$, including known classes $y$ and unknown classes $y^q$, we sample $\mu_y \sim \text{Unif}(-4, 4)$ and $\sigma_y^2 \sim \text{Unif}(0.01, 1)$. We use 2-layer MLPs to estimate $\hat{m}$ and $\hat{g}$.

**Architecture** For all methods in our synthetic data experiments, the VAE encoder and decoder are 6-layer MLP's with a hidden dimension of 32 and Leaky-ReLU ($\alpha = 0.2$) activation functions. For our method, we use component-wise spline flows (Durkan et al., 2019) with monotonic linear rational splines to modulate the change components. We use 8 bins for the linear splines and set the bound to be 5.

**Implementation Details** To simplify the simulations experiments, we employ a tailored version of our methodology for synthetic data, given that all data are numerical. Specifically, the observation $\mathbf{x}$ directly corresponds to the synthetic generation results. To promote the discrimination of concepts cross categories, we apply the cross-entropy loss directly to the obtained concepts $\hat{\mathbf{z}}$ and $\mathbf{z}$, with a Sigmoid function $\sigma(\cdot)$ is additionally applied to $\hat{\mathbf{z}}$ to encourage component-wise separation on high-level. We choose $g$, $f$ and $m$ to be a MLPs with the Leaky-ReLU activation function.

**Hyper-parameters Settings** We apply AdamW to train VAE and flow models for 200 epochs. We use a learning rate of 0.002 with a batch size of 128. The weight decay parameter of AdamW is set to 0.0001. For VAE training, we set the $\beta$ parameter of the KL loss term to 0.1.

Table A1: Mapping from theoretical assumptions (A1–A6) to intuition and concrete components in our implementation.

| Assump. | Meaning | Intuition | Components |
|---|---|---|---|
| A1 | Well-posed density over $(\hat{z}, \hat{c} \mid x)$ and over the semantic subspace. | The latent variables should have a smooth, non-degenerate probability landscape, so that small changes in $z$ or $c$ lead to bounded (rather than abrupt) changes in log-probability almost everywhere. This is exactly the motivation behind KL regularization in VAEs. | (i) Standard VAE / ELBO-based modeling (Eq. 10) ensures a desired joint density. (ii) Conditional normalizing flow with $L_{\text{flow}}$ (Eq. 7–8) yields a smooth, strictly positive density over the semantic subspace. |
| A2 | *Context invariance / comparison of $c$*: identifiability of $c$ without capturing semantic variations. | Contextual variables $c$ capture shared background information (scene, imaging conditions, etc.) that is stable across labels. It is separated from the semantic information stored in the $\mathcal{Z}$-space so that changing $y$ does not shift the distribution of $c$. | (i) Contextual loss (Eq. 4) enforces semantic invariance on $\hat{c}$, while label-discriminative variation is pushed into $\hat{z}$, separated by the dynamic MoE gating $M(s)$ via the Bernoulli mask (Eq. 6). (ii) Together with A3, this separation enables identifiability of semantic concepts. |
| A3 | *Semantic sufficient contrast in $z$*: local "shape vectors" of $\log p(\hat{z} \mid y)$ are affinely linearly independent across labels. | In a small neighborhood around $z$, each label $y$ induces its own local "shape" of $\log p(z \mid y)$ (via gradients, curvatures, selected mixed derivatives). Linear independence means no class's local shape can be reconstructed as a linear combination of the others. Geometrically, each label bends the probability landscape of semantic concepts in its own direction(s), which makes each identifiable. | (i) Prototype-based supervised loss $L_{\text{proto}}$ encourages $\hat{z}$ to be semantic-discriminative and to span distinct directions. (ii) Conditional flow with $L_{\text{flow}}$ (Eq. 7–8) provides an explicit, label-conditioned parametrization of $\log p(\hat{z} \mid y)$, inducing class-conditioned variations. |
| A4 | *Sparse Markov structure* over semantic concepts with modular dependencies. | Semantic concepts form a sparse dependency graph. When the learned graph is (even) sparser than the true one, each learned concept $\hat{z}_i$ corresponds either to a true semantic concept $z_{\pi(i)}$ or to a small neighborhood (cluster) around it, preserving interpretable modular structure. | (i) Sparsely-gated MoE learns a sparse adjacency structure over semantic concepts. (ii) $\ell_1$ sparsity loss $L_s$ (Eq. 5) on $\hat{z}$ promotes semantic sparsity. (iii) Flow regularization is restricted to $M_z \odot \hat{z}$, so only edges selected by $M_z$ can induce nonzero mixed derivatives. |
| A5 | *Support separation*: different concept values occupy distinct regions on the semantic manifold of each concept. | Different values of a concept (e.g., "striped" vs. "spotted") occupy separated rather than overlapping regions in the semantic support; otherwise, the semantic meaning of the same point/region can be ambiguous. This guarantees unique decoding of concept values. | (i) Approximated by hash loss $L_{\text{hash}}$ that pushes discrete hash codes of different semantics apart. (ii) Approximated by prototype-based $L_{\text{proto}}$ that pulls samples with different concept compositions to distinct regions. |
| A6 | *Coverage*: unseen categories share the same semantic space with seen categories. | Under sufficient semantic diversity in seen categories, unseen categories can be expressed as new compositions of already learned primitives. | No explicit loss is introduced; to avoid ill-posed scenarios where we cannot separate novel categories. |

### A4.3 EXPERIMENTS ON REAL-WORLD DATA

**Implementation Details** Following the basic OCD setting of (Du et al., 2023; Zheng et al., 2024), we use the DINO-pretrained ViT-B-16 (Dosovitskiy, 2020) as the backbone. During training, only the

---

**Algorithm 1** C³ Training Algorithm for OCD

---

**Require:** Training set $\mathcal{D}_S = \{(o_i, y_i)\}_{i=1}^N$,
1: Frozen vision backbone $f_{\text{enc}}$,
2: Encoder $\hat{g}_\phi^{-1}$, Selection Module $M_\psi$, Decoder $g_\theta$,
3: Prototype projection head $g_p$,
4: Hashing projection head $h_{\text{hash}}$,
5: MoE structure learner $(W_g, b_g)$,
6: Invertible flow $f_{\text{flow}}$,
7: Hyperparameters $\lambda_1, \lambda_2, \lambda_3, \eta$.
**Ensure:** Trained parameters $\Theta = \{\phi, \psi, \theta, g_p, h_{\text{hash}}, W_g, b_g, f_{\text{flow}}\}$.

8: **for** each epoch $e = 1, \ldots, E$ **do**
9:      **for** each mini-batch $\{(o, y)\}$ sampled from $\mathcal{D}_S$ **do**
10:          $\mathbf{x} \leftarrow f_{\text{enc}}(o)$        ▷ Extract frozen feature ([CLS] token)
11:          $\mathbf{s} \leftarrow \hat{g}_\phi^{-1}(\mathbf{x})$        ▷ Map to intermediate feature
12:          $[\hat{\mathbf{z}}, \hat{\mathbf{c}}] \leftarrow M_\psi(\mathbf{s})$        ▷ Select semantic and contextual concepts
13:          $\tilde{\mathbf{x}} \leftarrow g_\theta(\hat{\mathbf{z}}, \hat{\mathbf{c}})$        ▷ Reconstruct frozen feature
         **// Variational objective (ELBO), Eq. 10**
14:          $L_{\text{ELBO}} \leftarrow -\log p_\theta(\mathbf{x} \mid \hat{\mathbf{z}}, \hat{\mathbf{c}}) + \text{KL}(q_\phi(\hat{\mathbf{z}} \mid \mathbf{x}) \,\|\, p(\hat{\mathbf{z}})) + \text{KL}(q_\phi(\hat{\mathbf{c}} \mid \mathbf{x}) \,\|\, \mathcal{N}(\mathbf{0}, \mathbf{I}))$
         **// Context invariance loss, Eq. 4**
15:          $L_{\text{ctx}} \leftarrow \text{KL}(p(\hat{\mathbf{c}}) \,\|\, \mathcal{N}(\mathbf{0}, \mathbf{I}))$
         **// Semantic sparsity loss, Eq. 5**
16:          $L_s \leftarrow \sum_i \|[\hat{\mathbf{z}}]_i\|_1$
         **// Structure learning on semantic concepts, Eqs. 6–8**
17:          $\mathbf{G} \leftarrow \text{Softmax}(W_g \hat{\mathbf{z}} + b_g)$
18:          $\mathbf{M}_z \leftarrow \text{TopKMask}(\mathbf{G}, k)$     ▷ Keep $k$ strongest relations among semantic concepts
19:          $\hat{\boldsymbol{\epsilon}} \leftarrow f_{\text{flow}}(\mathbf{M}_z \odot \hat{\mathbf{z}}, y)$
20:          $L_{\text{flow}} \leftarrow -\left(\log p_s(f_{\text{flow}}(\hat{\boldsymbol{\epsilon}})) + \log\left|\det \frac{\partial f_{\text{flow}}}{\partial \hat{\boldsymbol{\epsilon}}}\right|\right)$
         **// Supervised comparison objective on $\hat{\mathbf{z}}$**
21:          $L_{\text{proto}} \leftarrow$ prototype contrastive loss using $g_p(\hat{\mathbf{z}})$ and label $y$
22:          $L_{\text{hash}} \leftarrow$ supervised hashing loss on binary codes from $h_{\text{hash}}(\hat{\mathbf{z}})$
23:          $L_{\text{sup}} \leftarrow L_{\text{proto}} + L_{\text{hash}}$
         **// Overall loss, Eq. 11**
24:          $L_{\text{all}} \leftarrow L_{\text{sup}} + L_{\text{ELBO}} + \lambda_1 L_{\text{flow}} + \lambda_2 L_s + \lambda_3 L_{\text{ctx}}$
25:          $\Theta \leftarrow \Theta - \eta \nabla_\Theta L_{\text{all}}$        ▷ Update all trainable parameters
26:      **end for**
27: **end for**
28: **return** $\Theta$

---

Table A2: Statistics of datasets.

| | CUB | Scars | Pets | Food | Fungi | Arachnida | Animalia | Mollusca |
|---|---|---|---|---|---|---|---|---|
| $|Y_S|$ | 100 | 196 | 38 | 101 | 121 | 56 | 77 | 93 |
| $|Y_Q|$ | 200 | 98 | 19 | 51 | 61 | 28 | 39 | 47 |
| $|\mathcal{D}_S|$ | 1.5K | 2.0K | 0.9K | 19.1K | 1.8K | 1.7K | 1.5K | 2.4K |
| $|\mathcal{D}_Q|$ | 4.5K | 6.1K | 2.7K | 56.6K | 5.8K | 4.3K | 5.1K | 7.0K |

final block of ViT-B-16 is fine-tuned. In our approach, the low-level concept learner $\hat{m}^{-1}$ is a single linear layer with an output dimension set to 768, and then use a MLP to map the 3072, the low-level changing concepts $\mathbf{z}$ in our case. Each category has $k = 10$ prototypes. The fully connected layer in Eq. 2 is non-trainable, which uses positive weights 1 for prototypes from the same category and negative weights $-0.5$ for prototypes from different categories. The function $m^{-1}$ consists of three linear layers with an output dimension set to $n_d = 32$. We align all the experiments with setting this dimension for fair comparison. We set $\lambda_1 = 5 \times 10^{-2}$, $\lambda_2 = 1 \times 10^{-2}$, and $\lambda_3 = 1 \times 10^{-3}$.

**Contrastive Learning**. Additionally, we adopt the supervised contrastive learning (Khosla et al., 2020) on $\theta(y_i)$ to implicitly constraint that *Discriminative Component* assumption required in 2

$$\mathcal{L}_{\text{scl}} = \sum_{i \in \mathcal{I}} \frac{-1}{|\mathcal{P}(i)|} \sum_{p \in \mathcal{P}(i)} \log \frac{\exp\left(\text{sim}(\theta(y_i), \theta(y_p))/\tau\right)}{\sum_{a \in \mathcal{I} \setminus \{i\}} \exp\left(\text{sim}(\theta(y_i), \theta(y_a))/\tau\right)}, \tag{35}$$

where $\mathcal{I}$ is the set of all indices in the batch, $\mathcal{P}(i) = \{p \in \mathcal{I} \mid y_p = y_i, p \neq i\}$ is the set of indices corresponding to samples with the same label as $i$, $\theta(y_i)$ is the feature representation of the sample $i$ based on its label $y_i$, $\text{sim}(\theta(y_i), \theta(y_j)) = \frac{\theta(y_i)^\top \theta(y_j)}{\|\theta(y_i)\|\|\theta(y_j)\|}$ is the cosine similarity between representations. $\tau$ is a temperature scaling parameter.

**Comparison with Baselines.** Since OCD is a relatively new task that demands instantaneous inference, traditional baselines from NCD and GCD do not align with this setting. Therefore, we compare our approach against **SMILE** (Du et al., 2023) and **PHE** (Zheng et al., 2024), as well as three strong alternatives:

- **Sequential Leader Clustering (SLC)** (Hartigan, 1975), a classical clustering method designed for sequential data.
- **Ranking Statistics (RankStat)** (Han et al., 2021), which identifies categories based on the top-3 indices of feature embeddings.
- **Winner-Take-All (WTA)** (Jia et al., 2021), which determines category descriptors by selecting indices of maximum values within feature groups.

For consistency, we follow the baseline settings established in **SMILE** (Du et al., 2023) when configuring the three additional methods.

### A4.4 Statistical Robustness of Main Results

To assess the reliability of our findings, we report the average performance over three independent runs for each experimental setting. Table A3 provides the detailed results for our method, including both the mean and the population standard deviation, thereby quantifying the variability due to stochasticity in training and evaluation.

Table A3: Mean and standard deviation of accuracy across three independent runs for each setting.

| Dataset | All | Old | New |
|---|---|---|---|
| CUB | 39.7±0.18 | 60.9±1.43 | 29.1±0.67 |
| Stanford Cars | 33.3±0.24 | 68.3±1.09 | 17.2±0.51 |
| Oxford Pets | 53.7±0.35 | 62.5±2.16 | 49.1±1.42 |
| Food101 | 30.3±0.27 | 67.0±0.38 | 11.2±0.46 |
| Fungi | 32.6±0.30 | 69.2±1.53 | 16.0±0.49 |
| Arachnida | 37.6±0.22 | 70.1±0.88 | 13.2±0.54 |
| Animalia | 41.5±0.33 | 58.3±1.24 | 32.6±1.07 |
| Mollusca | 41.6±0.29 | 69.4±2.01 | 27.0±1.18 |

### A4.5 Analysis on Hyperparameters.

Fig. A1 presents a systematic analysis of key hyperparameters and their influence on model performance. We investigate three core factors: the dimensionality of the semantic representation ($d_z$), the dimensionality of the contextual representation ($d_c$), and the structure regularization weight ($\lambda_{\text{flow}}$), which governs the contribution of structure learning, a central component of our method. Experiments are conducted on two fine-grained datasets to ensure robustness across varying conditions. Results show that model performance is sensitive to $d_z$ and the sparsity imposed on contextual invariance. Notably, an appropriately tuned $\lambda_1$ is essential for stable and effective structure discovery. These findings underscore the necessity of learning identifiable and well-structured latent concepts to support the subsequent OCD.

### A4.6 Effect of Sparsity Regularization

To evaluate the sensitivity of our method to the sparsity regularization coefficient $\lambda_{\text{sparse}}$ on the gated network, which is responsible for identifying distinctive concepts and modular concepts, we conduct

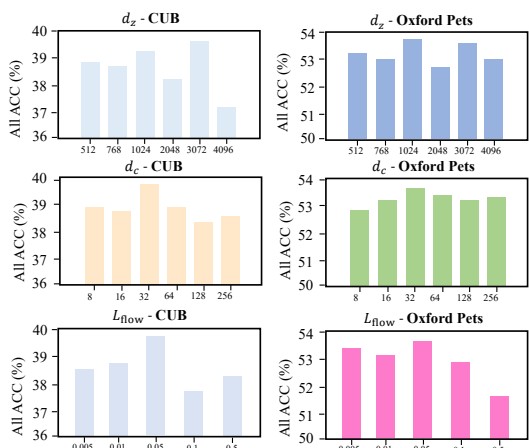

Figure A1: **Analysis on hyper-parameters.**

experiments under four different settings. Table A4 presents the ALL accuracy (mean ± std) across three independent runs. We observe that an inappropriate choice of sparsity coefficient significantly impairs performance.

Table A4: Effect of $\lambda_{\text{sparse}}$ for concept selection on $C^3$ performance (mean ± std over three runs). The setting $\lambda = 0.1$ corresponds to the main results in Table 1.

| Dataset | $\lambda = 0.0$ | $\lambda = 0.01$ | $\lambda = 0.1$ | $\lambda = 1.0$ |
|---|---|---|---|---|
| CUB | $36.8 \pm 0.5$ | $38.9 \pm 0.3$ | $\mathbf{39.7 \pm 0.2}$ | $35.1 \pm 0.6$ |
| Stanford Cars | $30.7 \pm 0.6$ | $32.6 \pm 0.4$ | $\mathbf{33.3 \pm 0.3}$ | $29.4 \pm 0.7$ |
| Oxford Pets | $49.2 \pm 0.4$ | $51.6 \pm 0.3$ | $\mathbf{53.7 \pm 0.2}$ | $48.1 \pm 0.5$ |
| Food101 | $27.9 \pm 0.7$ | $29.4 \pm 0.4$ | $\mathbf{30.3 \pm 0.3}$ | $26.5 \pm 0.6$ |
| Average | $36.1 \pm 0.5$ | $38.1 \pm 0.3$ | $\mathbf{39.2 \pm 0.2}$ | $34.8 \pm 0.6$ |

**Results on Synthetic Data.** As shown in Table A5, several results support the validity of our theoretical insights. The Mean Correlation Coefficiency (MCC) quantifies the degree of component-wise identifiability of $\mathbf{z}$, whereas $R^2$ measures the subspace identifiability of $\hat{\mathbf{c}}$. Higher values for all metrics indicate better identifiability. First, we observe that both $R^2$ and the MCC of our method increase *monotonically* with the number of known classes within the evaluated range (up to $n_s$=24), at which both subspace and component-wise latent variables achieve high identifiability scores. This confirms our theoretical prediction and aligns with the intuition that a sufficient number of known classes is necessary for identifiability. Specifically, when the condition $2d_z + |\mathcal{M}| + 1 \leq n_s$ is not satisfied, both MCC and accuracy drop to impractically low values, further validating the assumption that adequate comparisons are required. Notably, we find that strong performance can still be achieved with a large number of concepts, *e.g.*, $d_z = 9$, highlighting why our method remains effective even in complex fine-grained datasets.

**Concept-guided Classification** We also provide a cross-category concept analysis between *Black-Footed Albatross* and *Sooty Albatross* to further validate our theoretical claims. From their pictures, their "body" and "head" are visually distinct. We apply the learned concepts to perform binary classification, as shown in Table A6. Specifically, we group latent variables into concept modules (*e.g.*, head, wings, body) based on the learned Markov network. Using these learned concepts, we train a simple binary logistic classifier to distinguish Black-footed vs. Sooty Albatross, where the resulting accuracy quantifies each concept's discriminative ability. The results show that the distinctive concepts are semantically meaningful and serve as faithful concept descriptors, verifying our theorems and claims on "comparison" in identifying true latent concepts.

**Scaling to larger foundation model backbones.** We evaluate our framework with large vision–language backbones by adopting only the *visual encoders* from the CLIP family, since the OCD

| $d_z$ | Metric | $n_s=6$ | $n_s=12$ | $n_s=18$ | $n_s=24$ |
|---|---|---|---|---|---|
| 2 | MCC | 0.73 | 0.89 | 0.90 | 0.95 |
| | $R^2$ | 62.6 | 75.3 | 90.3 | 95.5 |
| | Acc. | 51.3 | 65.7 | 73.6 | 79.8 |
| 5 | MCC | 0.81 | 0.85 | 0.91 | 0.92 |
| | $R^2$ | 73.6 | 77.2 | 79.3 | 89.2 |
| | Acc. | 43.5 | 59.9 | 67.0 | 72.2 |
| 9 | MCC | 0.31 | 0.65 | 0.75 | 0.74 |
| | $R^2$ | 12.6 | 67.6 | 82.3 | 86.1 |
| | Acc. | 10.9 | 35.5 | 40.4 | 68.4 |

Table A5: **Identifiability Results on Synthetic Data**. $n_s$ denotes the number of known categories.

| Concept | Latent Variables | Cls Acc. (%) | Notes |
|---|---|---|---|
| Body | $z_1, z_2$ | 92.3 | discriminative latent concepts, comparison |
| Tail | $z_5$ | 64.1 | not discriminative |
| Wings | $z_3, z_4$ | 78.5 | not discriminative |
| Head | $z_8$ | 95.0 | discriminative latent concepts, comparison |

Table A6: Concept-guided classification results by latent variables.

setting provides no language supervision. The CLIP image embeddings are used as inputs to our network without modifying the training protocol. Results are shown below.

**Scaling to larger datasets with more semantic categories.** We evaluate the scalability of C$^3$ by merging iNaturalist subsets—*Fungi, Arachnida, Animalia*, and *Mollusca*—to substantially increase the number of unseen categories. As shown in Table A8, as more subsets are merged, *New* accuracy consistently increases while *Old* decreases moderately; the overall *All* metric also changes accordingly. This confirms that comparison-and-composition benefits scale with available seen semantics rather than overfitting to them.

**Varying the number of unknown labels.** We follow the same protocol while adapting the known : unknown ratio and report CUB performance as the number of unknown labels increases (Table A9). Results show consistent gains in *All/Old/New* as the unknown set shrinks (i.e., more seen semantics are available).

**Ablation on comparison and composition.** We perform a leave-one-out ablation to quantify the contribution of each component in C$^3$, separating *comparison* (e.g., $\mathcal{L}_{\text{flow}}$, $\mathcal{L}_{\text{proto}}$, $\mathcal{L}_{\text{ctx}}$) from *composition* (e.g., sparsity $\mathcal{L}_S$ and residual integration $f_{\text{res}}$), while the reconstruction term $\mathcal{L}_{\text{ELBO}}$ stabilizes training. As reported in Table A10, removing any single module consistently degrades performance; the full model achieves the best overall results, indicating that comparison and composition play complementary roles.

### A4.7 STABILITY OF TRAINING

**Training Curves: Compatible Regularizers.** As shown in Figure A2, we include the training curves of the independent regularizers, including the flow loss, likelihood (ELBO), prototype contrast, and sparsity on the Markov network.

**Computational efficiency and scalability.** We analyze the computational cost and scaling behavior of $C^3$ on CUB. As summarized in Table A11, the flow and MoE modules introduce only marginal overhead—approximately X% additional parameters and Y% increase in inference latency—while substantially improving model expressiveness and convergence stability. The added costs are limited because both modules operate in a low-dimensional, sparsity-inducing space. Moreover, the training loss exhibits smooth, stable convergence across datasets (Fig. A2), and we observe near-linear scaling with respect to the number of latent variables and input dimensions, indicating practicality for larger-scale applications.

| | CUB (%) | | | SCars (%) | | |
|---|---|---|---|---|---|---|
| Method | All | Old | New | All | Old | New |
| CLIP (ViT-B/16) | 33.5 | 58.9 | 21.8 | 26.7 | 52.1 | 14.9 |
| CLIP (ViT-L/14) | 28.9 | 51.0 | 14.6 | 22.3 | 45.5 | 10.2 |
| Ours (DINO ViT-B/16) | 40.1 | 62.1 | 29.5 | 34.1 | 69.0 | 17.8 |

Table A7: Scaling to larger foundation backbones: comparison on CUB and Stanford Cars using CLIP visual encoders.

| Merge setting | All (%) | Old (%) | New (%) |
|---|---|---|---|
| $C^3$ (Fungi only) | 32.9 | 69.8 | 16.2 |
| $C^3$ (Fungi + Arachnida) | 34.5 | 66.7 | 19.8 |
| $C^3$ (Fungi + Arachnida + Animalia) | 36.4 | 63.5 | 22.9 |
| $C^3$ (All four merged) | 38.7 | 60.1 | 25.8 |

Table A8: $C^3$ **on merged iNaturalist subsets** (Fungi, Arachnida, Animalia, Mollusca). As the merge size grows, *New* improves and *Old* decreases modestly, indicating stronger open-set generalization with broader semantics.

## A5 DISCUSSION

**Compatibility of A2 and A3** We want to clarify that A2 is a weaker version of A3 (since isolating subspace $\mathbf{z}$ and $\mathbf{c}$ requires less diversity than disentangling concept), and then, Assumption A2 / A3 are imposed on the data-generating process and is, in fact, a mild requirement for any successful semantic classification: if the latent concepts themselves lack sufficient diversity, neither humans nor machines can distinguish categories meaningfully. However, standard neural networks often fail to capture such diversity in practice, as their training objectives can converge to degenerate local minima that entangle concepts across categories.

## A6 BROADER IMPACTS

The proposed framework advances On-the-fly Category Discovery by enabling the identification and composition of semantically meaningful latent concepts. This supports robust generalization to novel categories under open-world conditions, addressing key limitations in current recognition systems. By aligning learned representations with human-interpretable concepts, we enhance model transparency and support applications requiring semantic understanding in dynamic environments, such as robotics, healthcare, and autonomous systems. Its theoretical foundations in identifiability also contribute to broader efforts in interpretable and causally grounded representation learning. Our method thus offers both practical utility and theoretical insight for building adaptive, explainable AI systems.

One potential limitation is that, in practice, novel categories may contain semantic primitives/concepts beyond compositions of seen primitives encountered during training, especially due to the insufficient seen category diversity, thus violating our coverage assumption on the shared identified concept

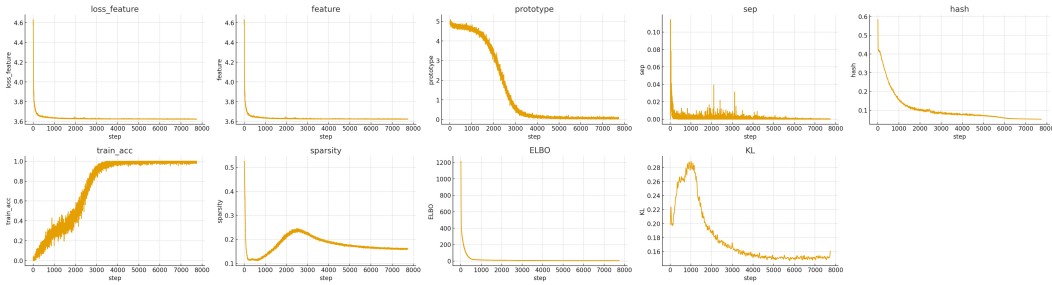

Figure A2: Loss curves in training stage.

| # Unknown labels | CUB All (%) | CUB Old (%) | CUB New (%) |
|---|---|---|---|
| 150 | 23.0 | 32.1 | 17.5 |
| 120 | 31.5 | 51.3 | 22.3 |
| 100 | 40.1 | 62.1 | 29.5 |

Table A9: **Scaling with unknown category count on CUB.** Performance improves as the number of unknown labels decreases (more seen semantics), supporting the scalability of $C^3$ under increasing semantic coverage.

| | CUB (%) | | | SCars (%) | | |
|---|---|---|---|---|---|---|
| Configuration | All | Old | New | All | Old | New |
| **Full (all modules)** | **40.1** | **62.1** | **29.5** | **34.1** | **69.0** | **17.8** |
| $- \mathcal{L}_{\text{flow}}$ | 38.5 | 61.3 | 27.0 | 30.2 | 59.6 | 16.0 |
| $- \mathcal{L}_S$ (sparsity) | 39.3 | 62.4 | 27.7 | 31.1 | 59.0 | 17.6 |
| $- \mathcal{L}_{\text{ELBO}}$ (reconstruction) | 36.9 | 59.8 | 24.5 | 27.6 | 54.2 | 15.5 |
| $- \mathcal{L}_{\text{ctx}}$ (context invariance) | 39.3 | 61.7 | 28.9 | 33.0 | 66.5 | 17.2 |
| $- \mathcal{L}_{\text{proto}}$ (prototype contrast) | 36.1 | 60.7 | 24.3 | 28.3 | 55.8 | 15.2 |
| $- f_{\text{res}}$ (integration on Markov network) | 39.5 | 61.8 | 28.5 | 33.2 | 67.8 | 17.1 |

Table A10: **Extended ablation study on $C^3$.** We remove one module at a time from the full model. Performance drops across both benchmarks confirm that each component contributes and that comparison & composition are complementary.

space. We argue that this problem, though challenging, is fundamental to understanding deep network generalizability. We therefore leave this for future work, including but not limited to test-time learning or evolution strategies, as well as establishing more general learnability conditions for genuine semantic generalization.

## A7 DISCLOSURE OF LLM USAGE

In accordance with the ICLR 2026 policy on the use of Large Language Models (LLMs), we disclose that LLMs were used to aid in polishing the writing of this paper. No part of the research ideation, experiment design, implementation, or analysis relied on LLMs. The authors take full responsibility for the contents of the paper.

| Model Variant | Params (M) | Training Time (min) | Inference Time (ms) | GPU Memory (GB) |
|---|---|---|---|---|
| PHE (baseline) | 22.4 | 70.5 | 1.18 | 5.2 |
| Base (w/o Flow, MoE) | 23.1 | 72.3 | 1.21 | 6.4 |
| + Flow Module | 24.4 | 75.8 | 1.25 | 6.8 |
| + MoE Module | 24.9 | 76.2 | 1.27 | 6.9 |
| **Full $C^3$ (Flow + MoE)** | **25.0** | **76.8** | **1.29** | **7.0** |

Table A11: **Computational efficiency on CUB dataset.** The flow and MoE modules add minimal overhead while improving convergence stability; both operate in a low-dimensional, sparse regime.

