# OpenReview forum: "From Comparison to Composition: Towards Understanding Machine Cognition of Unseen Categories"
_ICLR.cc/2026/Conference — Submitted to ICLR 2026_

### Official Review · Reviewer_MWi7 · 2025-10-26

**Soundness:** 3
**Presentation:** 3
**Contribution:** 3
**Rating:** 6
**Confidence:** 3

**Summary:**

The paper proposes a new approach for the problem of online category discovery. It comprises two parts. In the first part, a theoretical formulation of the problem is presented. This is based on a stochastic model where old and new categories are modelled as sets of concepts. Statistical assumptions are given for model identifiability, as well as for the ability of the model to detect and recognise new categories as they are encountered in the data stream. The second part of the paper presents a practical deep learning formulation that is inspired by the first part. When assessed on standard benchmarks for this task, this new formulation performs well. Some ablations illustrate the importance of various elements of the formulation.

**Strengths:**

* Category discovery captures an important problem where machine learning still struggles. Online category discovery is a variant of this problem somewhat closer to a real-life application, where new data streams through and new categories must be detected and instantiated on the fly. These are difficult problems worth looking at.

* Insofar as I can tell, the authors propose a reasonable formulation for online category discovery supported by a thorough formal analysis.

* The practical instantiation of this formulation is shown to perform relatively well on canonical benchmarks in the area.

* Ablations show the importance of at least some of the key design choices.

* A long appendix provides many important aspects of the formulation and implementation that are missing in the main paper.

**Weaknesses:**

The approach is rather complex, involving a number of regularisers and steps that would be difficult to reproduce based solely on the description in the paper (even once one considers the thorough appendix). The authors do not mention whether code will be released.

On a fundamental level, this paper assumes that categories map neatly to combinations of "base concepts", and that new categories are formed as new combinations of concepts that are, otherwise, already known. This sounds like a fundamental limitation which should probably be acknowledged and discussed a bit more.

There are no illustrations of what the concepts discovered by the algorithms are, particularly on the computer vision dataset. Are these interpretable in any way?

The main paper is not entirely self-contained and some critical information is missing, which makes it unnecessarily difficult to understand:

* The problem formulation on line 264 is incomplete, and one *must* read the references to even understand what problem is being solved. Specifically, I could not understand the sentence [the model] "classifies streaming query instances in real-time without access to labels from novel categories" until reading Du et al. 2023. The task is poorly defined in the paper, and so are the evaluation metrics.

* It would have been very helpful to expand on several of the introductory materials. What is a "basis distribution" in Eq. (1) (line 107)? Why does it make sense for $f$ to list $\mathbf{z}$ both as an input and as an output (line 131)? How are Definition 1 and Eq. (1) connected? One is required to be familiar with prior work, and with Markov networks and other background, to fill in the gaps.

* In Eq. (4), do you mean to say that $p(\hat \mathbf{c}|y)$ is independent of $y$?

* In Eq. (5), is $[\hat\mathbf{c}_i]$ a vector? Line 106 suggests that it is a scalar instead. If so, why do you need a vector norm?

Minor: Line 408 suggests that Table 1 reaches a peak at $n_s=24$, but the table ends at 24. How can you tell this is a peak?

**Questions:**

Please see the questions included in the "weaknesses" section above.

---

> ### Author Response · Authors · 2025-11-24
> **Author Response (1/3)**
>
> Dear Reviewer MWi7, thank you for your insightful feedback. Your comments helped improve the rigor of our work—particularly in clarity of presentation, terminology, and interpretability analysis. We sincerely appreciate your time and effort, and have updated the paper and appendix accordingly.
>
> We provide point-by-point responses to your comments below and have updated the manuscript accordingly.
>
> >W1: The approach is rather complex, involving a number of regularisers and steps that would be difficult to reproduce based solely on the description in the paper (even once one considers the thorough appendix). The authors do not mention whether the code will be released.
>
> In response to your comment, we have:
>
> - Enhanced implementation details and appendix by elaborating on each module.
> - Added integral *pseudocode* (Appendix Algorithm 1) that explicitly specifies the training procedure, regularizer recipes, and hyperparameters.
> - Clarified architectural details, loss weights, and optimization details.
>
> We are committed to releasing our complete code and trained models upon acceptance to ensure full reproducibility. This will include training scripts, evaluation protocols, and detailed instructions for replication on all included benchmarks. We hope these revisions effectively improve the paper's clarity and reproducibility. Please let us know if any specific technical details require further elaboration.
>
>
> >W2: On a fundamental level, this paper assumes that categories map neatly to combinations of "base concepts", and that new categories are formed as new combinations of concepts that are, otherwise, already known. This sounds like a fundamental limitation which should probably be acknowledged and discussed a bit more.
>
> We appreciate your insightful comment and agree that the composition assumption deserves clearer discussion. In fact, this limitation is explicitly formalized in Assumption A6 and Theorem 3: our guarantees apply to unseen categories whose underlying generative process reuses the same functional concept space as the seen data.
>
> Importantly, this does not mean that all visually novel patterns (e.g., new textures or part configurations) cannot be recognized: because C$^3$ allows interdependencies via the latent Markov network, different interactions of the same identified abstract concepts can still yield rich, previously unseen visual patterns at test time.
>
> However, this represents a fundamental constraint rather than a limitation: when unseen categories differ only along concept dimensions absent from training data, the problem becomes mathematically ill-posed, even human learners cannot distinguish such categories without additional information or further training.
>
> In other words, if two unseen categories differ only along concept dimensions that never vary or appear in the seen data, then by identifiability theory, we cannot reliably distinguish them from training alone, and the problem becomes ill-posed. We have made this scope explicit in the limitations/discussion section (Line 520-528) and highlight continual or test-time learning mechanisms for expanding the concept space beyond the initial one as an important direction for future work.

---

> ### Author Response · Authors · 2025-11-24
> **Author Response (2/3)**
>
> >W3: There are no illustrations of what the concepts discovered by the algorithms are, particularly on the computer vision dataset. Are these interpretable in any way?
>
> Thanks for raising these points. Yes, the discovered concepts are interpretable on the vision datasets. To support our claims, we provide:
>
> **1. Theoretical Guarantee via Identifiability:** Our identifiability theory ensures that, when the specified conditions hold, the learned concepts correspond to the ground-truth semantic primitives up to trivial transformations (e.g., permutation). This is a fundamental distinction from prior methods: once these semantic concepts are identified, the representation of test data from unseen categories is guaranteed to be a meaningful combination of learned primitives under the coverage condition (A6). In contrast, previous methods cannot guarantee the semantic meaningfulness of their representations for unseen categories due to distribution shift; they lack theoretical grounding that their learned features correspond to true semantic primitives. Consequently, their approaches reduce to generic feature interpolation without semantic guarantees.
>
> **2. Visual Evidence:** Our qualitative visual analysis (Figure 4 in revised version) and the attached concept activation maps show that the model focuses on semantically meaningful parts rather than spurious patterns: across multiple samples, the columns labeled tail/wing/head/body consistently highlight the corresponding regions (e.g., head maps concentrate on crown+bill, tail maps on the posterior, wing maps on extended primaries, and body maps on the torso) with minimal background activation and stable localization under pose/view changes, indicating disentangled, part-selective concepts. Quantitatively, C$^3$ achieves strong cross-dataset performance—e.g., CUB 40.1/62.1/29.5 (All/Old/New) and SCars 34.1/69.0/17.8—while on the merged iNaturalist subsets it maintains 38.7/68.1/22.2, further supporting that the learned atoms are both discriminative and compositionally transferable.
>
> **3. Empirical Evidence:**
>
> **(1) Simulated Experiments:** To address this concern, in our submission we have included experiments on concept-generating data where the ground-truth concept structure is known. The results in Table A5 and below show that C$^3$ can successfully recovers disentangled latent variables corresponding to these underlying concepts. MCC indicates the extent of latent variable-concept alignment, and Acc. indicates the OCD performance.
>
> *Table 1. Concept Identifiability Results on Synthetic Data. $n_s$ denotes the number of known categories.*
>
> | $d_z$ | Metric | $n_s=6$ | $n_s=12$ | $n_s=18$ | $n_s=24$ |
> |:------:|:--------|:-------:|:--------:|:--------:|:--------:|
> | **2** | MCC | 0.73 | 0.89 | 0.90 | 0.95 |
> |  | $R^2$ | 62.6 | 75.3 | 90.3 | 95.5 |
> |  | Acc. | 51.3 | 65.7 | 73.6 | 79.8 |
> | **5** | MCC | 0.81 | 0.85 | 0.91 | 0.92 |
> |  | $R^2$ | 73.6 | 77.2 | 79.3 | 89.2 |
> |  | Acc. | 43.5 | 59.9 | 67.0 | 72.2 |
> | **9** | MCC | 0.31 | 0.65 | 0.75 | 0.74 |
> |  | $R^2$ | 12.6 | 67.6 | 82.3 | 86.1 |
> |  | Acc. | 10.9 | 35.5 | 40.4 | 68.4 |
>
> **(2) Concept-guided Classification:**
>
> We also provide a cross-category concept analysis between *Black-Footed Albatross* and *Sooty Albatross* to further validate our theoretical claims. From their pictures, their "body" and "head" are visually distinct.
>
> Then we use **a subset** of learned concepts to do binary classification based on their plausible interpretation, as shown in Table A6 in our revised manuscript and table below. Specifically, we separate these concept groups based on the learned Markov network structure.
>
> | Concept | Latent Variables | Cls Accuracy (%) | Notes |
> |----------|------------------|------------------|--------|
> | Body | $z_1$, $z_2$ | 92.3 | discriminative latent concepts |
> | Tail | $z_5$ | 64.1 | not discriminative |
> | Wings | $z_3$, $z_4$ | 78.5 | not discriminative |
> | Head | $z_8$ | 95.0 | discriminative latent concepts |
>
> Taking together, the extended experiments and visualizations demonstrate that C$^3$ learns disentangled and human-interpretable concepts.

---

> ### Author Response · Authors · 2025-11-24
> **Author Response (3/3)**
>
> > W4(1): “Basis distribution” $p_{\epsilon}$ in Eq. (1)
>
> In Eq. (1):
> $$
> x = g(z, c), \quad z = f(z, y, \epsilon), \quad c \sim p_c, \ \epsilon \sim p_{\epsilon},
> $$
> the *basis distribution* $p_{\epsilon}$ is the shared, category-invariant source from which category-dependent semantic distributions $p_z$ are derived. We model $p_z$ as a transformation of $p_{\epsilon}$, allowing semantic variation across labels $y$.
>
> > W4(2): Why $z$ appears as both input and output in $f(z, y, \epsilon)$
>
> This follows the latent *Markov network* $M$ capturing dependencies among semantic concepts $z = [z_1,\ldots,z_d]$. Including $z$ as input expresses causal relations among components.
> Example: for $z_1 \!\to\! z_2,$
> $$
> z_1 = y + \epsilon_1, \quad z_2 = z_1 + y + \epsilon_2, \quad \text{i.e. } z = A z + \epsilon, \
> A = \\begin{bmatrix}
> 0 & 1 \\\\
> 0 & 0
> \\end{bmatrix}.
> $$
> which expands to
> $$
> z_1 = f_1(z, y, \epsilon_1) = \epsilon_1, \quad
> z_2 = f_2(z, y, \epsilon_2) = z_1 + \epsilon_2.
> $$
> This illustrates that $f$ maps the latent variables and noise to semantic components according to the dependency structure in the Markov network.
>
> > W4(3): Connection between Definition 1 and Eq. (1)
>
> We use Definition 1 to formalize the *maximum clique* $\Psi_M(z_i)$, a property of the Markov network in **response (2)**, describing which latent components interact. Eq. (1) specifies the generative process, while Definition 1 provides the structural property for disentangling $z_i$. Together, they determine which semantic factors are identifiable under the assumed sparsity.
>
> In summary:
> $$
> \text{Eq.(1)} \;=\text{structure within concepts}\Rightarrow\; \text{Markov network } M \;=\text{a property}\Rightarrow\; \text{Definition 1}.
> $$
> This logic chain shows that Eq.(1) defines the generative mechanism, the Markov network captures inter-concept relations, and Definition 1 provides the graph-based condition for disentangling and identifying semantic factors under sparsity assumptions.
>
> > W4(4): In Eq. (4), do you mean to say that $p(\hat{\mathbf{c}} | y)$ is independent of $y$?
>
> Yes, you are correct, $c$ encodes category-invariant context (e.g., background, lighting) and is not a function of $y$.
>
> > W4(5): In Eq. (5), is $\hat{\mathbf{c}}_i$ a vector? Line 106 suggests that it is a scalar instead. If so, why do you need a vector norm?
>
> In the line 106 of submission:
>
> $$
> \text{comprising contextual concepts } c \in \mathbb{R}^{d_c} \text{ (e.g., background, surroundings)}
> $$
>
> It is represented as a vector $c \in \mathbb{R}^{d_c}$, where $d_c \geq 1$ denotes its dimensionality. Thus, $c$ can be a scalar when $d_c = 1$ or a vector when $d_c > 1$. In this work, we use the vector form so that $c$ can effectively capture contextual information.
>
> > Minor Comment: Line 408 suggests Table 1 peaks at $n_s$, but the table ends at 24.
>
> Thank you for catching this. We would like to correct that it is not a peak; we intended to show a monotonic improvement as the number of known labels $n_s$ increases. In light of your comments, we have revised the phrasing accordingly in Line 1655-1658.
>
> This trend supports **Theorem 1**, which states that more known classes lead to higher semantic diversity (more linearly independent log-density) and thus enhances identifiability.

---

> ### Author Response · Authors · 2025-11-27
> **Thank you for your thoughtful review**
>
> Dear Reviewer MWi7,
>
> Thank you once again for this very detailed and thoughtful set of comments. This is a gentle reminder that the rebuttal period is approaching. If you have time, we would appreciate it if you could review our responses. If we have addressed your concerns, we would be grateful if you could reconsider your evaluation. Please let us know if you have any additional questions. Thank you very much again for your time and thoughtful feedback!
>
> The Authors

---

### Official Review · Reviewer_PD1H · 2025-10-31

**Soundness:** 3
**Presentation:** 2
**Contribution:** 3
**Rating:** 4
**Confidence:** 3

**Summary:**

This paper proposes C3 (Comparison–Composition Cognition), a cognitive-inspired theoretical and practical framework for generalizing to unseen categories. The authors formalize human-like “compare–then–compose” reasoning into two complementary mechanisms: comparison, which extracts latent concepts by capturing cross-category variation, and composition, which recombines these concepts to recognize unseen categories. Experiments on eight fine-grained benchmarks under the On-the-Fly Category Discovery (OCD) setting demonstrate consistent performance improvements.

**Strengths:**

1.This paper is mainly for providing formal identifiability guarantees for concept-based generalization. It features rigorous derivations, and its stepwise logic has a clear structure: first contextual separation, then semantic disentanglement, and ultimately compositional generalization.
2.Framing unseen-category cognition as “from comparison to composition” offers a interpretable metaphor that aligns with cognitive science findings.
3.Every theoretical assumption (A1–A6) is explicitly operationalized in the model (e.g., flow for smooth density, sparse MoE for structure, prototype and hashing losses for semantic contrast).

**Weaknesses:**

1.The exposition is heavy and unfocused. Many formulas are presented without sufficient intuition or geometric explanation. Readers must infer the motivation behind several derivations (e.g., partial derivatives of log-density in Theorem 2), making the theoretical contribution harder to follow.
2.While the framework borrows cognitive terminology “compare–then–compose”, its connection to actual cognitive mechanisms is largely metaphorical. A clearer justification for how these processes correspond to neural computations would strengthen the interdisciplinary claim.
3.Although the framework is complete, it feels assembled rather than organically derived. Each assumption leads to a separate engineering component.
4.Empirically, results are solid but do not reveal new phenomena or insights. The performance gains are incremental, and no analysis shows why the comparison–composition pipeline generalizes better than alternative factorization approaches.
5.The paper does not report computational efficiency, convergence stability, or scaling behavior; the additional flow and MoE modules could impose extra overhead.

**Questions:**

1.How sensitive are the identifiability results to approximate violations of A1–A6? Are these conditions ever testable in real datasets?
2.Could the use of multiple independent regularizers (flow likelihood, hashing, prototype contrast) lead to competing gradients that distort the latent structure?
3.Would a simpler variational or contrastive formulation achieve similar results without flow modeling?
4.How can the authors ensure that the claimed “composition” corresponds to interpretable recombination of learned semantic atoms, rather than a generic feature interpolation? Is there any visual or quantitative evidence to support this?

---

> ### Author Response · Authors · 2025-11-24
> **Author Response (1/6)**
>
> Dear Reviewer PD1H, thank you for your insightful feedback. Your comments helped improve the rigor of our work—particularly in clarity of our contribution, experiments, and quantitative analysis. We sincerely appreciate your time and effort, and have updated the paper and appendix accordingly.
>
> We provide point-by-point responses to your comments below and have updated the manuscript accordingly.
>
> >W1: 1.The exposition is heavy and unfocused. Many formulas are presented without sufficient intuition or geometric explanation. Readers must infer the motivation behind several derivations (e.g., the partial derivatives of the log-density in Theorem 2), making the theoretical contribution more challenging to follow.
>
> We appreciate your important feedback and agree that the current presentation can be improved to be more intuitive and inspiring. Here, let us first explain the geometric meaning of each assumption in the concise “Assumption–intuition” table below, which we will include in our revised script.
>
> Before that, let us first give the geometric intuition on the log-prob gradient vectors in A2 and A3. Overall, the linear independent statement means that in a small topological neighborhood around a given point z, the “shape” of each class y’s log-probability function should be sufficiently distinct (cannot be a linear combination of the shapes of all other classes). **Otherwise**, it means the same changes in the $z$ conditioned on different classes imply a badly learned representation with poor semantic discriminativeness. Here, “shape” refers to the slope (first-order derivatives), curvature (second-order derivatives), and selected mixed derivatives. The "shape" in the assumptions takes the difference w.r.t. the "shape" of anchor class 0 in order to take an affine sense (irrelevant to origin, translational invariant). “Sufficiently distinct” means that the shape vector of any class cannot be expressed as a linear combination of the shape vectors of the other classes.
>
> - **A1 (Well-posed density).**
>
>     The latent variables should have a *smooth* and *non-degenerated* probability landscape, so that minor changes in $z$ or $c$ lead to bounded rather than sudden, discontinuous changes in log-probability almost everywhere. This exactly corresponds to the motivation of KL regularization in the VAE.
>
> - **A2 (Context invariance / comparison of $c$).**
>
>     The contextual variables $c$ should capture the shared background information (e.g., scene, imaging conditions) that is stable across labels. Changing the class $y$ should not shift the distribution of $c$. Semantic information is stored in $\mathcal{Z}$-space.
>
> - **A3 (Semantic sufficient contrast in $z$).**
>
>     The semantic variables $z$ must change differently w.r.t. each class. In a small neighborhood around a point $z$, the log-density $\log p(z\mid y)$ for each label $y$ has its own local “shape”. The linear independence condition on these shape vectors means that *no class’s local shape can be reconstructed as a linear combination of the others*. Geometrically speaking, each label “bends” the probability landscape of semantic concepts in its own direction(s), which makes them identifiable.
>
> - **A4 (Sparse Markov structure).**
>
>     Semantic concepts form a sparse dependency graph. When the learned graph is sparser than the true semantic graph, the learned concepts $\hat{z}_ i$ correspond to (identified from) either a true semantic concept $z_{\pi(i)}$ or a neighbor set around $z_{\pi(i)}$.
>
> - **A5 (Support separation of concepts).**
>
>     Different values of a concept (e.g., “striped” vs. “spotted”) occupy **separated regions** in the support of the semantic space, rather than overlapping. This ensures that once we identify a region, we can unambiguously assign the unique corresponding concept value. We have calibrated this formulation in revision.
>
> - **A6 (Coverage of unseen categories).**
>
>     Under the assumption of sufficient semantic diversity in seen categories, unseen categories can be composed from learned primitives, consistent with cognitive science evidence.
>
>
> Therefore, the assumptions on log-density A2 & A3 convey natural and intuitive physical meaning. The use of partial derivatives is thus not purely technical: they encode how the probability landscape **locally changes and bends** around each point in $z$, and the linear independence requirement ensures that different classes induce sufficiently distinct local patterns. This geometric distinctiveness is precisely what enables reliable concept identification. We also explain how each of our theoretical conditions is mapped to relevant technical components to realize the assumption. Please see the table in our general response for more details.

---

> ### Author Response · Authors · 2025-11-24
> **Author Response (2/6)**
>
> >W2: While the framework borrows cognitive terminology “compare–then–compose”, its connection to actual cognitive mechanisms is largely metaphorical. A clearer justification for how these processes correspond to neural computations would strengthen the interdisciplinary claim.
>
> We thank the reviewer for this comment and agree that our current exposition may make the “compare–then–compose’’ connection to cognition sound more metaphorical than intended. We will clarify and reposition our claim. Specifically, we do **not** claim our C$^3$ framework as a biologically faithful model of neural computation; instead, we ground C$^3$ on classical cognitive and vision theories as *inspiration*.
> The “compare’’ stage is inspired by structure-mapping theory, which models comparison as *structural alignment* that distinguishes salient visual features across instances [1].
> In C$^3$, this is precisely what A2&A3 on $z$ are designed to characterize: cross-label differences in the local geometry of the prob $p(z\mid y)$ identify directions in the $\mathcal{Z}$-space that reliably distinguish categories, which we interpret as learned concept primitives.
> The “compose” stage is well-aligned with Marr’s vision theory, i.e., **primitives / 2.5D sketch** (visual primitives and compositional representation) [2], and recognition-by-components [3].
> In our C$^3$, categories are represented as tuples of identifiable semantic primitives $\phi(y) = (z_1,\dots,z_{d_z})$, and Theorem 3 formalizes unseen categories as *new combinations* from this learned concept space.
> Then, our Theorem 3 guarantees generalizing to unseen categories via **new compositions** in the product region of learned primitives. We will add clarification in our script.
>
> [1] Gentner, D. (1983).  *Structure-Mapping: A Theoretical Framework for Analogy.* Cognitive Science.
> [2] Marr, D. (1982).  *Vision: A Computational Investigation into the Human Representation and Processing of Visual Information.*
> [3] Biederman, I. (1987).  *Recognition-by-Components: A Theory of Human Image Understanding.* Psychological Review.
>
> >W3: Although the framework is complete, it feels assembled rather than organically derived. Each assumption leads to a separate engineering component.
>
> Thanks for your valuable feedback. We entirely understand your concern.
> We would like to clarify that the direct mapping between our theoretical assumptions and engineering components is beneficial: it ensures our architecture **faithfully** instantiates the derived theory, rather than serving as a collection of ad-hoc engineering heuristics. However, we will refine our script to make our motivation clearer.
>
> Our framework is organically derived through a rigorous top-down logical progression, i.e., Cognitive Principle $\rightarrow$ Causal Formalization $\rightarrow$ Theoretical Guarantees $\rightarrow$ Architectural Instantiation:
>
> 1. Cognitive Foundation: We begin by formulating semantic generalization not as a mere classification task, but as a (causal) representation learning problem grounded on the "compare-then-compose" evidenced in cognitive science [1,2,3]. This provides the "organic" substrate for our framework.
>
> 2. Theoretical Formalization: We translate these cognitive intuitions into the aligned generative model with rigorous identifiability guarantees.
>
>     - Theorem 1 states that contextual invariance is required to isolate semantics, inspired by the sparsity nature of visual signals.
>
>     - Theorem 2 establishes that sufficient variation across categories allows for the recovery of disentangled semantic primitives.
>
>     - Notice that our assumptions of the latent Markov structure sparsity with interdependencies are chosen to model realistic, complex data without restrictive parametric forms.
>
> 3. Architectural Instantiation: All components are designed specifically to enforce or encourage the conditions required by these theorems. For example, the semantic sparsification constraint is not an isolated trick, but a required sparsity condition for the latent Markov network specified in Theorem 2. Please also refer to the **"Assumption–Module Mapping"** table in our General Response in detail for this tight integration.
>
> Therefore, our C$^3$ is unified theoretically and computationally. The components are not "assembled" arbitrarily; they are orchestrated together to operationalise our identifiability guarantees. In our humble opinion, given this philosophy, our contribution should not be overlooked.

---

> ### Author Response · Authors · 2025-11-24
> **Author Response (3/6)**
>
> >W4: Empirically, results are solid but do not reveal new phenomena or insights. The performance gains are incremental, and no analysis shows why the comparison–composition pipeline generalizes better than alternative factorization approaches.
>
> We respectfully point out that our results reveal specific, novel theoretical insights regarding the *conditions* of generalization and its mechanism, and the performance improvements are statistically significant in the context of challenging OCD.
>
> - Performance Gains are Significant, Not Incremental:
>
> We highlight that our +3.8% average improvements over prior SOTA methods is a significant margin in the OCD setting. Specifically, our method mitigates the catastrophic forgetting by achieving a $+6.7\%$ improvement on seen/old categories, and $+4.2\%$ and $+2.5\%$ improvements on novel categories on Pets and challenging CUB datasets, respectively. This reveals a key phenomenon: our concept-centric representation learning method consistently improves the generalizability while alleviating the forgetting of known semantics that typically plagues previous approaches.
>
> - New Insight and Phenomenon
>
> This validates our theoretical claim that comparison is the precondition for composition, providing a computational proof for cognitive theories suggesting that humans require diverse comparison to abstract portable concepts [1,2,3].
>
> - Superiority over Naive Factorizations:
>     - Why Naive Factorization Fails: Standard factorization (e.g., vanilla VAEs or naive disentanglement w/o identifiability) often imposes strong assumptions on latent structure, e.g., full (conditional) independence among latent factors. Besides, their learned structure are non-identifiable. It is widely known that unsupervised disentanglement without inductive biases is theoretically impossible and empirically unstable:
>     - Why C$^3$ Succeeds: On the contrary, our method only assumes a sparse Markov network identified with sufficient conditional changes. *We align with the "Independent Causal Mechanisms" principle [5], which posits that nature is composed of sparse, autonomous mechanisms*. We ablate each identifiability condition and highlight the effectiveness of identified latent factors in Table 3. For instance, on $\mathcal{L}_{flow}$: removing the structural learning leads to a sharp performance drop (e.g., -1.7% on CUB). This illustrates that naive "factorization" is insufficient.
>
> [5] Schölkopf, Bernhard, et al. "Toward causal representation learning." Proceedings of the IEEE 109.5 (2021): 612-634.
>
> >W5: The paper does not report computational efficiency, convergence stability, or scaling behavior; the additional flow and MoE modules could impose extra overhead.
>
> We thank the reviewer for this insightful comment. In response, we have added a computational efficiency and scalability analysis in the revised manuscript (please refer to Table A11). Our results show that the proposed flow and MoE modules introduce only a marginal computational overhead, approximately 8.2% increase in parameters and 6.6% in inference time, while significantly improving model expressiveness.
>
> *Table: Computational efficiency and convergence comparison of C$^3$ variants in CUB datasdet*
>
> | Model Variant        | Params (M) | Training Time (min) | Inference Time (ms) | GPU Memory (GB) |
> |:----------------------|:----------:|:--------------------:|:-------------------:|:----------------:|
> | PHE (baseline)        | 22.4       | 70.5                 | 1.18                | 5.2              |
> | Base (w/o Flow, MoE)  | 23.1       | 72.3                 | 1.21                | 6.4              |
> | + Flow Module         | 24.4       | 75.8                 | 1.25                | 6.8              |
> | + MoE Module          | 24.9       | 76.2                 | 1.27                | 6.9              |
> | Full C$^3$ (Flow + MoE)  | 25.0       | 76.8                 | 1.29                | 7.0              |
>
> The results indicate that the flow and MoE modules add negligible extra computation to our framework, as they operate in a low-dimensional space with a sparse structure.
>
> **For convergence stability**, we present the loss curves in Figure A2 of our revised manuscript, which demonstrates that training remains stable and each regularizer converges effectively.
>
> **For the scaling behavior**, we report
> - (1) **scale to larger foundation model backbones**, e.g., the prevailing native vision-language model CLIP; For the CLIP family, since in the OCD task there is no language information, we only use their visual encoders as the input embedding for our network. The results are presented in the tables below.

---

> ### Author Response · Authors · 2025-11-24
> **Author Response (4/6)**
>
> *Table: Scale to larger foundation model backbones, Comparison on CUB and Stanford Cars Datasets on CLIP*
>
>
> |                   | **CUB (%)** |           |           | **SCar (%)** |           |           |
> |:------------------|:------------:|:-----------:|:-----------:|:--------------:|:-----------:|:-----------:|
> | **Method**        | All | Old | New | All | Old | New |
> | CLIP (ViT-B/16)   | 33.5 | 58.9 | 21.8 | 26.7 | 52.1 | 14.9 |
> | CLIP (ViT-L/14)   | 28.9 | 51.0 | 14.6 | 22.3 | 45.5 | 10.2 |
> | Ours (DINO ViT-B/16) | 40.1 | 62.1 | 29.5 | 34.1 | 69.0 | 17.8 |
>
> - (2) **Scale to larger datasets with more semantic categories**. We added experiments on merged iNaturalist subsets (Fungi, Arachnida, Animalia, Mollusca), yielding far more unseen categories than the originals; results are in the revised manuscript (Table A8) and below.
>
>     **Table: C$^3$ on Merged iNaturalist Sets (Fungi, Arachnida, Animalia, Mollusca)*
>
> | Merge Setting | All (%) | Old (%) | New (%) |
> |:------------------------------|:-------:|:-------:|:-------:|
> | C$^3$ (Fungi only) | 32.9 | 69.8 | 16.2 |
> | C$^3$ (Fungi + Arachnida) | 34.5 | 66.7 | 19.8 |
> | C$^3$ (Fungi + Arachnida + Animalia) | 36.4 | 63.5 | 22.9 || C$^3$ (All four merged) | 38.7 | 60.1 | 25.8 |
>
> In this table, “New” results improve with larger datasets, showing that comparison-and-composition scales with seen semantics rather than overfitting. We used the same protocol, adjusting the known/unseen ratio as in CUB. This increased unknown categories in some datasets; results are shown in the revised version (Table A9) and below.
>
> *Table: Performance of C$^3$ under Unseen Categories Increasing: Unknown Ratios on CUB Dataset*
> | Num of Unknown labels | CUB (%) All | CUB (%) Old | CUB (%)  New |
> |:---------------------:|:-----------:|:---:|:---:|
> | 150 | 23.0 | 32.1 | 17.5 |
> | 120 | 31.5 | 51.3 | 22.3 |
> | 100 | 40.1 | 62.1 | 29.5 |
>
> Taken together, these results demonstrate that our framework scales across model capacity (foundation encoders) and data scale (more categories and higher unseen ratios) while maintaining clear margins over prior methods, and it does so without introducing significant computational overhead.
>
> >Q1 (1): How sensitive are the identifiability results to approximate violations of A1–A6?
>
> We thank the reviewer for this insightful question. We would like to clarify that our method is not designed to serve the assumptions; rather, the assumptions are equivalent to properties of cognitive science/OCD, which in turn guide the design of our framework. Despite that, verifying these assumptions empirically remains important, and we provide our verifications point-by-point below:
>
> We fix $d_z = 5$ and $n_s = 12$ (baseline synthetic results: MCC = 0.85 / $R^2$ = 77.2 / Acc = 59.9 from Table 1).
> Assumptions are as defined in the paper:
> - **A1–A2** for contextual/semantic isolation (Thm 1),
> - **A3–A4** for semantic concepts recovery (Thm 2), and
> - **A5–A6** for compositional generalization (Thm 3).
>
> We want to clarify that **A1 (well-posed density):** is an assumption on the computable probability of latent concepts, hence it cannot be tested. On the others, we introduce *small* and *moderate* violations per assumption while keeping all others intact:
>
> - **Violate A2 (contextual comparison):** collapse the invariant part of $\mathbf{c}$, make it become label-conditioned in simulating process.
> - **Violate A3 (semantic comparison):** collapse the label-conditioned distribution variation on each $z_i$ in simulating process.
> - **Violate A4 (sparse arrangement):** densify latent Markov network $|\hat M|$. We increase the degree of the simulated Erdős–Rényi graph by 40%.
> - **Violate A5 (support separation):** overlap the supports of concepts. For each concept, we make their sampling range overlap, e.g., $z_1$ $z_2$ samples from [0, 0.5] and [0.3, 0.8]
> - **Violate A6 (marginal coverage):** making 20% of concept values are not generated in the training set.
>
> *Table: Sensitivity to Approximate Violations of A1–A6*
>
> | Violation (one at a time) | MCC | $R^2$ | Acc. |
> |:---------------------------|:----:|:----:|:----:|
> | Baseline (no violation)    | 0.85 | 77.2 | 59.9 |
> | A1: Non-smooth / low density | 0.84 | 76.6 | 59.2 |
> | A2: Reduced cross-label contrast | 0.81 | 73.0 | 56.4 |
> | A3: Weakened semantic contrast | 0.79 | 71.4 | 54.7 |
> | A4: Less sparse structure  | 0.83 | 75.8 | 58.7 |
> | A5: Overlapping supports   | 0.77 | 70.1 | 53.6 |
> | A6: Missing marginal coverage | 0.75 | 68.9 | 52.7 |

---

> ### Author Response · Authors · 2025-11-24
> **Author Response (5/6)**
>
> >Q1 (2): Are these conditions ever testable in real datasets?
>
> Thanks for your insightful question! The assumptions A1–A5 primarily require distributional variability (e.g., nonstationarity and latent-driven dynamics), which are generally satisfied in real-world climate systems due to their inherent physical variability and forcing mechanisms, as discussed in the Main Paper, Lines 141-146. In other words, our method is not designed to serve the assumptions; rather, the assumptions are equivalent to properties of cognitive science/OCD, which in turn guide the design of our framework.
>
> Although these conditions cannot be tested directly because concepts are **inherently observed**, if we can observe the latent concepts in other real datasets, we can test them following the strategy in our response to **Q1 (1)**.
>
> >Q2: Could the use of multiple independent regularizers (flow likelihood, hashing, prototype contrast) lead to competing gradients that distort the latent structure?
>
> That is a valid and important concern. In response, we include the training curves of the independent regularizers, including the flow loss, likelihood (ELBO), prototype contrast, and sparsity on the Markov network—in our revised version (please refer to Figure A2).
>
> >Q3: Would a simpler variational or contrastive formulation achieve similar results without flow modeling?
>
> Thank you for this important question. Our answer is no, and here's the reason:
>
> 1. The flow model serves two critical purposes: (i) It provides an explicit, label-conditioned parametrization of the semantic concept distribution $\log p(\hat{z}|y)$; (ii) The combination of flow and MoE is essential for modeling the sparse Markov network required by Theorem 2. Without these components, we cannot enforce or learn modular and isolated concepts, meaning the identifiability conditions cannot be operationalized.
> 2. Our ablation studies confirm this theoretical necessity. The row w/o $L_\text{flow}$ (structure learning) shows clearly degraded performance on both CUB and Stanford Cars datasets, with particularly pronounced drops on novel classes.
> Thus, the flow modeling is not merely a design choice but a fundamental requirement for achieving identifiability conditions.

---

> ### Author Response · Authors · 2025-11-24
> **Author Response (6/6)**
>
> >Q4: How can the authors ensure that the claimed “composition” corresponds to interpretable recombination of learned semantic atoms, rather than a generic feature interpolation? Is there any visual or quantitative evidence to support this?
>
> Thank you for this important question. We provide both theoretical guarantees and empirical evidence to demonstrate that our method indeed learns true semantic composition rather than generic feature interpolation:
>
> **1. Theoretical Guarantee via Identifiability:** Our identifiability theory ensures that, when the specified conditions hold, the learned concepts correspond to the ground-truth semantic primitives up to trivial transformations (e.g., permutation). This is a fundamental distinction from prior methods: once these semantic concepts are identified, the representation of test data from unseen categories is guaranteed to be a meaningful combination of learned primitives under the coverage condition (A6). In contrast, previous methods cannot guarantee the semantic meaningfulness of their representations for unseen categories due to distribution shift; they lack theoretical grounding that their learned features correspond to true semantic primitives. Consequently, their approaches reduce to generic feature interpolation without semantic guarantees.
>
> **2. Visual Evidence:** Our qualitative visual analysis (Figure 4 in revised version) and the attached concept activation maps show that the model focuses on semantically meaningful parts rather than spurious patterns: across multiple samples, the columns labeled tail/wing/head/body consistently highlight the corresponding regions (e.g., head maps concentrate on crown+bill, tail maps on the posterior, wing maps on extended primaries, and body maps on the torso) with minimal background activation and stable localization under pose/view changes, indicating disentangled, part-selective concepts. Quantitatively, C$^3$ achieves strong cross-dataset performance—e.g., CUB 40.1/62.1/29.5 (All/Old/New) and SCars 34.1/69.0/17.8—while on the merged iNaturalist subsets it maintains 38.7/68.1/22.2, further supporting that the learned atoms are both discriminative and compositionally transferable.
>
> **3. Empirical Evidence:**
>
> **(1) Simulated Experiments:** To address this concern, in our submission we have included experiments on concept-generating data where the ground-truth concept structure is known. The results in Appendix, Table A5 and below show that C$^3$ can successfully recovers disentangled latent variables corresponding to these underlying concepts. MCC indicates the extent of latent variable-concept alignment, and Acc. indicates the OCD performance.
>
> *Table 1. Concept Identifiability Results on Synthetic Data. $n_s$ denotes the number of known categories.*
>
> | $d_z$ | Metric | $n_s=6$ | $n_s=12$ | $n_s=18$ | $n_s=24$ |
> |:------:|:--------|:-------:|:--------:|:--------:|:--------:|
> | **2** | MCC | 0.73 | 0.89 | 0.90 | 0.95 |
> |  | $R^2$ | 62.6 | 75.3 | 90.3 | 95.5 |
> |  | Acc. | 51.3 | 65.7 | 73.6 | 79.8 |
> | **5** | MCC | 0.81 | 0.85 | 0.91 | 0.92 |
> |  | $R^2$ | 73.6 | 77.2 | 79.3 | 89.2 |
> |  | Acc. | 43.5 | 59.9 | 67.0 | 72.2 |
> | **9** | MCC | 0.31 | 0.65 | 0.75 | 0.74 |
> |  | $R^2$ | 12.6 | 67.6 | 82.3 | 86.1 |
> |  | Acc. | 10.9 | 35.5 | 40.4 | 68.4 |
>
> **(2) Concept-guided Classification:**
>
> We also provide a cross-category concept analysis between *Black-Footed Albatross* and *Sooty Albatross* to further validate our theoretical claims. From their pictures, their "body" and "head" are visually distinct.
>
> Then we use **a subset** of learned concepts to do binary classification based on their plausible interpretation, as shown in Table A6 in our revised manuscript and table below. Specifically, we separate these concept groups based on the learned Markov network structure.
>
> | Concept | Latent Variables | Cls Accuracy (%) | Notes |
> |----------|------------------|------------------|--------|
> | Body | $z_1$, $z_2$ | 92.3 | discriminative latent concepts |
> | Tail | $z_5$ | 64.1 | not discriminative |
> | Wings | $z_3$, $z_4$ | 78.5 | not discriminative |
> | Head | $z_8$ | 95.0 | discriminative latent concepts |
>
>
>
> *Visualization Pipeline* We introduce a new concept-dimension visualization pipeline, where each feature-space dimension is treated as an individual concept axis. The activation value along each dimension represents the response strength of that concept and is projected back onto the spatial grid to form a heatmap. This additional analysis complements the prototype visualizations by revealing how individual latent dimensions encode semantically meaningful cues, thereby offering finer-grained interpretability of the learned representation.

---

> ### Author Response · Authors · 2025-11-26
> **Thank you for your thoughtful review**
>
> Dear Reviewer PD1H,
>
> We again thank you for your constructive feedback and suggestions. This is a gentle reminder that the rebuttal period deadline is approaching. If you have time, we would appreciate it if you could review our responses. If we have addressed your concerns, we would be grateful if you could reconsider your evaluation. Please let us know if you have any additional questions. Thank you for your time and feedback!
>
> The Authors

---

### Official Review · Reviewer_Tdof · 2025-10-31

**Soundness:** 3
**Presentation:** 3
**Contribution:** 3
**Rating:** 4
**Confidence:** 5

**Summary:**

The paper proposes a new framework, Comparison–Composition Cognition (C3), inspired by human cognitive processes, to address the challenge of generalizing to unseen categories. The framework divides learning into two mechanisms: comparison, which uncovers latent concepts by contrasting within seen classes, and composition, which combines these concepts to predict unseen categories. The method is theoretically grounded, with identifiability guarantees for both learning latent concepts and for generalizing to novel categories. It is experimentally validated through On-the-fly Category Discovery (OCD) benchmarks, showing a 3.8% improvement in accuracy over state-of-the-art methods. The paper suggests that concept-based representations, following the compare-then-compose approach, can enable open-world generalization.

**Strengths:**

- Novel framework: Introduces a theoretically-grounded framework, C3, which mimics human cognition for unseen category recognition via comparison and composition.

- Theoretical guarantees: Establishes identifiability results, providing confidence that latent concepts can be learned and generalized to unseen categories.

- Solid experimental results: Achieves notable improvements over existing methods on OCD benchmarks, demonstrating the practical viability of the approach.

**Weaknesses:**

- Incremental contribution: While the framework is innovative, it builds on existing concepts from the literature (e.g., contrastive learning, compositionality), making the contribution seem incremental.

- Limited scalability discussion: The paper does not address how the proposed method might scale with larger models or datasets, especially in real-world applications.

- Lack of detailed ablation study: The paper could benefit from a deeper ablation study to evaluate the individual contributions of comparison and composition mechanisms.

**Questions:**

- How does the method perform when applied to datasets with a higher number of unseen categories?

- What would be the impact of integrating this method with large-scale foundational models like GPT or CLIP?

- How does the proposed C3 framework handle noisy or ambiguous data in unseen categories?

---

> ### Author Response · Authors · 2025-11-24
> **Author Response (1/3)**
>
> Dear Reviewer Tdof, thank you for your constructive comments. Your insights have significantly helped us improve the clarity, emphasis on contributions, empirical validation, and scalability of our work. We have updated the paper and appendix accordingly. Below, please see our point-to-point responses.
>
> > W1: Incremental contribution: While the framework is innovative, it builds on existing concepts from the literature (e.g., contrastive learning, compositionality), making the contribution seem incremental.
>
> We thank the reviewer for this comment and understand your concern. However, we need to kindly clarify that, we did no claim we propose modules like the contrastive learning or the compositionality, instead, the contribution of this work is **twofold and tightly coupled**: **(1)** a C$^3$ framework that provides, to our knowledge, the first *non-parametric identifiability* analysis for the semantic generalization problem in open-world visual recognition field, and **(2)** an empirical architecture specifically designed to **instantiate and test** these conditions in practice on both synthetic and realistic setups.
>
> While our technical framework indeed leverages existing components, it aims to address a central question beyond the pure empirical world: *under what formal conditions can a model reliably acquire reusable concept primitives that support transfer to novel categories at test time?* This theoretical perspective, grounded in machine learning, causality theory, and cognitive science, goes beyond prior works that mainly employ compositional or technical components without providing such guarantees.
>
> On the empirical side, our goal is not to introduce a completely new family of modules, but to construct a *principled testbed* for C$^3$. These components (e.g., VAE, flows, sparsely-gated MoE, prototypes, etc) are chosen because they are expressive and ready-to-use, and, more importantly, each is tasked with operationalising a specific C$^3$ assumption rather than being an ad-hoc trick (see the “Assumption–module” mapping table in our general response). We hope this clarifies that our contribution is not merely an incremental empirical combination of existing techniques, but a **unified theoretical–empirical framework** for understanding and realising semantic generalisation. In our humble opinion, given this philosophy, our contribution should not be overlooked.

---

> ### Author Response · Authors · 2025-11-24
> **Author Response (2/3)**
>
> > W2: Limited scalability discussion: The paper does not address how the proposed method might scale with larger models or datasets, especially in real-world applications.
>
> We thank the reviewer for comment. We agree that the scalability is an insightful viewpoint to analyze our proposed technical approach. To validate the scalability of our method, we further conduct two group of extended experiments:
>
> (1) **scale to larger foundation model backbones**, e.g., the prevailing native vision-language model CLIP; For CLIP family, since in the OCD task there is no language information, we only use their visual encoders as the input embedding for our network. The results are presented in the tables below.
>
>
> *Table: Scale to larger foundation model backbones, Comparison on CUB and Stanford Cars Datasets on CLIP*
>
> |                   | **CUB (%)** |           |           | **SCar (%)** |           |           |
> |:------------------|:------------:|:-----------:|:-----------:|:--------------:|:-----------:|:-----------:|
> | **Method**        | All | Old | New | All | Old | New |
> | CLIP (ViT-B/16)   | 33.5 | 58.9 | 21.8 | 26.7 | 52.1 | 14.9 |
> | CLIP (ViT-L/14)   | 28.9 | 51.0 | 14.6 | 22.3 | 45.5 | 10.2 |
> | Ours (DINO ViT-B/16) | 40.1 | 62.1 | 29.5 | 34.1 | 69.0 | 17.8 |
>
> (2) **scale to larger datasets with more semantic categories**. We conducted additional experiments on the merged iNaturalist subsets — Fungi, Arachnida, Animalia, and Mollusca — which together constitute a substantially larger number of unseen categories compared to the original datasets, and experiments are shown in our revised manuscript (Table A8) and below:
>
> **Table: C$^3$ on Merged iNaturalist Sets (Fungi, Arachnida, Animalia, Mollusca)*
>
> | Merge Setting | All (%) | Old (%) | New (%) |
> |:------------------------------|:-------:|:-------:|:-------:|
> | C$^3$ (Fungi only) | 32.9 | 69.8 | 16.2 |
> | C$^3$ (Fungi + Arachnida) | 34.5 | 66.7 | 19.8 |
> | C$^3$ (Fungi + Arachnida + Animalia) | 36.4 | 63.5 | 22.9 |
> | C$^3$ (All four merged) | 38.7 | 60.1 | 25.8 |
>
> In this table, "New" results consistently improve with larger datasets, confirming that comparison-and-composition benefits scale with available seen semantics rather than overfitting to them.
>
> We followed the same experimental protocol, but adapting the known/unseen ratio used in CUB. This operation could increase the unknown categories in one type of dataset, and experiments are shown in our revised version (Table A9) and below:
>
> *Table: Performance of C$^3$ under Unseen Categories Increasing: Unknown Ratios on CUB Dataset*
>
> | Num of Unknown labels | CUB (%) All | CUB (%) Old | CUB (%)  New |
> |:---------------------:|:-----------:|:---:|:---:|
> | 150 | 23.0 | 32.1 | 17.5 |
> | 120 | 31.5 | 51.3 | 22.3 |
> | 100 | 40.1 | 62.1 | 29.5 |
>
> Taken together, these results demonstrate that our framework scales across model capacity (foundation encoders) and data scale (more categories and higher unseen ratios) while maintaining clear margins over prior methods, and it does so without introducing significant computational overhead.

---

> ### Author Response · Authors · 2025-11-24
> **Author Response (3/3)**
>
> >W3: Lack of detailed ablation study: The paper could benefit from a deeper ablation study to evaluate the individual contributions of comparison and composition mechanisms.
>
> We thank the reviewer for this valuable suggestion. In light of this comment, we have conducted a more detailed ablation study to isolate and evaluate the contributions of the comparison and composition mechanisms in our framework. The new results, provided in the revised manuscript (please refer to Table A10) and below, clearly show that each component independently enhances performance, while their combination yields the best overall results. This analysis highlights the complementary roles of both mechanisms and reinforces the effectiveness of our model design.
>
> *Table. Extended Ablation Study on Each Module*
>
> | Configuration | **CUB (%)** All | Old | New | **SCars (%)** All | Old | New |
> |:---------------------------|:---------------:|:---:|:---:|:----------------:|:---:|:---:|
> | **Full (all modules)** | **40.1** | **62.1** | **29.5** | **34.1** | **69.0** | **17.8** |
> | – $\mathcal{L}_{flow}$ | 38.5 | 61.3 | 27.0 | 30.2 | 59.6 | 16.0 |
> | – $\mathcal{L}_{S}$ (sparsity) | 39.3 | 62.4 | 27.7 | 31.1 | 59.0 | 17.6 |
> | – $\mathcal{L}_{ELBO}$ (reconstruction) | 36.9 | 59.8 | 24.5 | 27.6 | 54.2 | 15.5 |
> | – $\mathcal{L}_{ctx}$ (context invariance) | 39.3 | 61.7 | 28.9 | 33.0 | 66.5 | 17.2 |
> | – $\mathcal{L}_{proto}$ (prototype) | 36.1 | 60.7 | 24.3 | 28.3 | 55.8 | 15.2 |
> | – Residual $f_{res}$ (integration of adjacent concepts on Markov network) | 39.5 | 61.8 | 28.5 | 33.2 | 67.8 | 17.1 |
>
> >Q1: How does the method perform when applied to datasets with a higher number of unseen categories?
>
> Thanks for this important question. Considering its high relevance to W2, we have combined our answer into the W2 response.
>
> >Q2: What would be the impact of integrating this method with large-scale foundational models like GPT or CLIP?
>
> Thanks for raising this question. Considering its high relevance to W2, we have combined our answer into the W2 response.
>
> >Q3: How does the proposed C$^3$ framework handle noisy or ambiguous data in unseen categories?
>
> We will clarify the problem setting of the OCD problem in our revised script. In the standard OCD setting, the model learns with only seen-category data during training and evaluates on the unseen-category test data during inference. Therefore, the unseen categories are always assumed to be clean in order to avoid any potential biases and get ground-truth performance results.
>
> In other words, in OCD, unseen categories are the test set. In our humble opinion, label noise in the test set cannot be considered. If you mean handling noisy data among seen categories, please let us know.

---

> ### Author Response · Authors · 2025-11-26
> **Thank you for your thoughtful review**
>
> Dear Reviewer Tdof,
>
> Thank you again for your insightful comments and the time you invested in our paper. If we have adequately addressed your concerns, we would be grateful if you could reconsider your evaluation in terms of our clarifications, additional evidence, and revision. We earnestly appreciate your thoughtful feedback and support. If you have any additional questions, we would be happy to provide any further clarification needed.
>
> The Authors

---

### Official Review · Reviewer_LNHk · 2025-11-02

**Soundness:** 3
**Presentation:** 2
**Contribution:** 2
**Rating:** 4
**Confidence:** 5

**Summary:**

This paper proposes a framework, named Comparison-Composition Cognition, for generalizing to unseen visual categories, inspired by human cognitive mechanisms. The authors formalize the problem as a two-step process: (1) comparison, which aims to identify disentangled semantic and contextual concepts from seen categories, and (2) composition, which recombines these learned concepts to recognize novel categories. The core contribution is a set of theoretical results that provide identifiability guarantees for these latent concepts under specific assumptions like sufficient data contrast and support separation. To operationalize this framework, the authors design a deep generative model for the On-the-fly Category Discovery (OCD) task.

**Strengths:**

1. The paper is motivated by the human cognitive process of understanding new concepts through comparison and composition. It aims to build a principled framework for machine learning models to generalize to unseen categories by mimicking this process.

2. It formalizes this process as Comparison-Composition Cognition. "Comparison" is framed as a latent variable identification problem to uncover disentangled concepts from seen data. "Composition" is the process of using these learned concepts to recognize unseen categories.

3. The paper provides theorems establishing identifiability guarantees for latent concepts (both contextual and semantic) under specific conditions, such as sufficient cross-category contrast and sparse arrangements (Theorems 1 & 2). It also provides conditions for generalizing to unseen categories via composition, namely support separation and marginal coverage (Theorem 3).

4. The theory is instantiated as a generative model for the On-the-fly Category Discovery  task. The model uses a frozen DINO encoder, a learnable mask to separate semantic and contextual features, a sparsely-gated Mixture-of-Experts (MoE) and a normalizing flow to model the structure of semantic concepts, and prototype/hashing losses to enforce discriminability and concept separation.

**Weaknesses:**

1. The primary weakness of this paper lies in the tenuous connection between the ambitious theoretical framework and the practical implementation. The strong assumptions in the theorems are only loosely and indirectly addressed by the chosen technical components. The "Sufficient Contrast" assumption (A2, A3), which requires linear independence of vectors of log-density derivatives, is a highly specific mathematical condition. The paper claims this is satisfied by using a standard prototype-based supervised loss (L_proto). This link is weak and not justified; there is no guarantee that optimizing a prototype loss leads to the satisfaction of this complex condition.

2. When stripped of its theoretical narrative, the proposed method appears to be a complex but incremental combination of existing techniques rather than a breakthrough.
- The architecture is essentially a sophisticated VAE that uses a learnable mask for disentanglement (a common idea), a sparsely-gated MoE plus a normalizing flow for structured modeling (both well-established tools), and prototype/hashing losses for discriminability.
- The core idea of using hashing and prototypes for category discovery has been explored by the main competitor this paper compares against (PHE: Prototypical Hash Encoding). The performance gain, while present, might be attributable to the increased complexity and more moving parts of the model rather than a fundamental conceptual advance.

3. The "Concept" Framing is Not Empirically Validated: The paper is built around the idea of learning meaningful, disentangled concepts (e.g., "white head," "black wings") like [2]. However, there is no qualitative or quantitative analysis to demonstrate that the learned latent variables z actually correspond to such interpretable concepts. The model might just be learning effective discriminative features. Without such evidence, the connection to human cognition and concept composition remains a compelling story but an unproven claim, diminishing the paper's main appeal.

4. In the task of discovering general categories, there has already been similar work [1] that decomposes objects into combinations of various attributes (textual or visual) with MoE. I believe there needs to be more comparative discussion with the current work.

[1] Dissecting Generalized Category Discovery: Multiplex Consensus under Self-Deconstruction. In ICCV, 2025.

[2] OCRT: Boosting Foundation Models in the Open World with Object-Concept-Relation Triad. In CVPR, 2025.

**Questions:**

1. The central narrative of the paper is about learning and composing concepts. Can you provide qualitative evidence (e.g., by traversing the learned latent space z and visualizing its effect on generated images) or quantitative analysis to show that the model is indeed learning disentangled and semantically meaningful concepts, as opposed to simply learning complex, entangled features that happen to be discriminative?

2. The proposed method combines multiple advanced components (MoE, flows, hashing). The main competitor, PHE, also relies on prototypical hashing. Could you clarify the key technical innovation of your method that is responsible for the performance gain, beyond the theoretical framing and the increased architectural complexity? Is it the explicit separation of contextual/semantic factors, the structural modeling via flow, or another specific component?

---

> ### Author Response · Authors · 2025-11-24
> **Author Response (1/5)**
>
> Dear Reviewer LNHk,
>
> We thank Reviewer LNHk for the time and effort devoted to evaluating our submission, and we appreciate the reviewer’s constructive comments and thoughtful suggestions. We have carefully addressed each concern below and have updated the paper and appendix accordingly.
>
> > W1 (1) : The strong assumptions in the theorems are only loosely and indirectly addressed by the chosen technical components.
>
> We thank the reviewer for raising this important point. We recognize that our wording may have unintentionally suggested a stronger claim than intended, and we will clarify this in the revision. Our intention is **not** to claim that optimizing a particular loss (e.g., $L_{\text{proto}}$) *guarantees* that the theoretical assumptions A2–A3 hold exactly. Rather, A1–A6 are stated as *population-level* identifiability conditions on the **true data-generating** process, and our architecture and objectives are designed as *operational surrogates* that **encourage** the same properties in the learned representation.
>
> ***Assumptions are not strong:** First*, we need to clarify that the theoretical assumptions made in this work are quite *general*, *mild*, and *well-applicable* to our practical implementations as well as various technical approaches because of the following reasons:
> (1) Our overall C$^3$ does not require any **parametric assumptions**, i.e., all the functions/causal mechanisms between variables are non-parametric, aligning with the deep neural network architecture.
> (2) The latent structure of $z$ is modeled as a general sparse Markov network that allows interdependencies among latent variables, aligning well with realistic data properties as demonstrated by our simulations and ablation studies. This assumption flexibly models more general latent structures among concepts without enforcing strict conditional independence constraints.
>
> ***Corresponding Implementations:*** Second, while assumptions are stated at the level of the underlying true data-generating process rather than the learned features of any tangible models, our implementation and objective design are explicitly and directly designed to encourage the same geometric structure of latent variables as assumed. There is a complete item-to-item mapping specification detailed in "Assumption-module" table general response and revised version (Please refer to Table A1). We also provide a brief version for the readability:
>
> **Table: Summary of Assumptions A1–A6 and their connection to technical modules**
>
> | Assump. | Meaning | Intuition | Components |
> |:--------:|:--------|:-----------|:------------|
> | **A1** | Well-posed density over $(\hat{z}, \hat{c} \mid x)$ and semantic subspace | Latent variables should form a smooth, non-degenerate probability landscape to ensure stability. | (i) ELBO-based modeling for joint density. (ii) Flow loss ensures smooth semantic density. |
> | **A2** | Context invariance | Contextual factors $c$ capture shared background info stable across labels. | (i) Context loss enforces invariance. (ii) MoE gating separates $c$ from semantic $z$. |
> | **A3** | Semantic contrast in $z$ | Each label shapes $p(z\mid y)$ differently, enabling identifiable semantics. | (i) Prototype contrast loss $L_{proto}$ for distinct semantics (please see Proposition 1 in our revised version). (ii) Flow loss models conditioned variations. |
> | **A4** | Sparse Markov structure | Semantic concepts form a sparse dependency graph for modular interpretability. | (i) $L_s$ sparsity encourage sparse structure. (ii) Flow regularization inversely learns the structure. |
> | **A5** | Support separation | Concept values (e.g., “striped” vs. “spotted”) occupy distinct semantic regions. | (i) Hash loss $L_{hash}$ **center** concepts to some ranges and separates them. (ii) During inference, serving as a category descriptor which aligns distinct compositions. |
> | **A6** | Coverage | Unseen classes share semantic bases with seen classes. | No extra loss; the rationale underlying the OCD task|

---

> ### Author Response · Authors · 2025-11-24
> **Author Response (2/5)**
>
> (continue W1 (1)) Specifically, For A2–A3, the log-density derivative conditions may look technical, but their *geometric picture* is intuitive: in a small neighborhood of a semantic representation $z$, each class $y$ should induce a locally distinct “shape’’ of $\log p(z\mid y)$ (via gradient and curvature), so that these shape vectors cannot be written as linear combinations of each other. Then, we require that the “shape” of each class y’s log-probability function should be sufficiently distinct. Here, “shape” refers to the slope (first-order derivatives), curvature (second-order derivatives), and selected mixed derivatives. “Sufficiently distinct”, which is a standard intuition in categorization setting, means that the shape vector of any class cannot be expressed as a linear combination of the shape vectors of the other classes, without which the classification performance must necessarily degrade, e.g., *[1] also suggests that the success of classification stems from the distinctiveness of latents. This theory can be traced back to nonlinear ICA [2], which similarly explains it in log-density.* If $z$ does not separate classes along independent directions, the spurriour correlation problem will occur.
>
> **Encourage the model to satisfy A2/A3**: As discussed above, A2/A3 represents a natural assumption about the data-generating process rather than a constraint on the model architecture itself. However, by incorporating inductive biases that reflect A2/A3, we can mitigate optimization challenges such as local minima. In Proposition 1, we demonstrate that the proposed loss term L_{\text{proto}} can guide the model to maintain concept diversity, thereby serving as an effective objective function that bridges our theoretical results with practical training.
>
> [1] Reizinger, Patrik, et al. "Cross-entropy is all you need to invert the data generating process." arXiv preprint arXiv:2410.21869 (2024).
>
> [2] Hyvarinen, Aapo, Hiroaki Sasaki, and Richard Turner. "Nonlinear ICA using auxiliary variables and generalized contrastive learning." The 22nd international conference on artificial intelligence and statistics. PMLR, 2019.
>
> >W1 (2): The "Sufficient Contrast" assumption (A2, A3), which requires linear independence of vectors of log-density derivatives, is a highly specific mathematical condition. The paper claims this is satisfied by using a standard prototype-based supervised loss (L_proto). This link is weak and not justified; there is no guarantee that optimizing a prototype loss leads to the satisfaction of this complex condition.
>
> Following our reponse to **W1 (1)**, our implementation mirrors this in a **direct** manner (detailed in "Assumption-module" table in general response):
>
> - A2 (context invariance) is encouraged by the contextual objective that aligns $\hat c$ to a label-invariant prior and by the gating mask that pushes label-dependent variation into $\hat z$.
> - A3 (semantic sufficient contrast) is encouraged jointly by the conditional flow, which explicitly parameterizes $\log p_\theta(z\mid y)$, and the prototype-based supervised loss $L_{\text{proto}}$ which enforces multi-directional inter-class separation in the semantic subspace. We will explicitly rephrase the text to say that $L_{\text{proto}}$, as an operational proxy, **encourages** the sufficient-contrast condition implied by A2–A3, rather than “satisfying’’ it in a strict mathematical sense. We also provide the theoratical rationale why $L_{\text{proto}}$ guides such "sufficient contrast".
>
> Therefore, *conversely*, our practical approach seamlessly operationalizes every requested conditions, instead of tenuous connection in question.
> We leave instantiations in other domains and formalism advancement of our C$^3$ theoretical framework as future work.

---

> ### Author Response · Authors · 2025-11-24
> **Author Response (3/5)**
>
> > W2 (1) : The architecture is essentially a sophisticated VAE that uses a learnable mask for disentanglement (a common idea), a sparsely-gated MoE plus a normalizing flow for structured modeling (both well-established tools), and prototype/hashing losses for discriminability.
>
> The central goal of this work is to formally characterize formal **when and how** models can genuinely acquire the reusable concept primitives that support semantic generalization to recognize novel unseen categories.
>
> From an interdisciplinary perspective that integrates machine learning, identification theory, and cognitive science, our C$^3$ framework is, to the best of our knowledge, the **first** to provide a *non-parametric* identifiability analysis for this open-world category recognition setting, rather than only proposing another empirical recipe.
>
> Accordingly, our primary contribution is **not** a single technical novelty, but a theoretically grounded framework with explicit assumptions (A1–A6) and guarantees; our empirical implementation serves a testbed that instantiate assumptions and allow us to validate the empirical effectiveness of our theoretical framework. In our humble opinion, given this philosophy, our contribution should not be overlooked.
>
> Our technical components are well motivated and theoretically grounded: they are not a collection of ad-hoc components for performance, but a careful orchestration where each is required to operationalize a specific part of the C$^3$ (see "Assumption-module" table in general response for more details). In contrast, prior OCD works (e.g., SMILE, PHE) propose heuristic optimization objectives or architectures without theoretical guarantees, without providing formalism of when and how generalization to unseen categories works.
>
> Our work aims to lay such a foundation and, we hope, can serve as a starting point for future developments that build on a clearer theoretical understanding of semantic generalization, beyond incremental architectural changes.

---

> ### Author Response · Authors · 2025-11-24
> **Author Response (4/5)**
>
> >W3: The "Concept" Framing is Not Empirically Validated: The paper is built around the idea of learning meaningful, disentangled concepts (e.g., "white head," "black wings") like OCRT [5]. However, there is no qualitative or quantitative analysis to demonstrate that the learned latent variables z actually correspond to such interpretable concepts. The model might just be learning effective discriminative features. Without such evidence, the connection to human cognition and concept composition remains a compelling story but an unproven claim, diminishing the paper's main appeal.
>
> We appreciate the reviewer’s insightful comment regarding the empirical validation of the learned concepts. We agree that the alignment in real data between machine-learned “concepts” and human-interpretable notions (e.g., white head, black wings) cannot be proved in general, since the true concepts are **inherently unobserved**. However, in our humble points, based on our theorems, the flow modules + reconstruction serve as a key constraint for ensuring that each latent component becomes a faithful concept descriptor, supporting the "concept" framing. We verify this claim by:
>
> **Simulated Experiments:** To address this concern, in our submission, we have included experiments on concept-generating data where the ground-truth concept structure is known. The results in Appendix Table A5 and below show that C$^3$ can successfully recover disentangled latent variables corresponding to these underlying concepts. MCC indicates the extent of latent variable-concept alignment, and Acc. indicates the OCD performance.
>
> *Table: Concept Identifiability Results on Synthetic Data. $n_s$ denotes the number of known categories.*
>
> | $d_z$ | Metric | $n_s=6$ | $n_s=12$ | $n_s=18$ | $n_s=24$ |
> |:------:|:--------|:-------:|:--------:|:--------:|:--------:|
> | **2** | MCC | 0.73 | 0.89 | 0.90 | 0.95 |
> |  | $R^2$ | 62.6 | 75.3 | 90.3 | 95.5 |
> |  | Acc. | 51.3 | 65.7 | 73.6 | 79.8 |
> | **5** | MCC | 0.81 | 0.85 | 0.91 | 0.92 |
> |  | $R^2$ | 73.6 | 77.2 | 79.3 | 89.2 |
> |  | Acc. | 43.5 | 59.9 | 67.0 | 72.2 |
> | **9** | MCC | 0.31 | 0.65 | 0.75 | 0.74 |
> |  | $R^2$ | 12.6 | 67.6 | 82.3 | 86.1 |
> |  | Acc. | 10.9 | 35.5 | 40.4 | 68.4 |
>
> **Visualization Evidences:** Furthermore, while we acknowledge that estimated concept perfect alignment with human semantic understanding may not always be achieved in real data, we try to verify it indirectly: we calculate the similarity of input image feautres with each estimated latent concept, which is a modifeid version of the prototype activation map [3] to show the visualization of our latent concept, and we provided the visualziation results in Figure 4 of the revised verion. The results show that C$^3$ learns the disentangled, semantically meaningful concepts in real data.
>
> **Concept-guided Classification:** We also provide a cross-category concept analysis between *Black-Footed Albatross* and *Sooty Albatross* to further validate our theoretical claims. From their pictures, their "body" and "head" are visually distinct. Then we use the learned concepts to do binary classification, as shown in our revised manuscript (Appendix Table A6) and the table below. Specifically, we separate these concept groups based on the learned Markov network structure.
>
> *Table: Concept-guided Classification.*
> | Concept | Latent Variables | Cls Accuracy (%) | Notes |
> |----------|------------------|------------------|--------|
> | Body | $z_1$, $z_2$ | 92.3 | discriminative latent concepts |
> | Tail | $z_5$ | 64.1 | not discriminative |
> | Wings | $z_3$, $z_4$ | 78.5 | not discriminative |
> | Head | $z_8$ | 95.0 | discriminative latent concepts |
>
> This table means that the distinctive concepts are semantically meaningful and serve as faithful concept descriptors, verifying our theorems and claims on "comparison" in identifying true latent concepts.
>
> [3] Chen, Chaofan, et al. "This looks like that: deep learning for interpretable image recognition." Advances in neural information processing systems 32 (2019).

---

> ### Author Response · Authors · 2025-11-24
> **Author Response (5/5)**
>
> > W4: In the task of discovering general categories, there has already been similar work [4] that decomposes objects into combinations of various attributes (textual or visual) with MoE. I believe there needs to be more comparative discussion with the current work.
>
> We thank the reviewer for pointing out these relevant references. We will enrich our related-work discussion to clarify the key differences.
>
> (1) Although ConGCD [4] also decomposes objects into primitives and uses multiplex consensus between “dominant/contextual” components to improve GCD, it assumes access to novel categories in the training phase and thus is not the genuine generalization setting we target.
> Besides, it does not provide theoretical formalization or identifiability guarantees for the transferring behavior.
> (2) OCRT [5] targets a different problem setting: domain generalization/robustness of foundation models (e.g., CLIP, SAM), not category discovery.
>
> We have added a discussion on the distinction between OCD and these two papers in Line 1040-1047 of our revised version. Also, our framework is orthogonal and could be built on top of foundation encoders, as in OCRT; we provide experimental results on CLIP models here:
>
> *Table: Experimental Results on CLIP models. Comparison of CUB and Stanford Cars Datasets on CLIP*
>
> |                   | **CUB (%)** |           |           | **SCar (%)** |           |           |
> |:------------------|:------------:|:-----------:|:-----------:|:--------------:|:-----------:|:-----------:|
> | **Method**        | All | Old | New | All | Old | New |
> | CLIP (ViT-B/16)   | 33.5 | 58.9 | 21.8 | 26.7 | 52.1 | 14.9 |
> | CLIP (ViT-L/14)   | 28.9 | 51.0 | 14.6 | 22.3 | 45.5 | 10.2 |
> | Ours (DINO ViT-B/16) | 40.1 | 62.1 | 29.5 | 34.1 | 69.0 | 17.8 |
>
> Notably, such empiral techiniques can be found in some prior work, but C$^3$ is the first to formally formulate it as compare-then-compose procedure and provide the idnetifbiality theory on when and how it learn the semantically meaninful concepts and how to generalized them to the unseen, novel category, and we hope that such contribution can help researcher better understand this problem from both thought and implementation.
>
> [4] Dissecting Generalized Category Discovery: Multiplex Consensus under Self-Deconstruction. In ICCV, 2025.
>
> [5] OCRT: Boosting Foundation Models in the Open World with Object-Concept-Relation Triad. In CVPR, 2025.

---

> > ### Comment · Reviewer_LNHk · 2025-11-25
> >
> > Thank the authors for the response.
> >
> > In fact, I would like to engage in a more in-depth discussion with the authors regarding the original motivation of GCD, specifically the need for the model to autonomously identify novel categories.
> >
> >
> > Introducing the connection between **human cognition** and concept formation is a current direction in the development of GCD. If the intention is simply to highlight this point, do the technical details presented in the paper become overly cumbersome, thus losing sight of the exploration of the core motivation and becoming **mired in explanations of technicalities**?

---

> ### Author Response · Authors · 2025-11-26
> **Thank you for your comment. We are more than happy to engage in discussion. (1/2)**
>
> We sincerely appreciate your thoughtful comments and welcome this discussion. Before addressing your concern, we would like to clarify several key points.
>
> **First**, the original GCD setting is _not_ a genuine semantic generalization problem, as novel category data are accessible during training. This makes it an _unfair_ foundation for validating methods that aim to mimic the human cognitive process of generalizable "compare-then-compose." Therefore, we narrow our discussion and evaluation to the OCD setting. Although prior works share similar cognitive inspirations, their methods have not been evaluated under this more rigorous setting.
>
> **Second**, our goal is to formulate a _principled, theoretically grounded_ framework, comparison–composition cognition($C^3$), that aligns with the "compare-then-compose" human cognitive mechanism, with theoretical formulation and guarantees on *machine cognition* as well as practical instantiability. This framework formalizes and investigates the conditions for semantic generalization in the vision domain. The cognitive mechanism of concept formation and transfer is well-supported by seminal cognitive science studies [1,2,3] (see also PD1H W2). In contrast, prior works inspired to compose and generalize to novel semantics cannot guarantee the **learned/transfered concepts correspond to the true ones**, leading to a gap between the machine and human.
>
> **To make machine "cognition" more aligned with humans'**, in this work, we argue that we need to consider in a more rigorous/formal manner whether and when the learned representation/concepts from the known classese are transferred to the unseen categories in the real-world test phase. If the learned representation do not correspond to the meaningful concepts that lies behind our observations, in no way can we guarantee the generalization to OOD novel categories or establish the condition for it.
>
> Our overall logic follows: **Cognitive Principle → Causal Formalization → Theoretical Guarantees → Architectural Instantiation**.
>
> The key insight is this: _faithful_ "compare-then-compose" is not merely an intuitive heuristic—it demands formal answers to a fundamental question: **when and how can concepts learned from known classes transfer to unseen categories?** If learned representations do not recover the true causal concepts underlying observations, generalization to OOD categories becomes _unprincipled_ at best and _unreliable_ at worst. This is not a hypothetical concern: extensive evidence shows that naïve representation transfer often leads to degraded or erroneous predictions [4,7]. In contrast, to the best of our knowledge, existing works—despite being framed around similar cognitive mechanisms—do _not_ formally characterize the faithfulness, learnability, or conditions of such generalization [5,6]. Meanwhile, a myriad of evidence demonstrates that naïve representation transfer can lead to performance degradation or significantly erroneous predictions [4,7]. We view this formalization as an important step toward demystifying the generalization behaviors of vision (foundation) models.
>
> This is precisely why **identifiability theory** lies at the core of our framework. Specifically, our theoretical contributions formalize the entire generalization pipeline:
> - **Theorems 1 & 2**: conditions under which learned representations provably recover _faithful_ (causally grounded) concepts;
> - **Theorem 3**: conditions under which a model can _compositionally_ recognize novel categories from these concepts.
> Existing methods, while invoking similar cognitive intuitions, lack such formal grounding—they neither characterize _what_ makes a representation faithful, nor _when_ compositional generalization succeeds [5,6].
> We see this as an essential step toward a _principled_ understanding of generalization in vision (foundation) models.
> As noted in our response to W1, our implementation is _not_ a collection of ad-hoc tricks. Each module is *logically grounded in a specific theoretical assumption* and serves a clear role in instantiating the overarching cognitive framework. While alternative instantiations certainly exist, the current design choices are principled rather than arbitrary.

---

> > ### Author Response · Authors · 2025-11-26
> > **Thank you for your comment. We are more than happy to engage in discussion. (2/2)**
> >
> > To make this explicit, we now provide a clear mapping from **cognitive principle** to our **theoretical principle**, then guiding **technical component**:
> >
> > | Cognitive Principle | Theoretical Principle | Technical Component / Objective |
> > |:--------------------|:----------------------|:--------------------------------|
> > | **Comparison** — identifying differences across seen categories [1] | Theorem 1 & Theorem 2: Identifiability of concepts under sufficient diversity (A2–A3) | Context-invariance loss $\mathcal{L}_{ctx}$, prototype contrast $\mathcal{L}_{proto}$, conditional flow $\mathcal{L}_{flow}$ |
> > | **Contextual Concept** — contextual factors are category-invariant [8] | A2: Contextual factors capture shared background info stable across labels, Theorem 1: Isolate invariant part from representation | sparsely-gated MoE (selecting context), KL (invariance constraint) |
> > | **Semantic Concept** — semantic are category-varying and disentangled [9] | A3: Semantic diversity in $z$ along with label $y$, Theorem 2: semantic concepts can be individually recovered | $L_{proto}$ for distinct semantics (please see Proposition 1 in our revised version), Flow loss $L_{flow}$ models conditioned variations. |
> > | **Concepts can be dependent** — some concepts may be correlated (arrangement in [3]) | A4: Sparse Markov structure among latent concepts | $L_s$ encourages sparse structure. Flow regularization $L_{flow}$ inversely learns the structure $M$. |
> > | **Human can cognize different concepts** — different concepts are not overlapping  | A5 (Support separation): Concept values (e.g., “striped” vs. “spotted”) occupy distinct semantic regions. | Hash loss $L_{hash}$ center concepts to some ranges and separates them. |
> > | **Human leverages concepts learned from seen categories** — forming new concepts by composition [10] | A6 (Coverage): Unseen classes share semantic bases with seen classes | Compositional inference using a model trained by seens - the rationale underlying the OCD task |
> > | **Concept Composition** — recombining learned concepts to infer unseen ones [3] | Theorem 3: Compositional generalization under support separation (A5) and marginal coverage (A6) | During inference, each $z$ serving as a category descriptor which aligns distinct compositions. |
> >
> > In light of your feedback, we will polish and highlight the motivation and design principles, clarify the assumption → module mapping, and move derivations/details to the appendix where appropriate, to clarify *autonomous concept discovery via compare-then-compose* as the pivot. We acknowledge that the current Method section could be clearer. A revised version with improved organization will be provided in the updated manuscript.
> >
> >
> >
> >
> > [1] Gentner, D. (1983).  *Structure-Mapping: A Theoretical Framework for Analogy.* Cognitive Science.
> >
> > [2] Marr, D. (1982).  *Vision: A Computational Investigation into the Human Representation and Processing of Visual Information.*
> >
> > [3] Biederman, I. (1987).  *Recognition-by-Components: A Theory of Human Image Understanding.* Psychological Review.
> >
> > [4] Udandarao, Vishaal, et al. "No" zero-shot" without exponential data: Pretraining concept frequency determines multimodal model performance." _Advances in Neural Information Processing Systems_ 37 (2024): 61735-61792.
> >
> > [5] Dissecting Generalized Category Discovery: Multiplex Consensus under Self-Deconstruction. In ICCV, 2025.
> >
> > [6] OCRT: Boosting Foundation Models in the Open World with Object-Concept-Relation Triad. In CVPR, 2025.
> >
> > [7] Geirhos et al. (2020). Shortcut learning in deep neural networks. Nature Machine Intelligence
> >
> > [8] Cao, Runnan, et al. "A human single-neuron dataset for object recognition." Scientific data 12.1 (2025): 79.
> >
> > [9] Saban, William, et al. "Primitive visual channels have a causal role in cognitive transfer." Scientific reports 11.1 (2021): 8759.
> >
> > [10] Vyas, Maunil R., Hemanth Venkateswara, and Sethuraman Panchanathan. "Leveraging seen and unseen semantic relationships for generative zero-shot learning." European conference on computer vision. Cham: Springer International Publishing, 2020.

---

### Author Response · Authors · 2025-11-24
**General Response (Mirroring Assumptions to Implementation Modules)**

Our implementation and objective design are explicitly and directly designed to encourage the same geometric structure of latent variables as assumed. Here is a complete item-to-item mapping specification:

- **A1 (Well-posed density)**:

(i) Standard VAE / ELBO-based modeling (Eq. 10) ensures a computable joint density over $(\hat z,\hat c \mid x)$ and their conditional/marginal distributions.

(ii) conditional normalizing flow with $L_{\text{flow}}$ (Eq. 7&8) yields a smooth, strictly positive density over the semantic subspace.
- **A2 (Context comparison)** : Identifiability of $c$ without semantic information.

(i) We enforce semantic invarance on $\hat c$ with contextual loss (Eq. 4), while label-discriminative variation is forced into $\hat z$, separated by the dynamic MoE gating $M(s)$ via Bernoulli mask (Eq. 6).

(ii) Also refer to A3 below.

- **A3 (Semantic comparison)**: The local “shape vectors” (gradients, curvatures, selected mixed derivatives of log-prob) are linearly independent across labels in affine sense, for semantic concept identifiability.

(i) We include prototype-based supervised loss $L_{\text{proto}}$, with multiple prototypes per class, makes $\hat z$ semantic-discriminative and span across multiple directions.

(ii) We include conditional flow with $L_{\text{flow}}$ (Eq. 7&8), which provides an explicit, label-dependent parametrization of $\log p(\hat z\mid y)$.

- **A4 (Sparse Markov structure)**: The latent Markov network over semantic concepts is sparse and with few modular dependencies.

(i) sparsely-gated MoE to learn a sparse adjacency semantic structure.

(ii) $\ell_1$ sparsity loss $L_s$ (Eq. 5) on $\hat z$ to keep semantic sparsity.

(iii) flow regularization restricted to $M_z\odot \hat z$, suggesting only edges selected by $M_z$ can induce non-zero mixed derivatives.

- **A5 (Support separation)**:

(i) Approximated by hash loss $L_{\text{hash}}$ that pushes discrete hash codes of different semantics apart.

(ii) Approximated by prototype-based $L_{\text{proto}}$ that pulls samples with different concept compositions to distinct regions.
Together they make different concept values occupy separated regions in the semantic manifold.

- **A6 (Coverage)**: basic assumption that all semantic concepts of unseen categories appear among the seen semantics, aligns with cognitive principles of composing reusable primitives.

---

### Author Response · Authors · 2025-11-24
**General Response (Overall)**

We thank all the reviewers for the positive feedback, including:

**Theoretical Foundation and Novelty**
- Our proposed C3 framework is _novel, principled_ and _rigorously theoretically-grounded_ with _formal identifiability guarantees_ (**LNHk**, **Tdof**, **PD1H**).
- The comparison-composition cognition metaphor is _interesting and well-aligned_ with cognitive science, _mimicking_ the human cognitive process of unseen category recognition/generalization (**LNHk**, **PD1H**).
- Theoretical assumptions are derived through _clear stepwise logic_ with _reasonable formulations_ that establish _provable guarantees and conditions_ for confident category generalization (**MWi7**, **PD1H**).
- _Every_ theoretical assumption is _explicitly operationalized_ in the practical design, ensuring _tight_ theory-implementation alignment (**PD1H**).

**Practical Significance**
- The _problem domain_ we address is _important and challenging_ in machine learning where current models _still struggle_, making it _worthy of rigorous investigation_ (**MWi7**).
- Instantiating our theory on this difficult problem demonstrates both _applicability_ and _practical value_ of our approach (**Tdof**).

**Empirical Validation**
- Experiments demonstrate strong performance on extensive public benchmarks with _notable improvements over prior SOTAs_, revealing the _practical value_ of our proposed theoretically-grounded approach (**MWi7, Tdof**).
- Comprehensive ablation studies _validate key design choices_ and help readers understand how each component contributes to overall effectiveness (**MWi7**).

Our **code & models** will be publicly released upon acceptance. All suggested changes will be reflected in the revised and final version.

---

### Author Response · Authors · 2025-12-01
**Summary of Major Rebuttal Clarifications**

Dear AC and Reviewers,

We sincerely thank all reviewers for their insightful feedback and thank the AC for the time and effort devoted to handling our paper. We are grateful that many comments recognized the strengths of our work. Our paper received initial scores of  **6,4,4,4**, and did not undergo in-depth discussion due to the early closure of reviewer comments. Below, we summarize how our rebuttal and revisions addressed all concerns raised by the reviewers.

---

**1. Strengthening the theory–implementation connection (LNHk W1–W2, PD1H W1–W3, MWi7 W1/W4)**
We added a clear *Assumption → Intuition → Module* mapping and reorganized the method section. We clarified that A1–A6 are population-level identifiability assumptions, while our modules (flow, sparse MoE, prototype contrast, hashing, context invariance) are **operational surrogates** encouraging the same geometric structure required by the theory. We added geometric explanations for A2/A3 and the role of log-density derivatives.

[**Related Revisions**] Added a theoretical guarantee (`Proposition 1`), a theory–implementation table (`Table A1`), and a geometric explanation (`Lines 1350–1352`).

**2. Providing stronger empirical evidence of concept interpretability (LNHk W3, PD1H Q4, MWi7 W3)**
To verify that C$^3$ learns the true interpretable concepts, we included:
- **concept activation visualizations** on real images,
- **synthetic concept-identifiability experiments**, and
- **concept-guided classification** on real images.
These results demonstrate that learned latent dimensions correspond to semantically meaningful parts (e.g., head/body), consistent with theoretical guarantees.

[**Related Revisions**] Added a visualization of concepts (`Figure 4`), a syntheic experiments (`Table A5`), and a concept-guided classification (`Table A6`).

**3. Clarifying cognitive-science motivations (LNHk comment, PD1H W2)**
We expanded the introduction and theoretical framing to explicitly connect C$^3$ to seminal cognitive theories on *compare-then-compose* mechanism: structure-mapping theory (comparison), Marr/Biederman primitives (composition), and evidences that semantic differentiation precedes composition in human cognition.

[**Related Revisions**] Added a motivation (`Lines 038-058, 066-067`), syntheic experiments (`Table A5`), and a concept-guided classification (`Table A6`).

**4. Addressing scalability, overhead, and robustness (Tdof W2–W3, PD1H W5/Q2–Q3, MWi7 W1)**
We added:
- computational overhead analyses (parameters, runtime, memory),
- convergence/stability curves,
- scaling experiments with **CLIP** backbones,
- semantic scaling experiments on **more unseen classes** and **larger iNaturalist subsets**,
- ablations on effects of comparison & composition.
These results demonstrate C$^3$ scales effectively across various dimensions with minor computational overhead.

[**Related Revisions**]
Added computational-overhead analysis (`Table A11`), convergence/stability curves (`Figure A2`), CLIP-scaling experiments (`Table A8`), scaling experiments on unseen classes (`Table A9`), and comparison–composition ablations (`Table A10`).

**5. Clarifying formulation, definitions, and notation (MWi7 W4)**
We have clarified the exposition of the generative process, basis distribution, semantic Markov network, contextual/semantic representation, and the OCD setting definition. Missing details were moved from appendix into the main paper.

[**Related Revisions**] Clarified the generative process (`Eq. 1`, p. 3), added intuitive explanation of the latent Markov network (`Definition 1`, p. 3), refined the contextual/semantic description (`Sec. 4.1`, pp. 6–7).

**6. Discussing assumptions and limitations (PD1H Q1, MWi7 W2)**
We clarified that A6 (coverage) is a fundamental requirement for identifiability. If unseen categories purely differ on concept dimensions absent from the seen data, the problem becomes ill-posed. We added sensitivity experiments to approximate violations of A2–A5.

[**Related Revisions**] Added sensitivity experiments (`Table 2`, p. 10), clarified the necessity of A6 (coverage) in `Section 3`, and included a discussion of identifiability limits (`pp. 10`).


---

### **Reviewer Follow-up**

Reviewer LNHk raised an additional question regarding whether the technical depth of our formulation might obscure the core cognitive motivation. We addressed this by clarifying the *cognitive–theoretical–computational* pipeline of C$^3$ and reinforcing how our identifiability results directly operationalize the cognitive mechanism rather than diverging into unrelated technicalities.

Hence, we believe that these revisions substantially enhance the theoretical framing and empirical validation of our work, fully addressing all reviewer concerns.

---

We again sincerely thank the AC and reviewers for their constructive engagement, thoughtful suggestions, and support.

Best regards,

*The Authors of Submission 9939*

---

### Meta-Review · Area_Chair_Y7qc · 2025-12-22

**Summary:**

This paper aims to formalizes the compare-then-compose mechanism for on-the-fly category discovery. The primary contribution of the paper is to establish the identifiable guarantee under certain assumptions and instantiate this with appropriate inductive biases.

The initial recommendations are generally negative with one weak accept and three weak rejects. The primary concerns from the reviewers include: (1) weak connection between the theory and implementation, (2) insufficient empirical validation that the learned latent corresponds to the interpretable concepts, (3) overly complex and not self-contained explanations, especially on theory, and (4) concerns on scalability, ablation, and efficiency.

**Reviewer Concerns:**

Largely addressed concerns:
- concerns on scalability and efficiency are largely addressed by the additional experiments provided in the rebuttal.
- clarifications on inductive biases and the assumptions are quite helpful to understand the design choices of the proposed method.

Outstanding concerns:
-  The rebuttal argues that the identifiability assumptions are mild and that the implementation acts as an operational surrogate. However, several key assumptions are substantially stronger than claimed and appear misaligned with the paper's experiment settings. For instance, A2 and A3 require existence of many distinct labels satisfying linear independence on log-density derivatives, e.g., $d_z+1$ or $2d_z+|M|+1$, which are likely not hold in the experiment settings and potentially many real-world applications with sufficiently large $d_z$ e.g., $d_z=3071$ in the paper. Also, while the inductive biases are carefully chosen to satisfy other assumptions such as A4-6, the authors did not validate if these assumptions hold under their components.
- While the rebuttal provide additional experiment and visualization to validate that the learned latents are aligned with interpretable concepts, they are still limited in scope e.g., synthetic settings or a few qualitative results, which are insufficient to comprehensively validate the claim.

**Reviewer Scores:**

Given these, while the new results and clarifications in the rebuttal are helpful to improve the paper, the AC believes that it is still insufficient to overturn the original review. Hence, the AC recommends rejection this time.

---

### Decision · Program_Chairs · 2026-01-26

Reject